# Momentum Centering and Asynchronous Update for Adaptive Gradient Methods

**Juntang Zhuang**[1]; **Yifan Ding**[2]; **Tommy Tang**[3]; **Nicha Dvornek**[1];
**Sekhar Tatikonda**[1]; **James S. Duncan**[1]
[1] Yale University; [2] University of Central Florida; [3] University of Illinois at Urbana-Champaign
`{j.zhuang;nicha.dvornek;sekhar.tatikonda;james.duncan}@yale.edu;`
`yf.ding@knights.ucf.edu;tommymt2@illinois.edu`

## Abstract

We propose ACProp (Asynchronous-centering-Prop), an adaptive optimizer which combines centering of second momentum and asynchronous update (e.g. for $t$-th update, denominator uses information up to step $t-1$, while numerator uses gradient at $t$-th step). ACProp has both strong theoretical properties and empirical performance. With the example by Reddi et al. (2018), we show that asynchronous optimizers (e.g. AdaShift, ACProp) have weaker convergence condition than synchronous optimizers (e.g. Adam, RMSProp, AdaBelief); within asynchronous optimizers, we show that centering of second momentum further weakens the convergence condition. We demonstrate that ACProp has a convergence rate of $O(\frac{1}{\sqrt{T}})$ for the stochastic non-convex case, which matches the oracle rate and outperforms the $O(\frac{logT}{\sqrt{T}})$ rate of RMSProp and Adam. We validate ACProp in extensive empirical studies: ACProp outperforms both SGD and other adaptive optimizers in image classification with CNN, and outperforms well-tuned adaptive optimizers in the training of various GAN models, reinforcement learning and transformers. To sum up, ACProp has good theoretical properties including weak convergence condition and optimal convergence rate, and strong empirical performance including good generalization like SGD and training stability like Adam. We provide the implementation at `https://github.com/juntang-zhuang/ACProp-Optimizer`.

## 1 Introduction

Deep neural networks are typically trained with first-order gradient optimizers due to their computational efficiency and good empirical performance [1]. Current first-order gradient optimizers can be broadly categorized into the stochastic gradient descent (SGD) [2] family and the adaptive family. The SGD family uses a global learning rate for all parameters, and includes variants such as Nesterov-accelerated SGD [3], SGD with momentum [4] and the heavy-ball method [5]. Compared with the adaptive family, SGD optimizers typically generalize better but converge slower, and are the default for vision tasks such as image classification [6], object detection [7] and segmentation [8].

The adaptive family uses element-wise learning rate, and the representatives include AdaGrad [9], AdaDelta [10], RMSProp [11], Adam [12] and its variants such as AdamW [13], AMSGrad [14] AdaBound [15], AdaShift [16], RAdam [17] and AdaBelief [18]. Compared with the SGD family, the adaptive optimizers typically converge faster and are more stable, hence are the default for generative adversarial networks (GANs) [19], transformers [20], and deep reinforcement learning [21].

We broadly categorize adaptive optimizers according to different criteria, as in Table. 1. (a) *Centered v.s. uncentered* Most optimizers such as Adam and AdaDelta uses uncentered second momentum in the denominator; RMSProp-center [11], SDProp [22] and AdaBelief [18] use square root of centered

Table 1: Categories of adaptive optimizers

| | Uncentered second momentum | Centered second momentum |
|---|---|---|
| Synchronous | Adam , RAdam, AdaDelta, RMSProp | RMSProp-center, SDProp, AdaBelief |
| Asynchronous | AdaShift | ACProp (ours) |

second momentum in the denominator. AdaBelief [18] is shown to achieve good generalization like the SGD family, fast convergence like the adaptive family, and training stability in complex settings such as GANs. (b) *Sync vs async* The synchronous optimizers typically use gradient $g_t$ in both numerator and denominator, which leads to correlation between numerator and denominator; most existing optimizers belong to this category. The asynchronous optimizers decorrelate numerator and denominator (e.g. by using $g_t$ as numerator and use $\{g_0, ... g_{t-1}\}$ in denominator for the $t$-th update), and is shown to have weaker convergence conditions than synchronous optimizers[16].

We propose Asynchronous Centering Prop (ACProp), which combines centering of second momentum with the asynchronous update. We show that ACProp has both good theoretical properties and strong empirical performance. Our contributions are summarized as below:

- **Convergence condition** *(a) Async vs Sync* We show that for the example by Reddi et al. (2018), asynchronous optimizers (AdaShift, ACProp) converge for any valid hyper-parameters, while synchronous optimizers (Adam, RMSProp et al.) could diverge if the hyper-paramaters are not carefully chosen. *(b) Async-Center vs Async-Uncenter* Within the asynchronous optimizers family, by example of an online convex problem with sparse gradients, we show that Async-Center (ACProp) has weaker conditions for convergence than Async-Uncenter (AdaShift).

- **Convergence rate** We demonstrate that ACProp achieves a convergence rate of $O(\frac{1}{\sqrt{T}})$ for stochastic non-convex problems, matching the oracle of first-order optimizers [23], and outperforms the $O(\frac{logT}{\sqrt{T}})$ rate of Adam and RMSProp.

- **Empirical performance** We validate performance of ACProp in experiments: on image classification tasks, ACProp outperforms SGD and AdaBelief, and demonstrates good generalization performance; in experiments with transformer, reinforcement learning and various GAN models, ACProp outperforms well-tuned Adam, demonstrating high stability. ACProp often outperforms AdaBelief, and achieves good generalization like SGD and training stability like Adam.

## 2 Overview of algorithms

### 2.1 Notations

$x, x_t \in \mathbb{R}^d$: $x$ is a $d-$dimensional parameter to be optimized, and $x_t$ is the value at step $t$.

$f(x), f^* \in \mathbb{R}$: $f(x)$ is the scalar-valued function to be minimized, with optimal (minimal) $f^*$.

$\alpha_t, \epsilon \in \mathbb{R}$: $\alpha_t$ is the learning rate at step $t$. $\epsilon$ is a small number to avoid division by 0.

$g_t \in \mathbb{R}^d$: The noisy observation of gradient $\nabla f(x_t)$ at step $t$.

$\beta_1, \beta_2 \in \mathbb{R}$: Constants for exponential moving average, $0 \leq \beta_1, \beta_2 < 1$.

$m_t \in \mathbb{R}^d$: $m_t = \beta_1 m_{t-1} + (1 - \beta_1)g_t$. The Exponential Moving Average (EMA) of observed gradient at step $t$.

$\Delta g_t \in \mathbb{R}^d$: $\Delta g_t = g_t - m_t$. The difference between observed gradient $g_t$ and EMA of $g_t$.

$v_t \in \mathbb{R}^d$: $v_t = \beta_2 v_{t-1} + (1 - \beta_2)g_t^2$. The EMA of $g_t^2$.

$s_t \in \mathbb{R}^d$: $s_t = \beta_2 s_{t-1} + (1 - \beta_2)(\Delta g_t)^2$. The EMA of $(\Delta g_t)^2$.

### 2.2 Algorithms

In this section, we summarize the AdaBelief [18] method in Algo. 1 and ACProp in Algo. 2. For the ease of notations, all operations in Algo. 1 and Algo. 2 are element-wise, and we omit the bias-correction step of $m_t$ and $s_t$ for simplicity. $\Pi_{\mathcal{F}}$ represents the projection onto feasible set $\mathcal{F}$.

We first introduce the notion of "sync (async)" and "center (uncenter)". *(a) Sync vs Async* The update on parameter $x_t$ can be generally split into a numerator (e.g. $m_t, g_t$) and a denominator (e.g.

| Algorithm 1: AdaBelief | Algorithm 2: ACProp |
|---|---|
| **Initialize** $x_0, m_0 \leftarrow 0, s_0 \leftarrow 0, t \leftarrow 0$ | **Initialize** $x_0, m_0 \leftarrow 0, s_0 \leftarrow 0, t \leftarrow 0$ |
| **While** $x_t$ not converged | **While** $x_t$ not converged |
| $\quad t \leftarrow t + 1$ | $\quad t \leftarrow t + 1$ |
| $\quad g_t \leftarrow \nabla_x f_t(x_{t-1})$ | $\quad g_t \leftarrow \nabla_x f_t(x_{t-1})$ |
| $\quad m_t \leftarrow \beta_1 m_{t-1} + (1 - \beta_1) g_t$ | $\quad m_t \leftarrow \beta_1 m_{t-1} + (1 - \beta_1) g_t$ |
| $\quad s_t \leftarrow \beta_2 s_{t-1} + (1 - \beta_2)(g_t - m_t)^2$ | $\quad x_t \leftarrow \prod_{\mathcal{F}, \sqrt{s_{t-1}}} \left( x_{t-1} - \frac{\alpha}{\sqrt{s_{t-1}+\epsilon}} g_t \right)$ |
| $\quad x_t \leftarrow \prod_{\mathcal{F}, \sqrt{s_t}} \left( x_{t-1} - \frac{\alpha}{\sqrt{s_t+\epsilon}} m_t \right)$ | $\quad s_t \leftarrow \beta_2 s_{t-1} + (1 - \beta_2)(g_t - m_t)^2$ |

$\sqrt{s_t}, \sqrt{v_t}$). We call it "sync" if the denominator depends on $g_t$, such as in Adam and RMSProp; and call it "async" if the denominator is independent of $g_t$, for example, denominator uses information up to step $t-1$ for the $t$-th step. *(b) Center vs Uncenter* The "uncentered" update uses $v_t$, the exponential moving average (EMA) of $g_t^2$; while the "centered" update uses $s_t$, the EMA of $(g_t - m_t)^2$.

**Adam (Sync-Uncenter)** The Adam optimizer [12] stores the EMA of the gradient in $m_t$, and stores the EMA of $g_t^2$ in $v_t$. For each step of the update, Adam performs element-wise division between $m_t$ and $\sqrt{v_t}$. Therefore, the term $\alpha_t \frac{1}{\sqrt{v_t}}$ can be viewed as the element-wise learning rate. Note that $\beta_1$ and $\beta_2$ are two scalars controlling the smoothness of the EMA for the first and second moment, respectively. When $\beta_1 = 0$, Adam reduces to RMSProp [24].

**AdaBelief (Sync-Center)** AdaBelief optimizer [18] is summarized in Algo. 1. Compared with Adam, the key difference is that it replaces the uncentered second moment $v_t$ (EMA of $g_t^2$) by an estimate of the centered second moment $s_t$ (EMA of $(g_t - m_t)^2$). The intuition is to view $m_t$ as an estimate of the expected gradient: if the observation $g_t$ deviates much from the prediction $m_t$, then it takes a small step; if the observation $g_t$ is close to the prediction $m_t$, then it takes a large step.

**AdaShift (Async-Uncenter)** AdaShift [16] performs temporal decorrelation between numerator and denominator. It uses information of $\{g_{t-n}, ...g_t\}$ for the numerator, and uses $\{g_0, ...g_{t-n-1}\}$ for the denominator, where $n$ is the "delay step" controlling where to split sequence $\{g_i\}_{i=0}^t$. The numerator is independent of denominator because each $g_i$ is only used in either numerator or denominator.

**ACProp (Async-Center)** Our proposed ACProp is the asynchronous version of AdaBelief and is summarized in Algo. 2. Compared to AdaBelief, the key difference is that ACProp uses $s_{t-1}$ in the denominator for step $t$, while AdaBelief uses $s_t$. Note that $s_t$ depends on $g_t$, while $s_{t-1}$ uses history up to step $t-1$. This modification is important to ensure that $\mathbb{E}(g_t/\sqrt{s_{t-1}}|g_0, ...g_{t-1}) = (\mathbb{E}g_t)/\sqrt{s_{t-1}}$. It's also possible to use a delay step larger than 1 similar to AdaShift, for example, use $EMA(\{g_i\}_{i=t-n}^t)$ as numerator, and $EMA(\{(g_i - m_i)^2\}_{i=0}^{t-n-1})$ for denominator.

## 3 Analyze the conditions for convergence

We analyze the convergence conditions for different methods in this section. We first analyze the counter example by Reddi et al. (2018) and show that async-optimizers (AdaShift, ACProp) always converge $\forall \beta_1, \beta_2 \in (0, 1)$, while sync-optimizers (Adam, AdaBelief, RMSProp et al.) would diverge if $(\beta_1, \beta_2)$ are not carefully chosen; hence, async-optimizers have weaker convergence conditions than sync-optimizers. Next, we compare async-uncenter (AdaShift) with async-center (ACProp) and show that momentum centering further weakens the convergence condition for sparse-gradient problems. Therefore, ACProp has weaker convergence conditions than AdaShift and other sync-optimizers.

### 3.1 *Sync vs Async*

We show that for the example in [14], async-optimizers (ACProp, AdaShift) have weaker convergence conditions than sync-optimizers (Adam, RMSProp, AdaBelief).

**Lemma 3.1** (Thm.1 in [14]). *There exists an online convex optimization problem where sync-optimizers (e.g. Adam, RMSProp) have non-zero average regret, and one example is*

$$f_t(x) = \begin{cases} Px, & if \quad t\%P = 1 \\ -x, & Otherwise \end{cases} \quad x \in [-1, 1], P \in \mathbb{N}, P \geq 3 \tag{1}$$

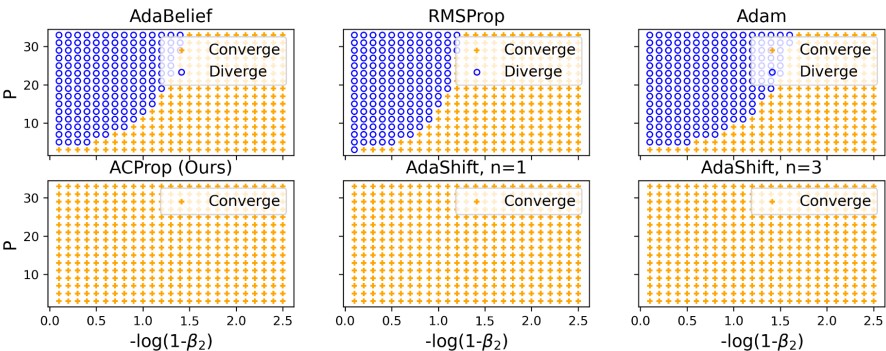

Figure 1: Numerical results for the example defined by Eq. (1). We set the initial value as $x_0 = 0$, and run each optimizer for $10^4$ steps trying different initial learning rates in $\{10^{-5}, 10^{-4}, 10^{-3}, 10^{-2}, 10^{-1}, 1.0\}$, and set the learning rate decays with $1/\sqrt{t}$. If there's a proper initial learning rate, such that the average distance between the parameter and its optimal value $x^* = -1$ for the last 1000 steps is below 0.01, then it's marked as "converge" (orange plus symbol), otherwise as "diverge" (blue circle). For each optimizer, we sweep through different $\beta_2$ values in a log grid ($x$-axis), and sweep through different values of $P$ in the definition of problem ($y$-axis). We plot the result for $\beta_1 = 0.9$ here; for results with different $\beta_1$ values, please refer to appendix. Our results indicate that in the $(P, \beta_2)$ plane, there's a threshold curve beyond which sync-optimizers (Adam, RMSProp, AdaBelief) will diverge; however, async-optimizers (ACProp, AdaShift) always converge for any point in the $(P, \beta_2)$ plane. Note that for AdaShift, a larger delay step $n$ is possible to cause divergence (see example in Fig. 2 with $n = 10$). To validate that the "divergence" is not due to numerical issues and sync-optimizers are drifting away from optimal, we plot trajectories in Fig. 2

**Lemma 3.2** ([25]). *For problem (1) with any fixed P, there's a threshold of $\beta_2$ above which RMSProp converges.*

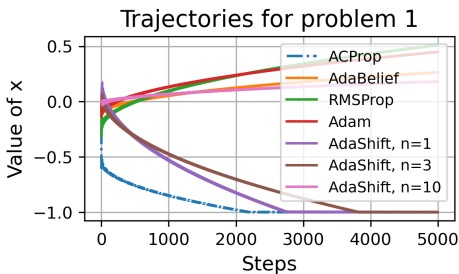

Figure 2: Trajectories of $x$ for different optimizers in Problem by Eq. 1. Initial point is $x_0 = 0$, the optimal is $x^* = -1$, the trajectories show that sync-optimizers (Adam, AdaBelief, RMSProp) diverge from the optimal, validating the divergent area in Fig. 1 is correct rather than artifacts of numerical issues. Async-optimizers (ACProp, AdaShift) converge to optimal value, but large delay step $n$ in AdaShift could cause non-convergence.

In order to better explain the two lemmas above, we conduct numerical experiments on the problem by Eq. (1), and show results in Fig. 1. Note that $\sum_{t=k}^{k+P} f_t(x) = x$, hence the optimal point is $x^* = -1$ since $x \in [-1, 1]$. Starting from initial value $x_0 = 0$, we sweep through the plane of $(P, \beta_2)$ and plot results of convergence in Fig. 1, and plot example trajectories in Fig. 2.

Lemma. 3.1 tells half of the story: looking at each vertical line in the subfigure of Fig. 1, that is, for each fixed hyper-parameter $\beta_2$, there exists sufficiently large $P$ such that Adam (and RMSProp) would diverge. Lemma 3.2 tells the other half of the story: looking at each horizontal line in the subfigure of Fig. 1, for each problem with a fixed period $P$, there exists sufficiently large $\beta_2$s beyond which Adam can converge.

The complete story is to look at the $(P, \beta_2)$ plane in Fig. 1. There is a boundary between convergence and divergence area for sync-optimizers (Adam, RMSProp, AdaBelief), while async-optimizers (ACProp, AdaShift) always converge.

**Lemma 3.3.** *For the problem defined by Eq.* (1)*, using learning rate schedule of $\alpha_t = \frac{\alpha_0}{\sqrt{t}}$, async-optimizers (ACProp and AdaShift with $n = 1$) always converge $\forall \beta_1, \beta_2 \in (0, 1), \forall P \in \mathbb{N}, P \geq 3$.*

The proof is in the appendix. Note that for AdaShift, proof for the always-convergence property only holds when $n = 1$; larger $n$ could cause divergence (e.g. $n = 10$ causes divergence as in Fig. 2). The always-convergence property of ACProp and AdaShift comes from the un-biased stepsize, while the stepsize for sync-optimizers are biased due to correlation between numerator and denominator. Taking RMSProp as example of sync-optimizer, the update is $-\alpha_t \frac{g_t}{\sqrt{v_t}} = -\alpha_t \frac{g_t}{\sqrt{\beta_2^t g_0^2 + \ldots + \beta_2 g_{t-1}^2 + g_t^2}}$. Note that $g_t$ is used both in the numerator and denominator, hence a large $g_t$ does not necessarily

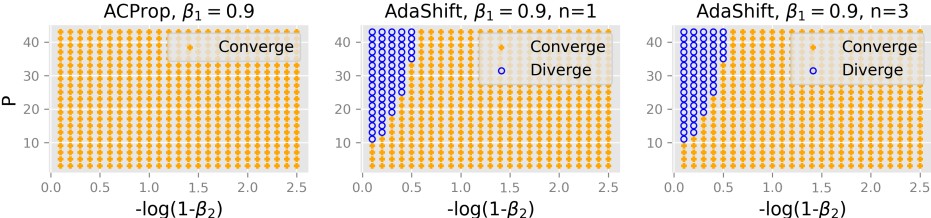

Figure 3: Area of convergence for the problem in Eq. (2). The numerical experiment is performed under the same setting as in Fig. 1. Our results experimentally validated the claim that compared with async-uncenter (AdaShift), async-center (ACProp) has a larger convergence area in the hyper-parameter space.

generate a large stepsize. For the example in Eq. (1), the optimizer observes a gradient of $-1$ for $P-1$ times and a gradient of $P$ once; due to the biased stepsize in sync-optimizers, the gradient of $P$ does not generate a sufficiently large stepsize to compensate for the effect of wrong gradients $-1$, hence cause non-convergence. For async-optimizers, $g_t$ is not used in the denominator, therefore, the stepsize is not biased and async-optimizers has the always-convergence property.

**Remark** Reddi et al. (2018) proposed AMSGrad to track the element-wise maximum of $v_t$ in order to achieve the always-convergence property. However, tracking the maximum in the denominator will in general generate a small stepsize, which often harms empirical performance. We demonstrate this through experiments in later sections in Fig. 6.

### 3.2 Async-Uncenter vs Async-Center

In the last section, we demonstrated that async-optimizers have weaker convergence conditions than sync-optimizers. In this section, within the async-optimizer family, we analyze the effect of centering second momentum. We show that compared with async-uncenter (AdaShift), async-center (ACProp) has weaker convergence conditions. We consider the following online convex problem:

$$f_t(x) = \begin{cases} P/2 \times x, & t\%P == 1 \\ -x, & t\%P == P-2 \\ 0, & otherwise \end{cases} \quad P > 3, P \in \mathbb{N}, x \in [0,1]. \tag{2}$$

Initial point is $x_0 = 0.5$. Optimal point is $x^* = 0$. We have the following results:

**Lemma 3.4.** *For the problem defined by Eq. (2), consider the hyper-parameter tuple $(\beta_1, \beta_2, P)$, there exists cases where ACProp converges but AdaShift with $n = 1$ diverges, but not vice versa.*

We provide the proof in the appendix. Lemma. 3.4 implies that ACProp has a larger area of convergence than AdaShift, hence the centering of second momentum further weakens the convergence conditions. We first validate this claim with numerical experiments in Fig. 3; for sanity check, we plot the trajectories of different optimizers in Fig. 4. We observe that the convergence of AdaShift is influenced by delay step $n$, and there's no good criterion to select a good value of $n$, since Fig. 2 requires a small $n$ for convergence in problem (1), while Fig. 4 requires a large $n$ for convergence in problem (2). ACProp has a larger area of convergence, indicating that both async update and second momentum centering helps weaken the convergence conditions.

We provide an intuitive explanation on why momentum centering helps convergence. Due to the periodicity of the problem, the optimizer behaves almost periodically as $t \rightarrow \infty$. Within each period, the optimizer observes one positive gradient $P/2$ and one negative gradient -1. As in Fig. 5, between observing non-zero gradients, the gradient is always 0. Within each period, ACprop will perform a positive update $P/(2\sqrt{s^+})$ and a negative update $-1/\sqrt{s^-}$, where $s^+$ ($s^-$) is the value of denominator before observing positive (negative) gradient. Similar notations for $v^+$ and $v^-$ in AdaShift. A net update in the correct direction requires $\frac{P}{2\sqrt{s^+}} > \frac{1}{\sqrt{s^-}}$, (or $s^+/s^- < P^2/4$).

When observing 0 gradient, for AdaShift, $v_t = \beta_2 v_{t-1} + (1-\beta_2)0^2$; for ACProp, $s_t = \beta_2 s_{t-1} + (1-\beta_2)(0-m_t)^2$ where $m_t \neq 0$. Therefore, $v^-$ decays exponentially to 0, but $s^-$ decays to a non-zero constant, hence $\frac{s^+}{s^-} < \frac{v^+}{v^-}$, hence ACProp is easier to satisfy $s^+/s^- < P^2/4$ and converge.

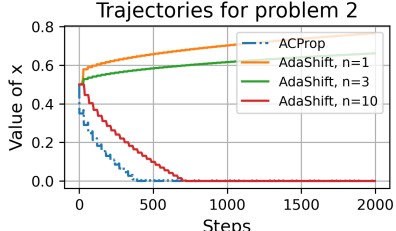
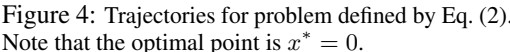
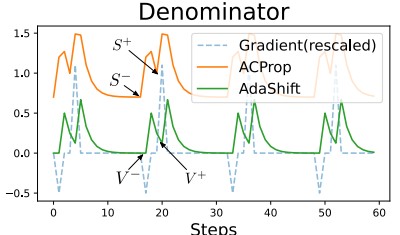

Figure 4: Trajectories for problem defined by Eq. (2). Note that the optimal point is $x^* = 0$.

Figure 5: Value of uncentered second momentum $v_t$ and centered momentum $s_t$ for problem (2).

## 4  Analysis on convergence rate

In this section, we show that ACProp converges at a rate of $O(1/\sqrt{T})$ in the stochastic nonconvex case, which matches the oracle [23] for first-order optimizers and outperforms the $O(logT/\sqrt{T})$ rate for sync-optimizers (Adam, RMSProp and AdaBelief) [26, 25, 18]. We further show that the upper bound on regret of async-center (ACProp) outperforms async-uncenter (AdaShift) by a constant.

For the ease of analysis, we denote the update as: $x_t = x_{t-1} - \alpha_t A_t g_t$, where $A_t$ is the diagonal preconditioner. For SGD, $A_t = I$; for sync-optimizers (RMSProp), $A_t = \frac{1}{\sqrt{v_t}+\epsilon}$; for AdaShift with $n = 1$, $A_t = \frac{1}{\sqrt{v_{t-1}}+\epsilon}$; for ACProp, $A_t = \frac{1}{\sqrt{s_{t-1}}+\epsilon}$. For async optimizers, $\mathbb{E}[A_t g_t | g_0, ... g_{t-1}] = A_t \mathbb{E} g_t$; for sync-optimizers, this does not hold because $g_t$ is used in $A_t$

**Theorem 4.1** (convergence for stochastic non-convex case). *Under the following assumptions:*

- *$f$ is continuously differentiable, $f$ is lower-bounded by $f^*$ and upper bounded by $M_f$. $\nabla f(x)$ is globally Lipschitz continuous with constant $L$:*

$$||\nabla f(x) - \nabla f(y)|| \leq L||x - y|| \tag{3}$$

- *For any iteration $t$, $g_t$ is an unbiased estimator of $\nabla f(x_t)$ with variance bounded by $\sigma^2$. Assume norm of $g_t$ is bounded by $M_g$.*

$$\mathbb{E}[g_t] = \nabla f(x_t) \quad \mathbb{E}[||g_t - \nabla f(x_t)||^2] \leq \sigma^2 \tag{4}$$

*then for $\beta_1, \beta_2 \in [0, 1)$, with learning rate schedule as: $\alpha_t = \alpha_0 t^{-\eta}$, $\alpha_0 \leq \frac{C_l}{LC_u^2}$, $\eta \in [0.5, 1)$ for the sequence $\{x_t\}$ generated by ACProp, we have*

$$\frac{1}{T}\sum_{t=1}^{T}\left|\left|\nabla f(x_t)\right|\right|^2 \leq \frac{2}{C_l}\left[(M_f - f^*)\alpha_0 T^{\eta-1} + \frac{LC_u^2\sigma^2\alpha_0}{2(1-\eta)}T^{-\eta}\right] \tag{5}$$

*where $C_l$ and $C_u$ are scalars representing the lower and upper bound for $A_t$, e.g. $C_l I \preceq A_t \preceq C_u I$, where $A \preceq B$ represents $B - A$ is semi-positive-definite.*

Note that there's a natural bound for $C_l$ and $C_u$: $C_u \leq \frac{1}{\epsilon}$ and $C_l \geq \frac{1}{2M_g}$ because $\epsilon$ is added to denominator to avoid division by 0, and $g_t$ is bounded by $M_g$. Thm. 4.1 implies that ACProp has a convergence rate of $O(1/\sqrt{T})$ when $\eta = 0.5$; equivalently, in order to have $||\nabla f(x)||^2 \leq \delta^2$, ACProp requires at most $O(\delta^{-4})$ steps.

**Theorem 4.2** (Oracle complexity [23]). *For a stochastic non-convex problem satisfying assumptions in Theorem. 4.1, using only up to first-order gradient information, in the worst case any algorithm requires at least $O(\delta^{-4})$ queries to find a $\delta$-stationary point $x$ such that $||\nabla f(x)||^2 \leq \delta^2$.*

**Optimal rate in big O** Thm. 4.1 and Thm. 4.2 imply that async-optimizers achieves a convergence rate of $O(1/\sqrt{T})$ for the stochastic non-convex problem, which matches the oracle complexity and outperforms the $O(logT/\sqrt{T})$ rate of sync-optimizers (Adam [14], RMSProp[25], AdaBelief [18]). Adam and RMSProp are shown to achieve $O(1/\sqrt{T})$ rate under the stricter condition that $\beta_{2,t} \to 1$ [27]. A similar rate has been achieved in AVAGrad [28], and AdaGrad is shown to achieve a similar rate [29]. Despite the same convergence rate, we show that ACProp has better empirical performance.

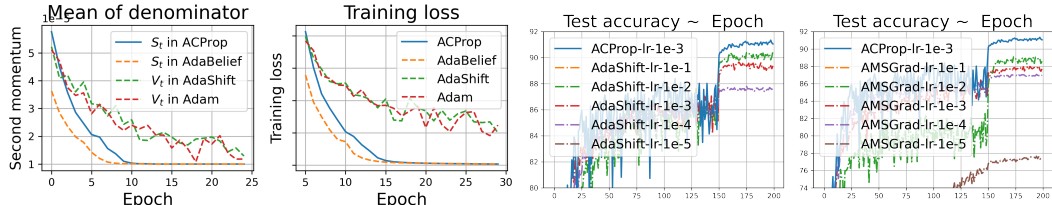

Figure 6: From left to right: (a) Mean value of denominator for a 2-layer MLP on MNIST dataset. (b) Training loss of different optimizers for the 2-layer MLP model. (c) Performance of AdaShift for VGG-11 on CIFAR10 varying with learning rate ranging from 1e-1 to 1e-5, we plot the performance of ACProp with learning rate 1e-3 as reference. Missing lines are because their accuracy are below display threshold. All methods decay learning rate by a factor of 10 at 150th epoch. (d) Performance of AMSGrad for VGG-11 on CIFAR10 varying with learning rate under the same setting in (c).

**Constants in the upper bound of regret** Though both async-center and async-uncenter optimizers have the same convergence rate with matching upper and lower bound in big O notion, the constants of the upper bound on regret is different. Thm. 4.1 implies that the upper bound on regret is an increasing function of $1/C_l$ and $C_u$, and
$$1/C_l = \sqrt{K_u} + \epsilon, \;\; C_u = 1/(\sqrt{K_l} + \epsilon)$$
where $K_l$ and $K_u$ are the lower and upper bound of second momentum, respectively.

We analyze the constants in regret by analyzing $K_l$ and $K_u$. If we assume the observed gradient $g_t$ follows some independent stationary distribution, with mean $\mu$ and variance $\sigma^2$, then approximately

$$\textit{Uncentered second momentum: } 1/C_l^v = \sqrt{K_u^v} + \epsilon \approx \sqrt{\mu^2 + \sigma^2} + \epsilon \qquad (6)$$

$$\textit{Centered second momentum: } 1/C_l^s = \sqrt{K_u^s} + \epsilon \approx \sqrt{\sigma^2} + \epsilon \qquad (7)$$

During early phase of training, in general $|\mu| \gg \sigma$, hence $1/C_l^s \ll 1/C_l^v$, and the centered version (ACProp) can converge faster than uncentered type (AdaShift) by a constant factor of around $\frac{\sqrt{\mu^2+\sigma^2}+\epsilon}{\sqrt{\sigma^2}+\epsilon}$. During the late phase, $g_t$ is centered around 0, and $|\mu| \ll \sigma$, hence $K_l^v$ (for uncentered version) and $K_l^s$ (for centered version) are both close to 0, hence $C_u$ term is close for both types.

**Remark** We emphasize that ACProp rarely encounters numerical issues caused by a small $s_t$ as denominator, even though Eq. (7) implies a lower bound for $s_t$ around $\sigma^2$ which could be small in extreme cases. Note that $s_t$ is an estimate of mixture of two aspects: the change in true gradient $||\nabla f_t(x) - \nabla f_{t-1}(x)||^2$, and the noise in $g_t$ as an observation of $\nabla f(x)$. Therefore, two conditions are essential to achieve $s_t = 0$: the true gradient $\nabla f_t(x)$ remains constant, and $g_t$ is a noise-free observation of $\nabla f_t(x)$. Eq. (7) is based on assumption that $||\nabla f_t(x) - \nabla f_{t-1}(x)||^2 = 0$, if we further assume $\sigma = 0$, then the problem reduces to a trivial ideal case: a linear loss surface with clean observations of gradient, which is rarely satisfied in practice. More discussions are in appendix.

**Empirical validations** We conducted experiments on the MNIST dataset using a 2-layer MLP. We plot the average value of $v_t$ for uncentered-type and $s_t$ for centered-type optimizers; as Fig. 6(a,b) shows, we observe $s_t \leq v_t$ and the centered-type (ACProp, AdaBelief) converges faster, validating our analysis for early phases. For epochs $> 10$, we observe that $\min s_t \approx \min v_t$, validating our analysis for late phases.

As in Fig. 6(a,b), the ratio $v_t/s_t$ decays with training, and in fact it depends on model structure and dataset noise. Therefore, empirically it's hard to compensate for the constants in regret by applying a larger learning rate for async-uncenter optimizers. As shown in Fig. 6(c,d), for VGG network on CIFAR10 classification task, we tried different initial learning rates for AdaShift (async-uncenter) and AMSGrad ranging from 1e-1 to 1e-5, and their performances are all inferior to ACProp with a learning rate 1e-3. Please see Fig.8 for a complete table varying with hyper-parameters.

## 5  Experiments

We validate the performance of ACProp in various experiments, including image classification with convolutional neural networks (CNN), reinforcement learning with deep Q-network (DQN), machine translation with transformer and generative adversarial networks (GANs). We aim to test

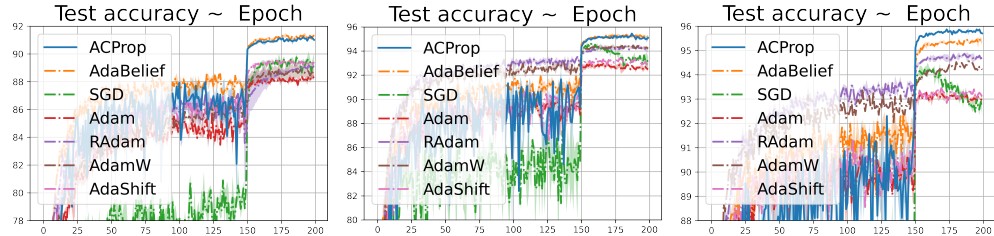

Figure 7: Test accuracy ($mean \pm std$) on CIFAR10 datset. Left to right: VGG-11, ResNet-34, DenseNet-121.

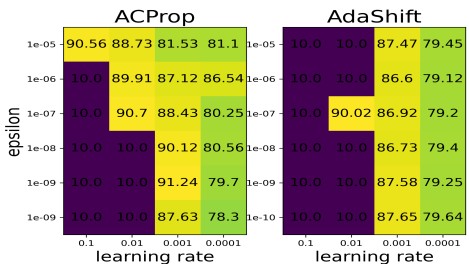

Figure 8: Test accuracy (%) of VGG network on CIFAR10 under different hyper-parameters. We tested learning rate in $\{10^{-1}, 10^{-2}, 10^{-3}, 10^{-4}\}$ and $\epsilon \in \{10^{-5}, ..., 10^{-9}\}$.

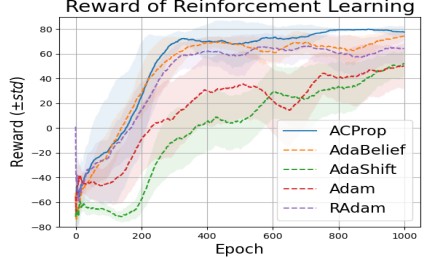

Figure 9: The reward (higher is better) curve of a DQN-network on the four-rooms problem. We report the mean and standard deviation across 10 independent runs.

Table 2: Top-1 accuracy of ResNet18 on ImageNet. $\diamond$ is reported in PyTorch Documentation, $\dagger$ is reported in [30], $*$ is reported in [17], $\ddagger$ is reported in [18]

| SGD | Adam | AdamW | RAdam | AdaShift | AdaBelief | ACProp |
|-----|------|-------|-------|----------|-----------|--------|
| 69.76$^\diamond$ (70.23$^\dagger$) | 66.54$^*$ | 67.93$^\dagger$ | 67.62$^*$ | 65.28 | 70.08$\ddagger$ | **70.46** |

both the generalization performance and training stability: SGD family optimizers typically are the default for CNN models such as in image recognition [6] and object detection [7] due to their better generalization performance than Adam; and Adam is typically the default for GANs [19], reinforcement learning [21] and transformers [20], mainly due to its better numerical stability and faster convergence than SGD. We aim to validate that ACProp can perform well for both cases.

**Image classification with CNN** We first conducted experiments on CIFAR10 image classification task with a VGG-11 [31], ResNet34 [6] and DenseNet-121 [32]. We performed extensive hyper-parameter tuning in order to better compare the performance of different optimizers: for SGD we set the momentum as 0.9 which is the default for many cases [6, 32], and search the learning rate between 0.1 and $10^{-5}$ in the log-grid; for other adaptive optimizers, including AdaBelief, Adam, RAdam, AdamW and AdaShift, we search the learning rate between 0.01 and $10^{-5}$ in the log-grid, and search $\epsilon$ between $10^{-5}$ and $10^{-10}$ in the log-grid. We use a weight decay of 5e-2 for AdamW, and use 5e-4 for other optimizers. We report the $mean \pm std$ for the best of each optimizer in Fig. 7: for VGG and ResNet, ACProp achieves comparable results with AdaBelief and outperforms other optimizers; for DenseNet, ACProp achieves the highest accuracy and even outperforms AdaBelief by 0.5%. As in Table 2, for ResNet18 on ImageNet, ACProp outperforms other methods and achieves comparable accuracy to the best of SGD in the literature, validating its generalization performance.

To evaluate the robustness to hyper-parameters, we test the performance of various optimizers under different hyper-parameters with VGG network. We plot the results for ACProp and AdaShift as an example in Fig. 8 and find that ACProp is more robust to hyper-parameters and typically achieves higher accuracy than AdaShift.

**Reinforcement learning with DQN** We evaluated different optimizers on reinforcement learning with a deep Q-network (DQN) [21] on the four-rooms task [33]. We tune the hyper-parameters in the same setting as previous section. We report the mean and standard deviation of reward (higher is better) across 10 runs in Fig. 9. ACProp achieves the highest mean reward, validating its numerical stability and good generalization.

**Neural machine translation with Transformer** We evaluated the performance of ACProp on neural machine translation tasks with a transformer model [20]. For all optimizers, we set

Table 3: BLEU score (higher is better) on machine translation with Transformer

|  | Adam | RAdam | AdaShift | AdaBelief | ACProp |
|---|---|---|---|---|---|
| DE-EN | 34.66±0.014 | 34.76±0.003 | 30.18±0.020 | 35.17±0.015 | **35.35±0.012** |
| EN-VI | 21.83±0.015 | 22.54±0.005 | 20.18±0.231 | 22.45±0.003 | **22.62±0.008** |
| JA-EN | 33.33±0.008 | 32.23±0.015 | 25.24±0.151 | **34.38±0.009** | 33.70±0.021 |
| RO-EN | 29.78± 0.003 | 30.26 ± 0.011 | 27.86±0.024 | 30.03±0.012 | **30.27±0.007** |

Table 4: FID (lower is better) for GANs

|  | Adam | RAdam | AdaShift | AdaBelief | ACProp |
|---|---|---|---|---|---|
| DCGAN | 49.29±0.25 | 48.24±1.38 | 99.32±3.82 | 47.25±0.79 | **43.43±4.38** |
| RLGAN | 38.18±0.01 | 40.61±0.01 | 56.18 ±0.23 | **36.58±0.12** | 37.15±0.13 |
| SNGAN | 13.14±0.10 | 13.00±0.04 | 26.62±0.21 | 12.70±0.17 | **12.44±0.02** |
| SAGAN | 13.98±0.02 | 14.25±0.01 | 22.11±0.25 | 14.17±0.14 | **13.54±0.15** |

Table 5: Performance comparison between AVAGrad and ACProp. $\uparrow$ ($\downarrow$) represents metrics that upper (lower) is better. $\star$ are reported in the AVAGrad paper [28]

|  | WideResNet Test Error ($\downarrow$) | | Transformer BLEU ($\uparrow$) | | GAN FID ($\downarrow$) | |
|---|---|---|---|---|---|---|
|  | CIFAR10 | CIFAR100 | DE-EN | RO-EN | DCGAN | SNGAN |
| AVAGrad | 3.80$^\star$±0.02 | 18.76$^\star$±0.20 | 30.23±0.024 | 27.73±0.134 | 59.32±3.28 | 21.02±0.14 |
| ACProp | **3.67±0.04** | **18.72±0.01** | **35.35±0.012** | **30.27±0.007** | **43.34±4.38** | **12.44±0.02** |

learning rate as 0.0002, and search for $\beta_1 \in \{0.9, 0.99, 0.999\}$, $\beta_2 \in \{0.98, 0.99, 0.999\}$ and $\epsilon \in \{10^{-5}, 10^{-6}, ...10^{-16}\}$. As shown in Table. 3, ACProp achieves the highest BLEU score in 3 out 4 tasks, and consistently outperforms a well-tuned Adam.

**Generative Adversarial Networks (GAN)** The training of GANs easily suffers from mode collapse and numerical instability [34], hence is a good test for the stability of optimizers. We conducted experiments with Deep Convolutional GAN (DCGAN) [35], Spectral-Norm GAN (SNGAN) [36], Self-Attention GAN (SAGAN) [37] and Relativistic-GAN (RLGAN) [38]. We set $\beta_1 = 0.5$, and search for $\beta_2$ and $\epsilon$ with the same schedule as previous section. We report the FID [39] on CIFAR10 dataset in Table. 4, where a lower FID represents better quality of generated images. ACProp achieves the best overall FID score and outperforms well-tuned Adam.

**Remark** Besides AdaShift, we found another async-optimizer named AVAGrad in [28]. Unlike other adaptive optimizers, AVAGrad is not scale-invariant hence the default hyper-parameters are very different from Adam-type ($lr = 0.1, \epsilon = 0.1$). We searched for hyper-parameters for AVAGrad for a much larger range, with $\epsilon$ between 1e-8 and 100 in the log-grid, and $lr$ between 1e-6 and 100 in the log-grid. For experiments with a WideResNet, we replace the optimizer in the official implementation for AVAGrad by ACProp, and cite results in the AVAGrad paper. As in Table 5, ACProp consistently outperforms AVAGrad in CNN, Transformer, and GAN training.

## 6 Related Works

Besides the aforementioned, other variants of Adam include NosAdam [40], Sadam [41], Adax [42]), AdaBound [15] and Yogi [43]. ACProp could be combined with other techniques such as SWATS [44], LookAhead [45] and norm regularization similar to AdamP [46]. Regarding the theoretical analysis, recent research has provided more fine-grained frameworks [47, 48]. Besides first-order methods, recent research approximate second-order methods in deep learning [49, 50, 51].

## 7 Conclusion

We propose ACProp, a novel first-order gradient optimizer which combines the asynchronous update and centering of second momentum. We demonstrate that ACProp has good theoretical properties: ACProp has a "always-convergence" property for the counter example by Reddi et al. (2018), while sync-optimizers (Adam, RMSProp) could diverge with uncarefully chosen hyper-parameter; for problems with sparse gradient, async-centering (ACProp) has a weaker convergence condition than async-uncentering (AdaShift); ACProp achieves the optimal convergence rate $O(1/\sqrt{T})$, outperforming the $O(logT/\sqrt{T})$ rate of RMSProp (Adam), and achieves a tighter upper bound on risk than AdaShift. In experiments, we validate that ACProp has good empirical performance: it achieves good generalization like SGD, fast convergence and training stability like Adam, and often outperforms Adam and AdaBelief.

## Acknowledgments and Disclosure of Funding

This research is supported by NIH grant R01NS035193.

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
