# Supplementary for: Momentum Centering and Asynchronous Update for Adaptive Gradient Methods

# Contents

# 1 Analysis on convergence conditions

## 1.1 Convergence analysis for Problem 1 in the main paper

**Lemma 1.1.** *There exists an online convex optimization problem where Adam (and RMSprop) has non-zero average regret, and one of the problem is in the form*

$$f_t(x) = \begin{cases} Px, & \text{if } t \bmod P = 1 \\ -x, & \text{Otherwise} \end{cases} \quad x \in [-1, 1], \exists P \in \mathbb{N}, P \geq 3 \tag{1}$$

*Proof.* See [1] Thm.1 for proof. $\qquad\square$

**Lemma 1.2.** *For the problem defined above, there's a threshold of $\beta_2$ above which RMSprop converge.*

*Proof.* See [2] for details. $\qquad\square$

**Lemma 1.3** (Lemma.3.3 in the main paper). *For the problem defined by Eq. (1), ACProp algorithm converges $\forall \beta_1, \beta_2 \in (0, 1), \forall P \in \mathbb{N}, P \geq 3$.*

*Proof.* We analyze the limit behavior of ACProp algorithm. Since the observed gradient is periodic with an integer period $P$, we analyze one period from with indices from $kP$ to $kP + P$, where $k$ is an integer going to $+\infty$.
From the update of ACProp, we observe that:

$$m_{kP} = (1 - \beta_1) \sum_{i=1}^{kP} \beta_1^{kP-i} \times (-1) + (1 - \beta_1) \sum_{j=0}^{k-1} \beta_1^{kP-(jP+1)}(P+1) \tag{2}$$

$$\left( \text{For each observation with gradient } P, \text{we break it into } P = -1 + (P+1) \right)$$

$$= -(1 - \beta_1) \sum_{i=1}^{kP} \beta_1^{kP-i} + (1 - \beta_1)(P+1)\beta_1^{-1} \sum_{j=0}^{k-1} \beta_1^{P(k-j)} \tag{3}$$

$$= -(1 - \beta_1^{kP}) + (1 - \beta_1)(P+1)\beta_1^{P-1} \frac{1 - \beta_1^{(k-1)P}}{1 - \beta_1^P} \tag{4}$$

$$\lim_{k \to \infty} m_{kP} = -1 + (P+1)(1 - \beta_1)\beta_1^{P-1} \frac{1}{1 - \beta_1^P} = \frac{(P+1)\beta_1^{P-1} - P\beta_1^P - 1}{1 - \beta_1^P} \tag{5}$$

$$\left( \text{Since } \beta_1 \in [0, 1) \right)$$

Next, we derive $\lim_{k \to \infty} S_{kP}$. Note that the observed gradient is periodic, and $\lim_{k \to \infty} m_{kP} = \lim_{k \to \infty} m_{kP+P}$, hence $\lim_{k \to \infty} S_{kP} = \lim_{k \to \infty} S_{kP+P}$. Start from index $kP$, we derive variables up to $kP + P$ with ACProp algorithm.

$$index = kP,$$
$$m_{kP}, S_{kP} \tag{6}$$

$index = kP + 1,$

$$m_{kP+1} = \beta_1 m_0 + (1 - \beta_1)P \tag{7}$$

$$S_{kP+1} = \beta_2 S_{kP} + (1 - \beta_2)(P - m_{kP})^2 \tag{8}$$

$index = kP + 2,$

$$m_{kP+2} = \beta_1 m_{kP+1} + (1 - \beta_1) \times (-1) \tag{9}$$

$$= \beta_1^2 m_{kP} + (1 - \beta_1)\beta_1 P + (1 - \beta_1) \times (-1) \tag{10}$$

$$S_{kP+2} = \beta_2 S_{kP+1} + (1 - \beta_2)(-1 - m_{kP+1})^2 \tag{11}$$

$$= \beta_2^2 S_{kP} + (1 - \beta_2)\beta_2(P - m_{kP})^2 + (1 - \beta_2)\big[\beta_1(P - m_{kP}) - (P + 1)\big]^2 \tag{12}$$

$index = kP + 3,$

$$m_{kP+3} = \beta_1 m_{kP+2} + (1 - \beta_1) \times (-1) \tag{13}$$

$$= \beta_1^3 m_{kP} + (1 - \beta_1)\beta_1^2 P + (1 - \beta_1)\beta_1 \times (-1) + (1 - \beta_1) \times (-1) \tag{14}$$

$$S_{kP+3} = \beta_2 S_2 + (1 - \beta_2)(-1 - m_{kP+2})^2 \tag{15}$$

$$= \beta_2^3 S_{kP} + (1 - \beta_2)\beta_2^2(P - m_{kP})^2$$
$$+ (1 - \beta_2)\beta_2\big[\beta_1(P - m_{kP}) - (P + 1)\big]^2(\beta_2 + \beta_1^2) \tag{16}$$

$index = kP + 4,$

$$m_{kP+4} = \beta_1^4 m_{kP} + (1 - \beta_1)\beta_1^3 P + (-1)(1 - \beta_1)(\beta_1^2 + \beta_1 + 1) \tag{17}$$

$$S_{kP+4} = \beta_2 S_{kP+3} + (1 - \beta_2)(-1 - m_{kP+3})^2 \tag{18}$$

$$= \beta_2^4 S_{kP} + (1 - \beta_2)\beta_2^3(P - m_{kP})^2$$
$$+ (1 - \beta_2)\beta_2\big[\beta_1(P - m_{kP}) - (P + 1)\big]^2(\beta_2^2 + \beta_2\beta_1^2 + \beta_1^4) \tag{19}$$

$$\cdots$$

$index = kP + P,$

$$m_{kP+P} = \beta_1^P m_{kP} + (1 - \beta_1)\beta_1^{P-1} P + (-1)(1 - \beta_1)\big[\beta_1^{P-2} + \beta_1^{P-3} + \ldots + 1\big] \tag{20}$$

$$= \beta_1^P m_{kP} + (1 - \beta_1)\beta_1^{P-1} P + (\beta_1 - 1)\frac{1 - \beta_1^{P-1}}{1 - \beta 1} \tag{21}$$

$$S_{kP+P} = \beta_2^P S_{kP} + (1 - \beta_2)\beta_2^{P-1}(P - m_{kP})^2$$
$$+ (1 - \beta_2)\big[\beta_1(P - m_{kP}) - (P + 1)\big]^2\big(\beta_2^{P-2} + \beta_2^{P-3}\beta_1^2 + \ldots + \beta_2^0\beta_1^{2P-4}\big) \tag{22}$$

$$= \beta_2^P S_{kP} + (1 - \beta_2)\beta_2^{P-1}(P - m_{kP})^2$$
$$+ (1 - \beta_2)\big[\beta_1(P - m_{kP}) - (P + 1)\big]^2\beta_2^{P-2}\frac{1 - (\beta_1^2/\beta_2)^{P-1}}{1 - (\beta_1^2/\beta_2)} \tag{23}$$

As $k$ goes to $+\infty$, we have

$$\lim_{k\to\infty} m_{kP+P} = \lim_{k\to\infty} m_{kP} \tag{24}$$

$$\lim_{k\to\infty} S_{kP+P} = \lim_{k\to\infty} S_{kP} \tag{25}$$

From Eq. (21) we have:

$$m_{kP+P} = \frac{(P+1)\beta_1^{P-1} - P\beta_1^P - 1}{1 - \beta_1^P} \tag{26}$$

which matches our result in Eq. (6). Similarly, from Eq. (23), take limit of $k \to \infty$, and combine with Eq. (25), we have

$$\lim_{k \to \infty} S_{kP} = \frac{1 - \beta_2}{1 - \beta_2^P} \left[ \beta_2^{P-1}(P - \lim_{k \to \infty} m_{kP})^2 + \left[ \beta_1(P - \lim_{k \to \infty} m_{kP}) - (P+1) \right]^2 \beta_2^{P-2} \frac{1 - (\beta_1^2/\beta_2)^{P-1}}{1 - (\beta_1^2/\beta_2)} \right] \tag{27}$$

Since we have the exact expression for the limit, it's trivial to check that

$$S_i \geq S_{kP}, \quad \forall i \in [kP+1, kP+P], i \in \mathbb{N}, k \to \infty \tag{28}$$

Intuitively, suppose for some time period, we only observe a constant gradient -1 without observing the outlier gradient $(P)$; the longer the length of this period, the smaller is the corresponding $S$ value, because $S$ records the difference between observations. Note that since last time that outlier gradient $(P)$ is observed (at index $kP+1-P$), index $kP$ has the longest distance from index $kP+1-P$ without observing the outlier gradient $(P)$. Therefore, $S_{kP}$ has the smallest value within a period of $P$ as $k$ goes to infinity.

For step $kP+1$ to $kP+P$, the update on parameter is:

$$index = kP+1, -\Delta_x^{kP+1} = \frac{\alpha_0}{\sqrt{kP+1}} \frac{P}{\sqrt{S_{kP}} + \epsilon} \tag{29}$$

$$index = kP+2, -\Delta_x^{kP+2} = \frac{\alpha_0}{\sqrt{kP+2}} \frac{-1}{\sqrt{S_{kP+1}} + \epsilon} \tag{30}$$

$$...$$

$$index = kP+P, -\Delta_x^{kP+P} = \frac{\alpha_0}{\sqrt{kP+P}} \frac{-1}{\sqrt{S_{kP+P-1}} + \epsilon} \tag{31}$$

So the negative total update within this period is:

$$\frac{\alpha_0}{\sqrt{kP+1}} \frac{P}{\sqrt{S_{kP}} + \epsilon} - \underbrace{\left[ \frac{\alpha_0}{\sqrt{kP+2}} \frac{1}{\sqrt{S_{kP+1}} + \epsilon} + ... + \frac{\alpha_0}{\sqrt{kP+P}} \frac{1}{\sqrt{S_{kP+P}} + \epsilon} \right]}_{P-1 \ terms} \tag{32}$$

$$\geq \frac{\alpha_0}{\sqrt{kP+1}} \frac{P}{\sqrt{S_{kP}} + \epsilon} - \underbrace{\left[ \frac{\alpha_0}{\sqrt{kP+1}} \frac{1}{\sqrt{S_{kP}} + \epsilon} + ... + \frac{\alpha_0}{\sqrt{kP+1}} \frac{1}{\sqrt{S_{kP}} + \epsilon} \right]}_{P-1 \ terms} \tag{33}$$

$$\left( Since \ S_{kP} \ is \ the \ minimum \ within \ the \ period \right)$$

$$= \frac{\alpha_0}{\sqrt{S_{kP}} + \epsilon} \frac{1}{\sqrt{kP+1}} \tag{34}$$

where $\alpha_0$ is the initial learning rate. Note that the above result hold for every period of length $P$ as $k$ gets larger. Therefore, for some $K$ such that for every $k > K$, $m_{kP}$ and $S_{kP}$ are close enough to their limits, the total update after $K$ is:

$$\sum_{k=K}^{\infty} \frac{\alpha_0}{\sqrt{S_{kP}} + \epsilon} \frac{1}{\sqrt{kP+1}} \approx \frac{\alpha_0}{\sqrt{\lim_{k\to\infty} S_{kP}} + \epsilon} \frac{1}{\sqrt{P}} \sum_{k=K}^{\infty} \frac{1}{\sqrt{k}} \quad \textit{If K is sufficiently large} \quad (35)$$

where $\lim_{k\to\infty} S_{kP}$ is a constant determined by Eq. (27). Note that this is the negative update; hence ACProp goes to the negative direction, which is what we expected for this problem. Also considering that $\sum_{k=K}^{\infty} \frac{1}{\sqrt{k}} \to \infty$, hence ACProp can go arbitrarily far in the correct direction if the algorithm runs for infinitely long, therefore the bias caused by first $K$ steps will vanish with running time. Furthermore, since $x$ lies in the bounded region of $[-1, 1]$, if the updated result falls out of this region, it can always be clipped. Therefore, for this problem, ACProp always converge to $x = -1, \forall \beta_1, \beta_2 \in (0, 1)$. When $\beta_2 = 1$, the denominator won't update, and ACProp reduces to SGD (with momentum), and it's shown to converge. $\qquad\square$

**Lemma 1.4.** *For any constant $\beta_1, \beta_2 \in [0, 1)$ such that $\beta_1 < \sqrt{\beta_2}$, there is a stochastic convex optimization problem for which Adam does not converge to the optimal solution. One example of such stochastic problem is:*

$$f_t(x) = \begin{cases} Px & \textit{with probability } \frac{1+\delta}{P+1} \\ -x & \textit{with probability } \frac{P-\delta}{P+1} \end{cases} \quad x \in [-1, 1] \quad (36)$$

*Proof.* See Thm.3 in [1]. $\qquad\square$

**Lemma 1.5.** *For the stochastic problem defined by Eq. (36), ACProp converge to the optimal solution, $\forall \beta_1, \beta_2 \in (0, 1)$.*

*Proof.* The update at step $t$ is:

$$\Delta_x^t = -\frac{\alpha_0}{\sqrt{t}} \frac{g_t}{\sqrt{S_{t-1}} + \epsilon} \quad (37)$$

Take expectation conditioned on observations up to step $t - 1$, we have:

$$\mathbb{E}\Delta_x^t = -\frac{\alpha_0}{\sqrt{t}} \frac{\mathbb{E}_t g_t}{\sqrt{S_{t-1}} + \epsilon} \quad (38)$$

$$= -\frac{\alpha_0}{\sqrt{t}\left(\sqrt{S_{t-1}} + \epsilon\right)} \mathbb{E}_t g_t \quad (39)$$

$$= -\frac{\alpha_0}{\sqrt{t}\left(\sqrt{S_{t-1}} + \epsilon\right)} \left[ P\frac{1+\delta}{P+1} - \frac{P-\delta}{P+1} \right] \quad (40)$$

$$= -\frac{\alpha_0 \delta}{\sqrt{t}\left(\sqrt{S_{t-1}} + \epsilon\right)} \quad (41)$$

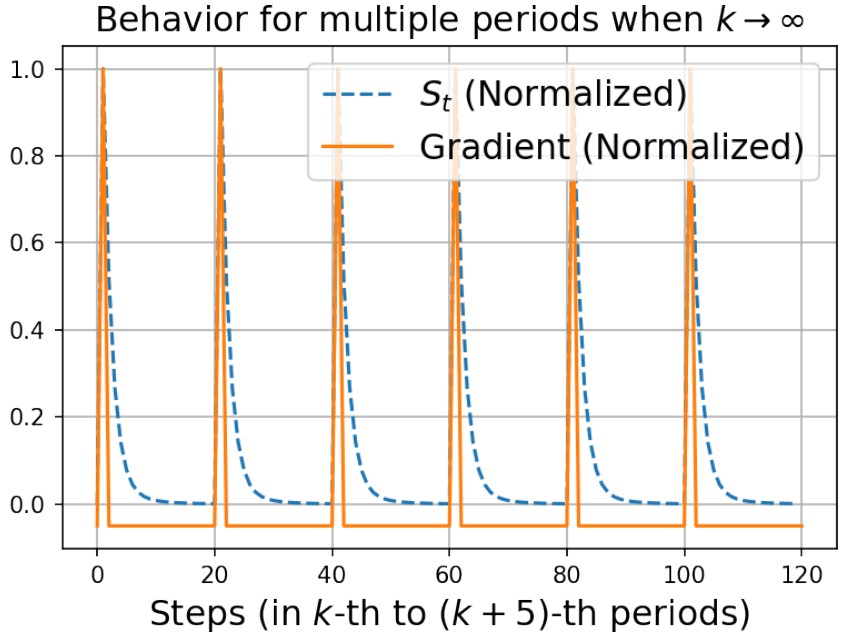

Figure 1: Behavior of $S_t$ and $g_t$ in ACProp of multiple periods for problem (1). Note that as $k \to \infty$, the behavior of ACProp is periodic.

$$\leq -\frac{\alpha_0 \delta}{\sqrt{t}\left(P + 1 + \epsilon\right)} \tag{42}$$

where the last inequality is due to $S_t \leq (P+1)^2$, because $S_t$ is a smoothed version of squared difference between gradients, and the maximum difference in gradient is $P + 1$. Therefore, for every step, ACProp is expected to move in the negative direction, also considering that $\sum_{t=1}^{\infty} \frac{1}{\sqrt{t}} \to \infty$, and whenever $x < -1$ we can always clip it to -1, hence ACProp will drift $x$ to -1, which is the optimal value. $\square$

### 1.1.1 Numerical validations

We validate our analysis above in numerical experiments, and plot the curve of $S_t$ and $g_t$ for multiple periods (as $k \to \infty$) in Fig. 1 and zoom in to a single period in Fig. 2. Note that the largest gradient $P$ (normalized as 1) appears at step $kP + 1$, and $S$ takes it minimal at step $kP$ (e.g. $S_{kP}$ is the smallest number within a period). Note the update for step $kP + 1$ is $g_{kP+1}/\sqrt{S_{kP}}$, it's the largest gradient divided the smallest denominator, hence the net update within a period pushes $x$ towards the optimal point.

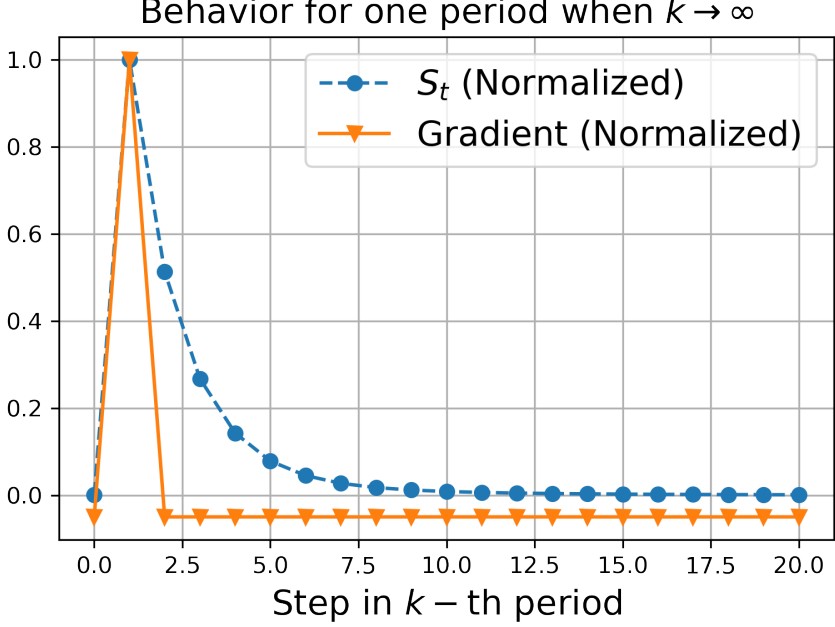

Figure 2: Behavior of $S_t$ and $g_t$ in ACProp of one period for problem (1).

## 1.2 Convergence analysis for Problem 2 in the main paper

**Lemma 1.6** (Lemma 3.4 in the main paper). *For the problem defined by Eq. (43), consider the hyper-parameter tuple $(\beta_1, \beta_2, P)$, there exists cases where ACProp converges but AdaShift with $n = 1$ diverges, but not vice versa.*

$$f_t(x) = \begin{cases} P/2 \times x, & t\%P == 1 \\ -x, & t\%P == P - 2 \quad P > 3, P \in \mathbb{N}, x \in [0, 1]. \\ 0, & otherwise \end{cases} \tag{43}$$

*Proof.* The proof is similar to Lemma. 1.3,we derive the limit behavior of different methods.

$$index = kP,$$

$$m_{kP}, v_{kP}, s_{kP}$$

$$index = kP + 1,$$

$$m_{kP+1} = m_{kP}\beta_1 + (1 - \beta_1)P/2 \tag{44}$$

$$v_{kP+1} = v_{kP}\beta_2 + (1 - \beta_2)P^2/4 \tag{45}$$

$$s_{kP+1} = s_{kP}\beta_2 + (1 - \beta_2)(P/2 - m_{kP})^2 \tag{46}$$

$$...$$

$$index = kP + P - 2,$$

$$m_{kP+P-2} = m_{kP}\beta_1^{P-2} + (1 - \beta_1)\frac{P}{2}\beta_1^{P-3} + (1 - \beta_1) \times (-1) \tag{47}$$

$$v_{kP+P-2} = v_{kP}\beta_2^{P-2} + (1 - \beta_2)\frac{P^2}{4}\beta_2^{P-3} + (1 - \beta_2) \tag{48}$$

$$s_{kP+P-2} = s_{kP}\beta_2^{P-2} + (1-\beta_2)\beta_2^{P-3}(\frac{P}{2} - m_{kP})^2 + (1-\beta_2)\beta_2^{P-4}m_{kP+1}^2 + \dots$$
$$+ (1-\beta_2)\beta_2 m_{kP+P-4}^2 + (1-\beta_2)(m_{kP+P-3}+1)^2 \tag{49}$$

$index = kP + P - 1,$

$$m_{kP+P-1} = m_{kP+P-1}\beta_1 \tag{50}$$

$$v_{kP+P-1} = v_{kP+P-2}\beta_2 \tag{51}$$

$$s_{kP+P-1} = s_{kP}\beta_2^{P-1} + (1-\beta_2)\beta_2^{P-1}(\frac{P}{2} - m_{kP})^2 + (1-\beta_2)\beta_2^{P-3}m_{kP+1}^2 + \dots$$
$$+ (1-\beta_2)\beta_2^2 m_{kP+P-4}^2 + (1-\beta_2)\beta_2(m_{kP+P-3}+1)^2 + (1-\beta_2)m_{kP+P-2}^2 \tag{52}$$

$index = kP + P,$

$$m_{kP+P} = m_{kP}\beta_1^P + (1-\beta_1)\frac{P}{2}\beta_1^{P-1} + (1-\beta_1)(-1)\beta_1^2 \tag{53}$$

$$v_{kP+P} = v_{kP}\beta_2^P + (1-\beta_2)\frac{P^2}{4}\beta_2^{P-1} + (1-\beta_2)\beta_2^2 \tag{54}$$

$$s_{kP+p} = s_{kP}\beta_2^P + (1-\beta_2)\beta_2^{P-1}(\frac{P}{2} - m_{kP})^2 + (1-\beta_2)\beta_2^{P-2}m_{kP+1}^2 + \dots$$
$$+ (1-\beta_2)\beta_2^3 m_{kP+P-4}^2 + (1-\beta_2)\beta_2^2(m_{kP+P-3}+1)^2$$
$$+ (1-\beta_2)m_{kP+P-2}^2\beta_2 + (1-\beta_2)m_{kP+P-1}^2 \tag{55}$$

Next, we derive the exact expression using the fact that the problem is periodic, hence $\lim_{k\to\infty} m_{kP} = \lim_{k\to\infty} m_{kP+P}, \lim_{k\to\infty} s_{kP} = \lim_{k\to\infty} s_{kP+P}, \lim_{k\to\infty} v_{kP} = \lim_{k\to\infty} v_{kP+P},$ hence we have:

$$\lim_{k\to\infty} m_{kP} = \lim_{k\to\infty} m_{kP}\beta_1^P + (1-\beta_1)\frac{P}{2}\beta_1^{P-1} + (1-\beta_1)(-1)\beta_1^2 \tag{56}$$

$$\lim_{k\to\infty} m_{kP} = \frac{1-\beta_1}{1-\beta_1^P}\left[\frac{P}{2}\beta_1^{P-1} - \beta_1^2\right] \tag{57}$$

$$\lim_{k\to\infty} m_{kP-1} = \frac{1}{\beta_1}\lim_{k\to\infty} m_{kP} \tag{58}$$

$$\lim_{k\to\infty} m_{kP-2} = \frac{1}{\beta_1}\left[\lim_{k\to\infty} m_{kP-1} - (1-\beta_1)0\right] \tag{59}$$

$$\lim_{k\to\infty} m_{kP-3} = \frac{1}{\beta_1}\left[\lim_{k\to\infty} m_{kP-2} - (1-\beta_1)(-1)\right] \tag{60}$$

Similarly, we can get

$$\lim_{k\to\infty} v_{kP} = \frac{1-\beta_2}{1-\beta_2^P}\left[\frac{P^2}{4}\beta_2^{P-1} + \beta_2^2\right] \tag{61}$$

$$\lim_{k\to\infty} v_{kP-1} = \frac{1}{\beta_2}\lim_{k\to\infty} v_{kP} \tag{62}$$

$$\lim_{k\to\infty} v_{kP-2} = \frac{1}{\beta_2}\lim_{k\to\infty} v_{kP-1} \tag{63}$$

$$\lim_{k\to\infty} v_{kP-3} = \frac{1}{\beta_2}\left[\lim_{k\to\infty} v_{kP-2} - (1-\beta_2) \times 1^2\right] \tag{64}$$

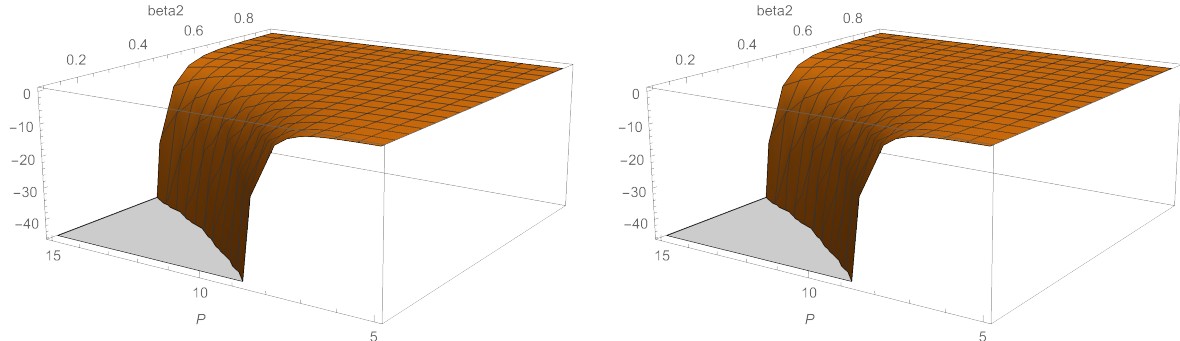

Figure 3: Value of $\frac{s^+}{s^-} - \frac{v^+}{v^-}$ when $\beta_1 = 0.2$     Figure 4: Value of $\frac{s^+}{s^-} - \frac{v^+}{v^-}$ when $\beta_1 = 0.9$

For ACProp, we have the following results:

$$\lim_{k\to\infty} s_{kP} = \lim_{k\to\infty} \frac{1-\beta_2}{1-\beta_2^P}\Big[\beta_2^{P-4}(\frac{P}{2} - m_{kP})^2 + \beta_2^3 \frac{\beta_2^{P-5} - \beta_1^{2(P-4)}\beta_2}{1-\beta_1^2\beta_2} + \beta_2^2(m_{kP+P-3}+1)^2$$
$$+ \beta_2 m_{kP+P-2}^2 + m_{kP+P-1}^2\Big] \tag{65}$$

$$\lim_{k\to\infty} s_{kP-1} = \lim_{k\to\infty} \frac{1}{\beta_2}\Big[s_{kP} - (1-\beta_2)m_{kP}^2\Big] \tag{66}$$

$$\lim_{k\to\infty} s_{kP-2} = \lim_{k\to\infty} \frac{1}{\beta_2}\Big[s_{kP-1} - (1-\beta_2)m_{kP-1}^2\Big] \tag{67}$$

$$\lim_{k\to\infty} s_{kP-3} = \lim_{k\to\infty} \frac{1}{\beta_2}\Big[s_{kP-2} - (1-\beta_2)(m_{kP-2}+1)^2\Big] \tag{68}$$

$$\tag{69}$$

Within each period, ACprop will perform a positive update $P/(2\sqrt{s^+})$ and a negative update $-1/\sqrt{s^-}$, where $s^+$ ($s^-$) is the value of denominator before observing positive (negative) gradient. Similar notations for $v^+$ and $v^-$ in AdaShift, where $s^+ = s_{kP}, s^- = s_{kP-3}, v^+ = v_{kP}, v^- = v_{kP-3}$. A net update in the correct direction requires $\frac{P}{2\sqrt{s^+}} > \frac{1}{\sqrt{s^-}}$, (or $s^+/s^- < P^2/4$). Since we have the exact expression for these terms in the limit sense, it's trivial to verify that $s^+/s^- \leq v^+/v^-$ (e.g. the value $\frac{s^+}{s^-} - \frac{v^+}{v^-}$ is negative as in Fig. 3 and 4), hence ACProp is easier to satisfy the convergence condition. $\qquad\square$

## 1.3   Numerical experiments

We conducted more experiments to validate previous claims. We plot the area of convergence for different $\beta_1$ values for problem (1) in Fig. 5 to Fig. 7, and validate the always-convergence property of ACProp with different values of $\beta_1$. We also plot the area of convergence for problem (2) defined by Eq. (43), results are shown in Fig. 8 to Fig. 10. Note that for this problem the always-convergence does not hold, but ACProp has a much larger area of convergence than AdaShift.

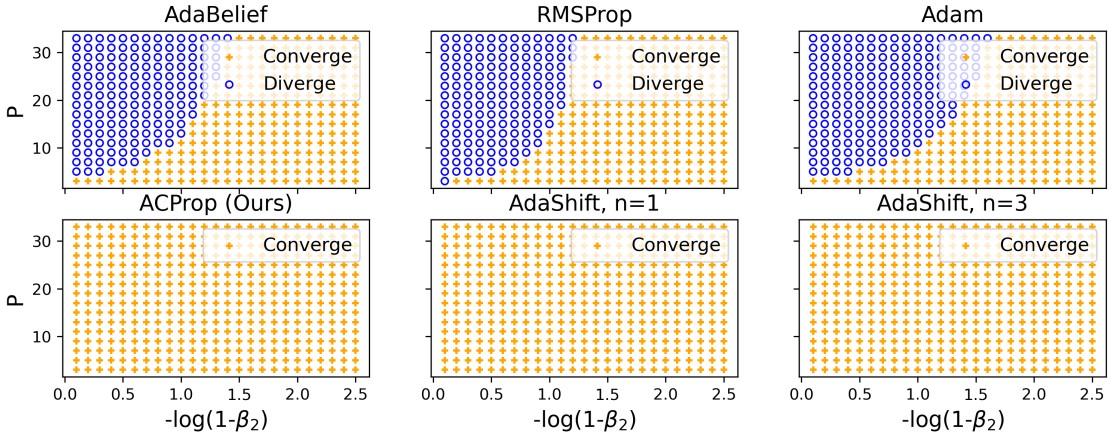

Figure 5: Numerical experiments on problem (1) with $\beta_1 = 0.5$

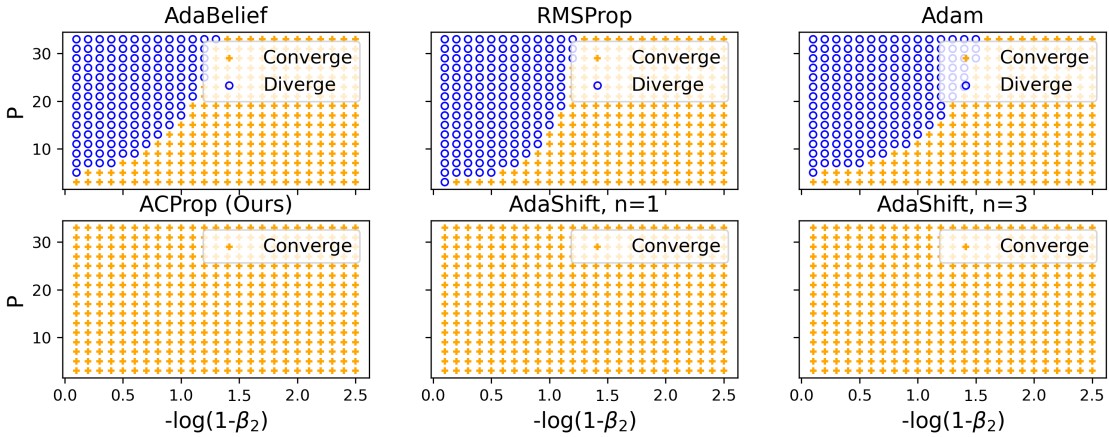

Figure 6: Numerical experiments on problem (1) with $\beta_1 = 0.5$

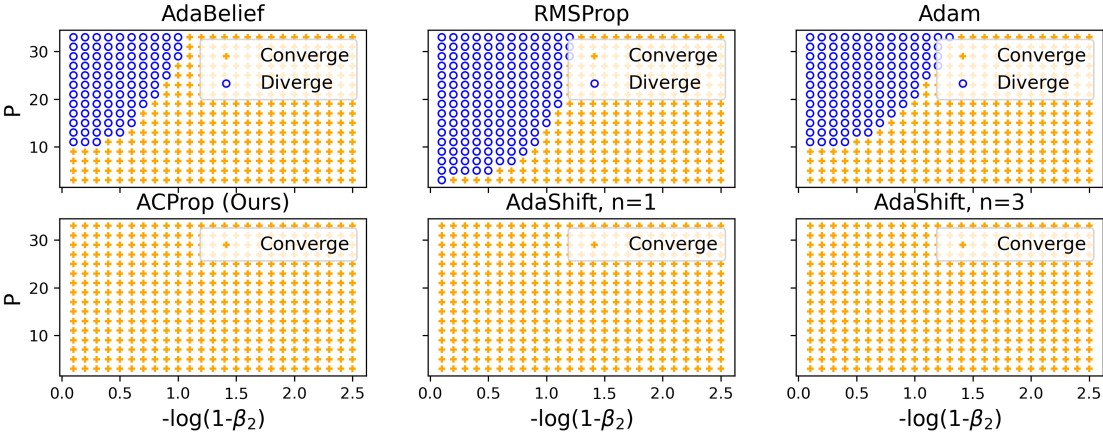

Figure 7: Numerical experiments on problem (1) with $\beta_1 = 0.9$

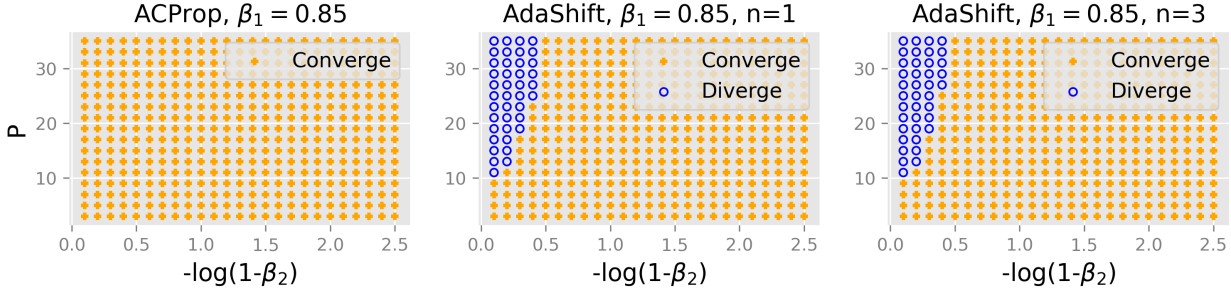

Figure 8: Numerical experiments on problem (43) with $\beta_1 = 0.85$

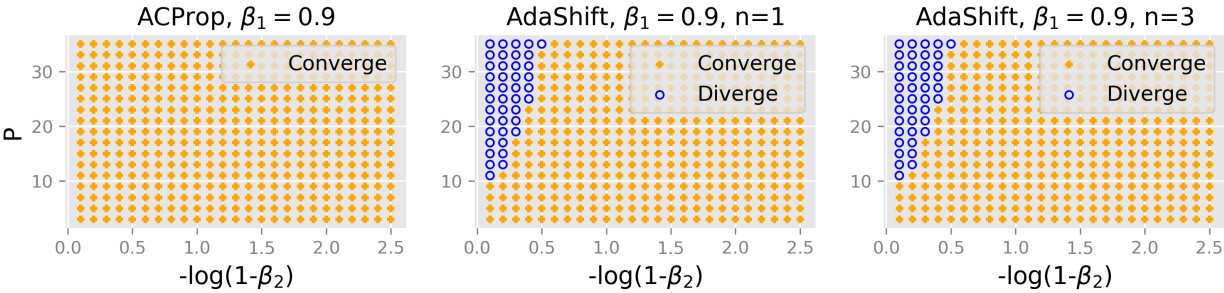

Figure 9: Numerical experiments on problem (43) with $\beta_1 = 0.9$

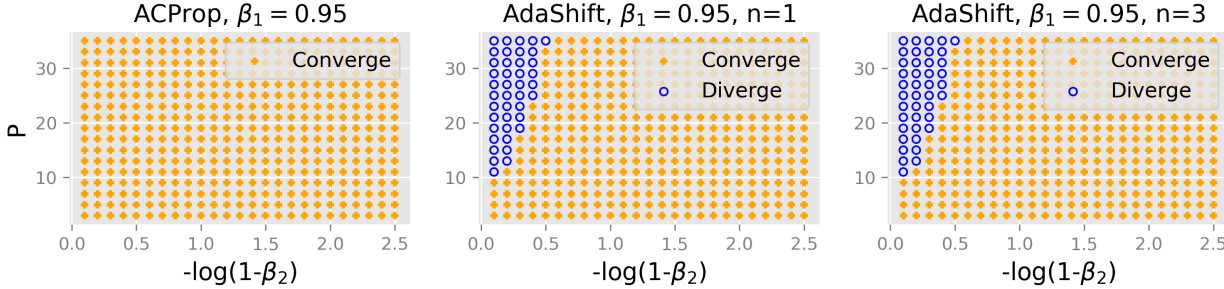

Figure 10: Numerical experiments on problem (43) with $\beta_1 = 0.95$

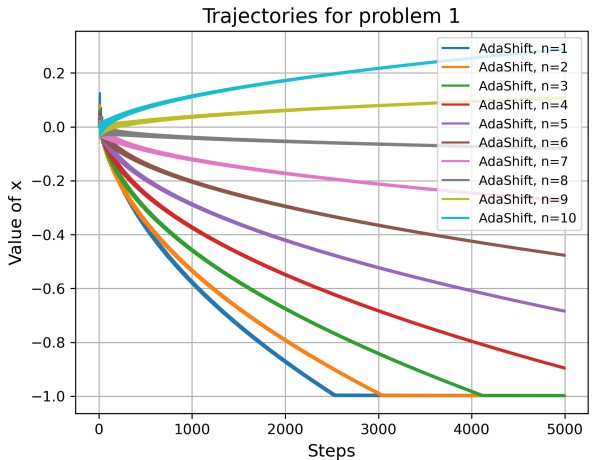

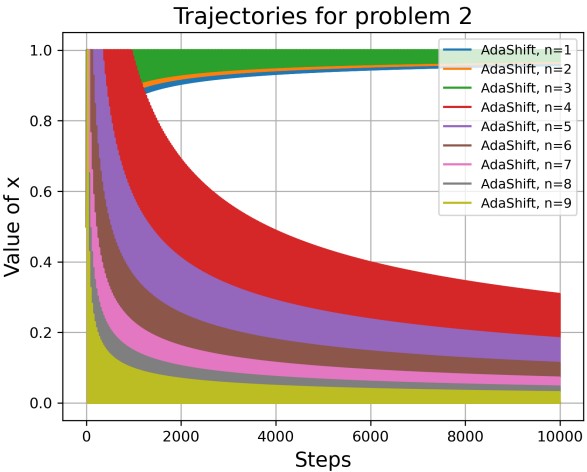

(a) Trajectories of AdaShift with various $n$ for problem (1). Note that optimal is $x^* = -1$. Note that convergence of problem (1) requires a small delay step $n$, but convergence of problem (2) requires a large $n$, hence there's no good criterion to select an optimal $n$.

(b) Trajectories of AdaShift with various $n$ for problem (43). Note that optimal is $x^* = 0.0$, and the trajectories are oscillating at a high frequency hence appears to be spanning an area.

# 2 Convergence Analysis for stochastic non-convex optimization

## 2.1 Problem definition and assumptions

The problem is defined as:
$$\min_{x \in \mathbb{R}^d} f(x) = \mathbb{E}[F(x, \xi)] \tag{70}$$

where $x$ typically represents parameters of the model, and $\xi$ represents data which typically follows some distribution.

We mainly consider the stochastic non-convex case, with assumptions below.

**A.1** $f$ is continuously differentiable, $f$ is lower-bounde by $f^*$. $\nabla f(f)$ is globalluy Lipschitz continuous with constant $L$:

$$||\nabla f(x) - \nabla f(y)|| \leq L||x - y|| \tag{71}$$

**A.2** For any iteration $t$, $g_t$ is an unbiased estimator of $\nabla f(x_t)$ with variance bounded by $\sigma^2$. The norm of $g_t$ is upper-bounded by $M_g$.

$$(a) \quad \mathbb{E}g_t = \nabla f(x_t) \tag{72}$$

$$(b) \quad \mathbb{E}\big[||g_t - \nabla f(x_t)||^2\big] \leq \sigma^2 \tag{73}$$

## 2.2 Convergence analysis of Async-optimizers in stochastic non-convex optimization

**Theorem 2.1** (Thm.4.1 in the main paper). *Under assumptions **A.1-2**, assume $f$ is upper bounded by $M_f$, with learning rate schedule as*

$$\alpha_t = \alpha_0 t^{-\eta}, \quad \alpha_0 \leq \frac{C_l}{LC_u^2}, \quad \eta \in [0.5, 1) \tag{74}$$

*the sequence generated by*

$$x_{t+1} = x_t - \alpha_t A_t g_t \tag{75}$$

*satisfies*

$$\frac{1}{T} \sum_{t=1}^{T} \Big|\Big|\nabla f(x_t)\Big|\Big|^2 \leq \frac{2}{C_l}\Big[(M_f - f^*)\alpha_0 T^{\eta-1} + \frac{LC_u^2 \sigma^2 \alpha_0}{2(1-\eta)}T^{-\eta}\Big] \tag{76}$$

*where $C_l$ and $C_u$ are scalars representing the lower and upper bound for $A_t$, e.g. $C_l I \preceq A_t \preceq C_u I$, where $A \preceq B$ represents $B - A$ is semi-positive-definite.*

*Proof.* Let

$$\delta_t = g_t - \nabla f(x_t) \tag{77}$$

then by **A.2**, $\mathbb{E}\delta_t = 0$.

$$f(x_{t+1}) \leq f(x_t) + \Big\langle \nabla f(x_t), x_{t+1} - x_t \Big\rangle + \frac{L}{2}\Big|\Big|x_{t+1} - x_t\Big|\Big|^2 \tag{78}$$

$$\Big(by\ L\text{-smoothness of } f(x)\Big)$$

$$= f(x_t) - \alpha_t \Big\langle \nabla f(x_t), A_t g_t \Big\rangle + \frac{L}{2}\alpha_t^2 \Big|\Big| A_t g_t \Big|\Big|^2 \tag{79}$$

$$= f(x_t) - \alpha_t \Big\langle \nabla f(x_t), A_t \big(\delta_t + \nabla f(x_t)\big) \Big\rangle + \frac{L}{2}\alpha_t^2 \Big|\Big| A_t g_t \Big|\Big|^2 \tag{80}$$

$$\leq f(x_t) - \alpha_t \Big\langle \nabla f(x_t), A_t \nabla f(x_t) \Big\rangle - \alpha_t \Big\langle \nabla f(x_t), A_t \delta_t \Big\rangle + \frac{L}{2}\alpha_t^2 C_u^2 \Big|\Big| g_t \Big|\Big|^2 \tag{81}$$

Take expectation on both sides of Eq. (81), conditioned on $\xi_{[t-1]} = \{x_1, x_2, ...x_{t-1}\}$, also notice that $A_t$ is a constant given $\xi_{[t-1]}$, we have

$$\mathbb{E}\big[f(x_{t+1})|x_1, ...x_t\big] \leq f(x_t) - \alpha_t \Big\langle \nabla f(x_t), A_t \nabla f(x_t) \Big\rangle + \frac{L}{2}\alpha_t^2 C_u^2 \mathbb{E}\Big|\Big| g_t \Big|\Big|^2 \tag{82}$$

$$\Big(A_t \text{ is independent of } g_t \text{ given } \{x_1, ...x_{t-1}\},\ \text{ and } \mathbb{E}\delta_t = 0\Big)$$

In order to bound RHS of Eq. (82), we first bound $\mathbb{E}\big[||g_t||^2\big]$.

$$\mathbb{E}\Big[\Big|\Big| g_t \Big|\Big|^2 \Big| x_1, ...x_t\Big] = \mathbb{E}\Big[\Big|\Big| \nabla f(x_t) + \delta_t \Big|\Big|^2 \Big| x_1, ...x_t\Big] \tag{83}$$

$$= \mathbb{E}\Big[\Big|\Big| \nabla f(x_t) \Big|\Big|^2 \Big| x_1, ...x_t\Big] + \mathbb{E}\Big[\Big|\Big| \nabla \delta_t \Big|\Big|^2 \Big| x_1, ...x_t\Big] + 2\mathbb{E}\Big[\Big\langle \delta_t, \nabla f(x_t) \Big\rangle \Big| x_1, ...x_t\Big] \tag{84}$$

$$\leq \Big|\Big| \nabla f(x_t) \Big|\Big|^2 + \sigma^2 \tag{85}$$

$$\Big(By\ \textbf{A.2},\ \text{and } \nabla f(x_t) \text{ is a constant given } x_t\Big)$$

Plug Eq. (85) into Eq. (82), we have

$$\mathbb{E}\Big[f(x_{t+1})\Big| x_1, ...x_t\Big] \leq f(x_t) - \alpha_t \Big\langle \nabla f(x_t), A_t \nabla f(x_t) \Big\rangle + \frac{L}{2}C_u^2 \alpha_t^2 \Big[\Big|\Big| \nabla f(x_t) \Big|\Big|^2 + \sigma^2 \Big] \tag{86}$$

$$= f(x_t) - \Big(\alpha_t C_l - \frac{LC_u^2}{2}\alpha_t^2\Big)\Big|\Big| \nabla f(x_t) \Big|\Big|^2 + \frac{LC_u^2 \sigma^2}{2}\alpha_t^2 \tag{87}$$

By **A.5** that $0 < \alpha_t \leq \frac{C_l}{LC_u^2}$, we have

$$\alpha_t C_l - \frac{LC_u^2 \alpha_t^2}{2} = \alpha_t\Big(C_l - \frac{LC_u^2 \alpha_t}{2}\Big) \geq \alpha_t \frac{C_l}{2} \tag{88}$$

Combine Eq. (87) and Eq. (88), we have

$$\frac{\alpha_t C_l}{2}\Big|\Big| \nabla f(x_t) \Big|\Big|^2 \leq \Big(\alpha_t C_l - \frac{LC_u^2 \alpha_t^2}{2}\Big)\Big|\Big| \nabla f(x_t) \Big|\Big|^2 \tag{89}$$

$$\leq f(x_t) - \mathbb{E}\Big[f(x_{t+1})\Big| x_1, ...x_t\Big] + \frac{LC_u^2 \sigma^2}{2}\alpha_t^2 \tag{90}$$

Then we have

$$\frac{C_l}{2}\Big|\Big| \nabla f(x_t) \Big|\Big|^2 \leq \frac{1}{\alpha_t}f(x_t) - \frac{1}{\alpha_t}\mathbb{E}\Big[f(x_{t+1})\Big| x_1, ...x_t\Big] + \frac{LC_u^2 \sigma^2}{2}\alpha_t \tag{91}$$

Perform telescope sum on Eq. (91), and recursively taking conditional expectations on the history of $\{x_i\}_{i=1}^T$, we have

$$\frac{C_l}{2}\sum_{t=1}^T \left\|\nabla f(x_t)\right\|^2 \le \sum_{t=1}^T \frac{1}{\alpha_t}\Big(\mathbb{E}f(x_t) - \mathbb{E}f(x_{t+1})\Big) + \frac{LC_u^2\sigma^2}{2}\sum_{t=1}^T \alpha_t \tag{92}$$

$$= \frac{\mathbb{E}f(x_1)}{\alpha_1} - \frac{\mathbb{E}f(x_{T+1})}{\alpha_T} + \sum_{t=2}^T \left(\frac{1}{\alpha_t} - \frac{1}{\alpha_{t-1}}\right)\mathbb{E}f(x_t) + \frac{LC_u^2\sigma^2}{2}\sum_{t=1}^T \alpha_t \tag{93}$$

$$\le \frac{M_f}{\alpha_1} - \frac{f^*}{\alpha_T} + M_f\sum_{t=1}^T \left(\frac{1}{\alpha_t} - \frac{1}{\alpha_{t-1}}\right) + \frac{LC_u^2\sigma^2}{2}\sum_{t=1}^T \alpha_t \tag{94}$$

$$\le \frac{M_f - f^*}{\alpha_T} + \frac{LC_u^2\sigma^2}{2}\sum_{t=1}^T \alpha_t \tag{95}$$

$$\le (M_f - f^*)\alpha_0 T^\eta + \frac{LC_u^2\sigma^2\alpha_0}{2}\left(\zeta(\eta) + \frac{T^{1-\eta}}{1-\eta} + \frac{1}{2}T^{-\eta}\right) \tag{96}$$

$$\Bigg(\textit{By sum of generalized harmonic series,}$$

$$\sum_{k=1}^n \frac{1}{k^s} \sim \zeta(s) + \frac{n^{1-s}}{1-s} + \frac{1}{2n^s} + O(n^{-s-1}), \tag{97}$$

$$\zeta(s) \textit{ is Riemann zeta function.}\Bigg)$$

Then we have

$$\frac{1}{T}\sum_{t=1}^T \left\|\nabla f(x_t)\right\|^2 \le \frac{2}{C_l}\left[(M_f - f^*)\alpha_0 T^{\eta-1} + \frac{LC_u^2\sigma^2\alpha_0}{2(1-\eta)}T^{-\eta}\right] \tag{98}$$

$\square$

### 2.2.1 Validation on numerical accuracy of sum of generalized harmonic series

We performed experiments to test the accuracy of the analytical expression of sum of harmonic series. We numerically calculate $\sum_{i=1}^N \frac{1}{i^\eta}$ for $\eta$ varying from 0.5 to 0.999, and for $N$ ranging from $10^3$ to $10^7$ in the log-grid. We calculate the error of the analytical expression by Eq. (97), and plot the error in Fig. 12. Note that the $y$-axis has a unit of $10^{-7}$, while the sum is typically on the order of $10^3$, this implies that expression Eq. (97) is very accurate and the relative error is on the order of $10^{-10}$. Furthermore, note that this expression is accurate even when $\eta = 0.5$.

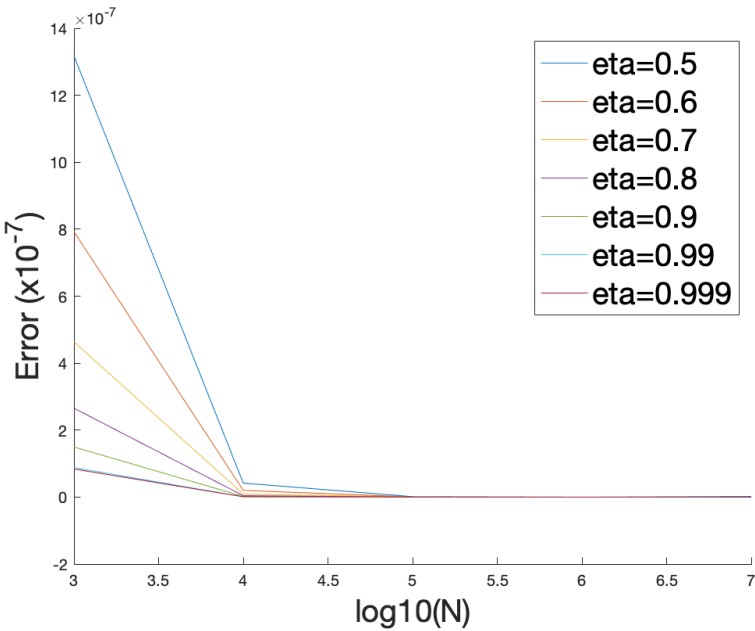

Figure 12: The error between numerical sum for $\sum_{i=1}^{N} \frac{1}{i^{\eta}}$ and the analytical form.

## 2.3 Convergence analysis of Async-moment-optimizers in stochastic non-convex optimization

**Lemma 2.2.** *Let $m_t = \beta_1 m_{t-1} + (1 - \beta_1)g_t$, let $A_t \in \mathbb{R}^d$, then*

$$\left\langle A_t, g_t \right\rangle = \frac{1}{1 - \beta_1}\left(\left\langle A_t, m_t \right\rangle - \left\langle A_{t-1}, m_{t-1} \right\rangle\right) + \left\langle A_{t-1}, m_{t-1} \right\rangle + \frac{\beta_1}{1 - \beta_1}\left\langle A_{t-1} - A_t, m_{t-1} \right\rangle \tag{99}$$

**Theorem 2.3.** *Under assumptions 1-4, $\beta_1 < 1, \beta_2 < 1$, also assume $A_{t+1} \leq A_t$ element-wise which can be achieved by tracking maximum of $s_t$ as in AMSGrad, $f$ is upper bounded by $M_f$, $\|g_t\|_{\infty} \leq M_g$, with learning rate schedule as*

$$\alpha_t = \alpha_0 t^{-\eta}, \quad \alpha_0 \leq \frac{C_l}{LC_u^2}, \quad \eta \in (0.5, 1] \tag{100}$$

*the sequence is generated by*

$$x_{t+1} = x_t - \alpha_t A_t m_t \tag{101}$$

*then we have*

$$\frac{1}{T}\sum_{t=1}^{T}\left\|\nabla f(x_t)\right\|^2 \leq \frac{1}{\alpha_0 C_l}T^{\eta-1}\left[M_f - f^* + EM_g^2\right] \tag{102}$$

*where*

$$E = \frac{\beta_1^2}{4L(1 - \beta_1)^2} + \frac{1}{1 - \beta_1}\alpha_0 M_g + \left(\frac{\beta_1}{1 - \beta_1} + \frac{1}{2}\right)L\alpha_0^2 C_u^2 \frac{1}{1 - 2\eta} \tag{103}$$

*Proof.* Let $A_t = \alpha_t A_t \nabla f(x_t)$ and let $A_0 = A_1$, we have

$$\sum_{t=1}^{T} \left\langle A_t, g_t \right\rangle = \frac{1}{1-\beta_1} \left\langle A_T, m_T \right\rangle + \sum_{t=1}^{T} \left\langle A_{t-1}, m_{t-1} \right\rangle + \frac{\beta_1}{1-\beta_1} \sum_{t=1}^{T} \left\langle A_{t-1} - A_t, m_{t-1} \right\rangle \quad (104)$$

$$= \frac{\beta_1}{1-\beta_1} \left\langle A_T, m_T \right\rangle + \sum_{t=1}^{T} \left\langle A_t, m_t \right\rangle + \frac{\beta_1}{1-\beta_1} \sum_{t=0}^{T-1} \left\langle A_t - A_{t+1}, m_t \right\rangle \quad (105)$$

First we derive a lower bound for Eq. (105).

$$\left\langle A_t, g_t \right\rangle = \left\langle \alpha_t A_t \nabla f(x_t), g_t \right\rangle \quad (106)$$

$$= \left\langle \alpha_t A_t \nabla f(x_t) - \alpha_{t-1} A_{t-1} \nabla f(x_t), g_t \right\rangle + \left\langle \alpha_{t-1} A_{t-1} \nabla f(x_t), g_t \right\rangle \quad (107)$$

$$= \left\langle \alpha_{t-1} A_{t-1} \nabla f(x_t), g_t \right\rangle - \left\langle (\alpha_{t-1} A_{t-1} - \alpha_t A_t) \nabla f(x_t), g_t \right\rangle \quad (108)$$

$$\geq \left\langle \alpha_{t-1} A_{t-1} \nabla f(x_t), g_t \right\rangle - \left\| \nabla f(x_t) \right\|_{\infty} \left\| \alpha_{t-1} A_{t-1} - \alpha_t A_t \right\|_1 \left\| g_t \right\|_{\infty} \quad (109)$$

$$\left( By \ H\ddot{o}lder's \ inequality \right)$$

$$\geq \left\langle \alpha_{t-1} A_{t-1} \nabla f(x_t), g_t \right\rangle - M_g^2 \left( \left\| \alpha_{t-1} A_{t-1} \right\|_1 - \left\| \alpha_t A_t \right\|_1 \right) \quad (110)$$

$$\left( Since \ \left\| g_t \right\|_{\infty} \leq M_g, \alpha_{t-1} \geq \alpha_t > 0, A_{t-1} \geq A_t > 0 \ element\text{-}wise \right) \quad (111)$$

Perform telescope sum, we have

$$\sum_{t=1}^{T} \left\langle A_t, g_t \right\rangle \geq \sum_{t=1}^{T} \left\langle \alpha_{t-1} A_{t-1} \nabla f(x_t), g_t \right\rangle - M_g^2 \left( \left\| \alpha_0 H_0 \right\|_1 - \left\| \alpha_T A_t \right\|_1 \right) \quad (112)$$

Next, we derive an upper bound for $\sum_{t=1}^{T} \left\langle A_t, g_t \right\rangle$ by deriving an upper-bound for the RHS of Eq. (105). We derive an upper bound for each part.

$$\left\langle A_t, m_t \right\rangle = \left\langle \alpha_t A_t \nabla f(x_t), m_t \right\rangle = \left\langle \nabla f(x_t), \alpha_t A_t m_t \right\rangle \quad (113)$$

$$= \left\langle \nabla f(x_t), x_t - x_{t+1} \right\rangle \quad (114)$$

$$\leq f(x_t) - f(x_{t+1}) + \frac{L}{2} \left\| x_{t+1} - x_t \right\|^2 \left( By \ L\text{-}smoothness \ of \ f \right) \quad (115)$$

Perform telescope sum, we have

$$\sum_{t=1}^{T} \left\langle A_t, m_t \right\rangle \leq f(x_1) - f(x_{T+1}) + \frac{L}{2} \sum_{t=1}^{T} \left\| \alpha_t A_t m_t \right\|^2 \quad (116)$$

$$\left\langle A_t - A_{t+1}, m_t \right\rangle = \left\langle \alpha_t A_t \nabla f(x_t) - \alpha_{t+1} A_{t+1} \nabla f(x_{t+1}), m_t \right\rangle \quad (117)$$

$$= \Big\langle \alpha_t A_t \nabla f(x_t) - \alpha_t A_t \nabla f(x_{t+1}), m_t \Big\rangle$$

$$+ \Big\langle \alpha_t A_t \nabla f(x_{t+1}) - \alpha_{t+1} A_{t+1} \nabla f(x_{t+1}), m_t \Big\rangle \tag{118}$$

$$= \Big\langle \nabla f(x_t) - \nabla f(x_{t+1}), \alpha_t A_t m_t \Big\rangle + \Big\langle (\alpha_t A_t - \alpha_{t+1} A_{t+1}) \nabla f(x_t), m_t \Big\rangle \tag{119}$$

$$= \Big\langle \nabla f(x_t) - \nabla f(x_{t+1}), x_t - x_{t+1} \Big\rangle + \Big\langle \nabla f(x_t), (\alpha_t A_t - \alpha_{t+1} A_{t+1}) m_t \Big\rangle \tag{120}$$

$$\leq L \Big\| x_{t+1} - x_t \Big\|^2 + \Big\langle \nabla f(x_t), (\alpha_t A_t - \alpha_{t+1} A_{t+1}) m_t \Big\rangle \tag{121}$$

$$\Big( \text{By smoothness of } f \Big)$$

$$\leq L \Big\| x_{t+1} - x_t \Big\|^2 + \Big\| \nabla f(x_t) \Big\|_\infty \Big\| \alpha_t A_t - \alpha_{t+1} A_{t+1} \Big\|_1 \Big\| m_t \Big\|_\infty \tag{122}$$

$$\Big( \text{By Hölder's inequality} \Big)$$

$$\leq L \Big\| x_{t+1} - x_t \Big\|^2 + M_g^2 \Big( \Big\| \alpha_t A_t \Big\|_1 - \Big\| \alpha_{t+1} A_{t+1} \Big\|_1 \Big) \tag{123}$$

$$\Big( \text{Since } \alpha_t \geq \alpha_{t+1} \geq 0, A_t \geq A_{t+1} \geq 0, \text{ element-wise} \Big) \tag{124}$$

Perform telescope sum, we have

$$\sum_{t=1}^{T-1} \Big\langle A_t - A_{t+1}, m_t \Big\rangle \leq L \sum_{t=1}^{T-1} \Big\| \alpha_t A_t m_t \Big\|^2 + M_g^2 \Big( \Big\| \alpha_1 H_1 \Big\|_1 - \Big\| \alpha_T A_t \Big\|_1 \Big) \tag{125}$$

We also have

$$\Big\langle A_T, m_T \Big\rangle = \Big\langle \alpha_T A_t \nabla f(x_T), m_T \Big\rangle = \Big\langle \nabla f(x_T), \alpha_T A_t m_T \Big\rangle \tag{126}$$

$$\leq L \frac{1 - \beta_1}{\beta_1} \Big\| \alpha_T A_t m_T \Big\|^2 + \frac{\beta_1}{4L(1 - \beta_1)} \Big\| \nabla f(x_T) \Big\|^2 \tag{127}$$

$$\Big( \text{By Young's inequality} \Big)$$

$$= L \frac{1 - \beta_1}{\beta_1} \Big\| \alpha_T A_t m_T \Big\|^2 + \frac{\beta_1}{4L(1 - \beta_1)} M_g^2 \tag{128}$$

Combine Eq. (116), Eq. (125) and Eq. (128) into Eq. (105), we have

$$\sum_{t=1}^{T} \Big\langle A_t, g_t \Big\rangle \leq \frac{\beta_1}{1 - \beta_1} \Big\langle A_T, m_T \Big\rangle + f(x_1) - f(x_{T+1}) + \frac{L}{2} \sum_{t=1}^{T} \Big\| \alpha_t A_t m_t \Big\|^2$$

$$+ \frac{\beta_1}{1 - \beta_1} L \sum_{t=1}^{T-1} \Big\| \alpha_t A_t m_t \Big\|^2 + \frac{\beta_1}{1 - \beta_1} M_g^2 \Big( \Big\| \alpha_1 H_1 \Big\|_1 - \Big\| \alpha_T A_t \Big\|_1 \Big) \tag{129}$$

$$\leq f(x_1) - f(x_{T+1}) + \Big( \frac{\beta_1}{1 - \beta_1} + \frac{1}{2} \Big) L \sum_{t=1}^{T} \Big\| \alpha_t A_t m_t \Big\|^2$$

$$+ \left( \frac{\beta_1^2}{4L(1-\beta_1)^2} + \frac{\beta_1}{1-\beta_1} \left\| \alpha_1 H_1 \right\|_1 \right) M_g^2 \tag{130}$$

Combine Eq. (112) and Eq. (130), we have

$$\sum_{t=1}^{T} \left\langle \alpha_{t-1} A_{t-1} \nabla f(x_t), g_t \right\rangle - M_g^2 \left( \left\| \alpha_0 H_0 \right\|_1 - \left\| \alpha_T A_t \right\|_1 \right) \leq \sum_{t=1}^{T} \left\langle A_t, g_t \right\rangle$$

$$\leq f(x_1) - f(x_{T+1}) + \left( \frac{\beta_1}{1-\beta_1} + \frac{1}{2} \right) L \sum_{t=1}^{T} \left\| \alpha_t A_t m_t \right\|^2$$

$$+ \left( \frac{\beta_1^2}{4L(1-\beta_1)^2} + \frac{\beta_1}{1-\beta_1} \left\| \alpha_1 H_1 \right\|_1 \right) M_g^2 \tag{131}$$

Hence we have

$$\sum_{t=1}^{T} \left\langle \alpha_{t-1} A_{t-1} \nabla f(x_t), g_t \right\rangle \leq f(x_1) - f(x_{T+1}) + \left( \frac{\beta_1}{1-\beta_1} + \frac{1}{2} \right) L \sum_{t=1}^{T} \left\| \alpha_t A_t m_t \right\|^2$$

$$+ \left( \frac{\beta_1^2}{4L(1-\beta_1)^2} + \left\| \alpha_0 H_0 \right\|_1 + \frac{\beta_1}{1-\beta_1} \left\| \alpha_1 H_1 \right\|_1 \right) M_g^2 \tag{132}$$

$$\leq f(x_1) - f^* + \left( \frac{\beta_1}{1-\beta_1} + \frac{1}{2} \right) L \alpha_0^2 M_g^2 C_u^2 \sum_{t=1}^{T} t^{-2\eta}$$

$$+ \left( \frac{\beta_1^2}{4L(1-\beta_1)^2} + \left\| \alpha_0 H_0 \right\|_1 + \frac{\beta_1}{1-\beta_1} \left\| \alpha_1 H_1 \right\|_1 \right) M_g^2 \tag{133}$$

$$\leq f(x_1) - f^*$$

$$+ M_g^2 \left[ \frac{\beta_1^2}{4L(1-\beta_1)^2} + \left\| \alpha_0 H_0 \right\|_1 + \frac{\beta_1}{1-\beta_1} \left\| \alpha_1 H_1 \right\|_1 + \left( \frac{\beta_1}{1-\beta_1} + \frac{1}{2} \right) L \alpha_0^2 C_u^2 \frac{T^{1-2\eta}}{1-2\eta} \right] \tag{134}$$

$$\leq f(x_1) - f^* + M_g^2 \underbrace{\left[ \frac{\beta_1^2}{4L(1-\beta_1)^2} + \frac{1}{1-\beta_1} \alpha_0 M_g + \left( \frac{\beta_1}{1-\beta_1} + \frac{1}{2} \right) L \alpha_0^2 C_u^2 \frac{1}{1-2\eta} \right]}_{E} \tag{135}$$

Take expectations on both sides, we have

$$\sum_{t=1}^{T} \left\langle \alpha_{t-1} A_{t-1} \nabla f(x_t), \nabla f(x_t) \right\rangle \leq \mathbb{E} f(x_1) - f^* + E M_g^2 \leq M_f - f^* + E M_g^2 \tag{136}$$

Note that we have $\alpha_t$ decays monotonically with $t$, hence

$$\sum_{t=1}^{T} \left\langle \alpha_{t-1} A_{t-1} \nabla f(x_t), \nabla f(x_t) \right\rangle \geq \alpha_0 T^{-\eta} \sum_{t=1}^{T} \left\langle A_{t-1} \nabla f(x_t), \nabla f(x_t) \right\rangle \tag{137}$$

$$\geq \alpha_0 T^{1-\eta} C_l \left[ \frac{1}{T} \sum_{t=1}^{T} \left\| \nabla f(x_t) \right\|^2 \right] \tag{138}$$

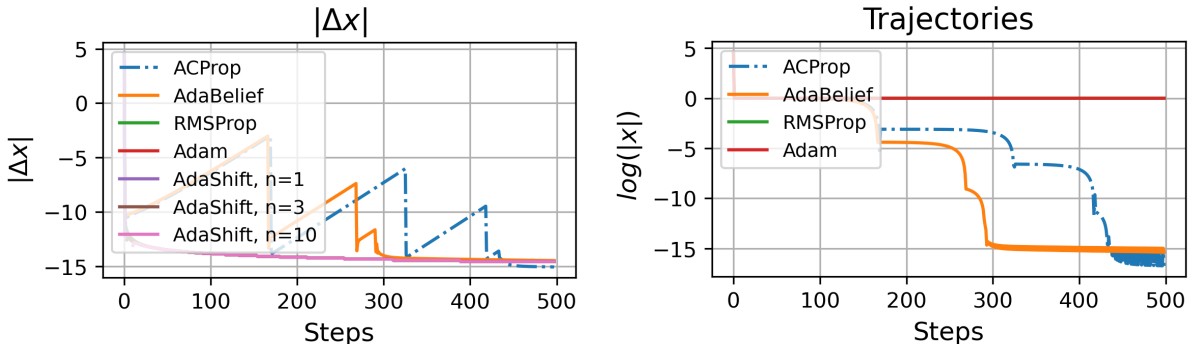

Figure 13: Behavior of ACProp for optimization of the function $f(x) = |x|$ with $lr = 0.00001$.

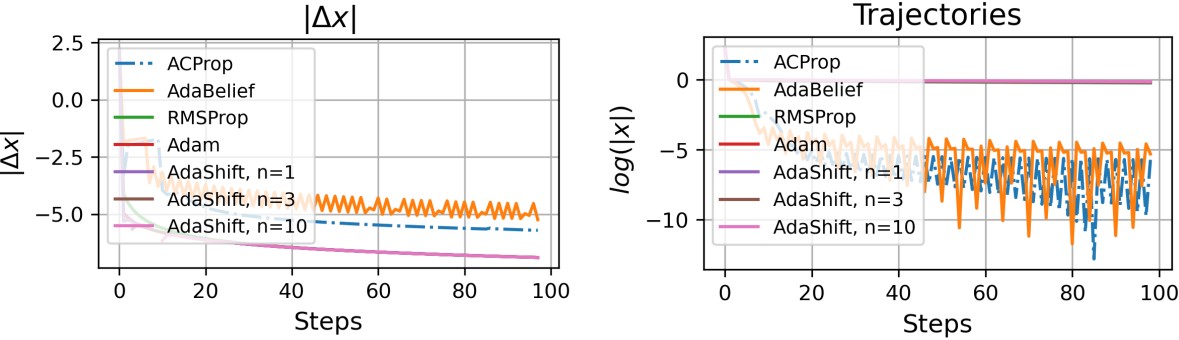

Figure 14: Behavior of ACProp for optimization of the function $f(x) = |x|$ with $lr = 0.01$.

Combine Eq. (136) and Eq. (138), assume $f$ is upper bounded by $M_f$, we have

$$\frac{1}{T} \sum_{t=1}^{T} \left|\left|\nabla f(x_t)\right|\right|^2 \leq \frac{1}{\alpha_0 C_l} T^{\eta-1} \left[M_f - f^* + EM_g^2\right] \tag{139}$$

$\square$

# 3 Experiments

## 3.1 Centering of second momentum does not suffer from numerical issues

Note that the centered second momentum $s_t$ does not suffer from numerical issues in practice. The intuition that "$s_t$ is an estimate of variance in gradient" is based on a strong assumption that the gradient follows a stationary distribution, which indicates that the true gradient $\nabla f_t(x)$ remains a constant function of $t$. In fact, $s_t$ tracks $EMA((g_t - m_t)^2)$, and it includes two aspects: the change in true gradient $||\nabla f_{t+1}(x) - \nabla f_t(x)||^2$, and the noise in gradient observation $||g_t - \nabla f_t(x)||^2$. In practice, especially in deep learning, the gradient suffers from large noise, hence $s_t$ does not take extremely small values.

Table 1: Hyper-parameters for ACProp in various experiments

|             | lr   | beta1 | beta2 | eps   |
|-------------|------|-------|-------|-------|
| ImageNet    | 1e-3 | 0.9   | 0.999 | 1e-12 |
| GAN         | 2e-4 | 0.5   | 0.999 | 1e-16 |
| Transformer | 5e-4 | 0.9   | 0.999 | 1e-16 |

Next, we consider an ideal case that the observation $g_t$ is noiseless, and conduct experiments to show that centering of second-momentum does not suffer from numerical issues. Consider the function $f(x) = |x|$ with initial value $x_0 = 100$, we plot the trajectories and stepsizes of various optimizers in Fig. 13 and Fig. 14 with initial learning rate $lr = 0.00001$ and $lr = 0.01$ respectively. Note that ACProp and AdaBelief take a large step at the initial phase, because a constant gradient is observed without noise. But note that the gradient remains constant only within half of the plane; when it cross the boundary $x = 0$, the gradient is reversed, hence $||\nabla f_{t+1}(x) - \nabla f_t(x)||^2 \neq 0$, and $s_t$ becomes a non-zero value when it hits a valley in the loss surface. Therefore, the stepsize of ACProp and AdaBelief automatically decreases when they reach the local minimum. As shown in Fig. 13 and Fig. 14, ACProp and AdaBelief does not take any extremely large stepsizes for both a very large (0.01) and very small (0.00001) learning rates, and they automatically decrease stepsizes near the optimal. We do not observe any numerical issues even for noise-free piecewise-linear functions. If the function is not piecewise linear, or the gradient does not remain constant within any connected set, then $||\nabla f_{t+1}(x) - \nabla f_t(x)||^2 \neq 0$ almost everywhere, and the numerical issue will never happen.

The only possible case where centering second momentum causes numerical issue has to satisfy two conditions simultaneously: (1) $||\nabla f_{t+1}(x) - \nabla f_t(x)||^2 = 0, \forall t$ and (2) $g_t$ is a noise-free observation of $\nabla f(x)$. This is a trivial case where the loss surface is linear, and gradient is noise-free. This is case is almost never encountered in practice. Furthermore, in this case, $s_t = 0$ and ACProp reduces to SGD with stepsize $1/\epsilon$. But note that the optimal is $-\infty$ and achieved at $\infty$ or $-\infty$, taking a large stepsize $1/\epsilon$ is still acceptable for this trivial case.

## 3.2  Image classification with CNN

We performed extensive hyper-parameter tuning in order to better compare the performance of different optimizers: for SGD we set the momentum as 0.9 which is the default for many cases, and search the learning rate between 0.1 and $10^{-5}$ in the log-grid; for other adaptive optimizers, including AdaBelief, Adam, RAdam, AdamW and AdaShift, we search the learning rate between 0.01 and $10^{-5}$ in the log-grid, and search $\epsilon$ between $10^{-5}$ and $10^{-10}$ in the log-grid. We use a weight decay of 5e-2 for AdamW, and use 5e-4 for other optimizers. We conducted experiments based on the official code for AdaBound and AdaBelief [1].

We further test the robustness of ACProp to values of hyper-parameters $\beta_1$ and $\beta_2$. Results are shown in Fig. 17 and Fig. 19 respectively. ACProp is robust to different values of $\beta_1$, and is more sensitive to values of $\beta_2$.

---

[1]https://github.com/juntang-zhuang/Adabelief-Optimizer

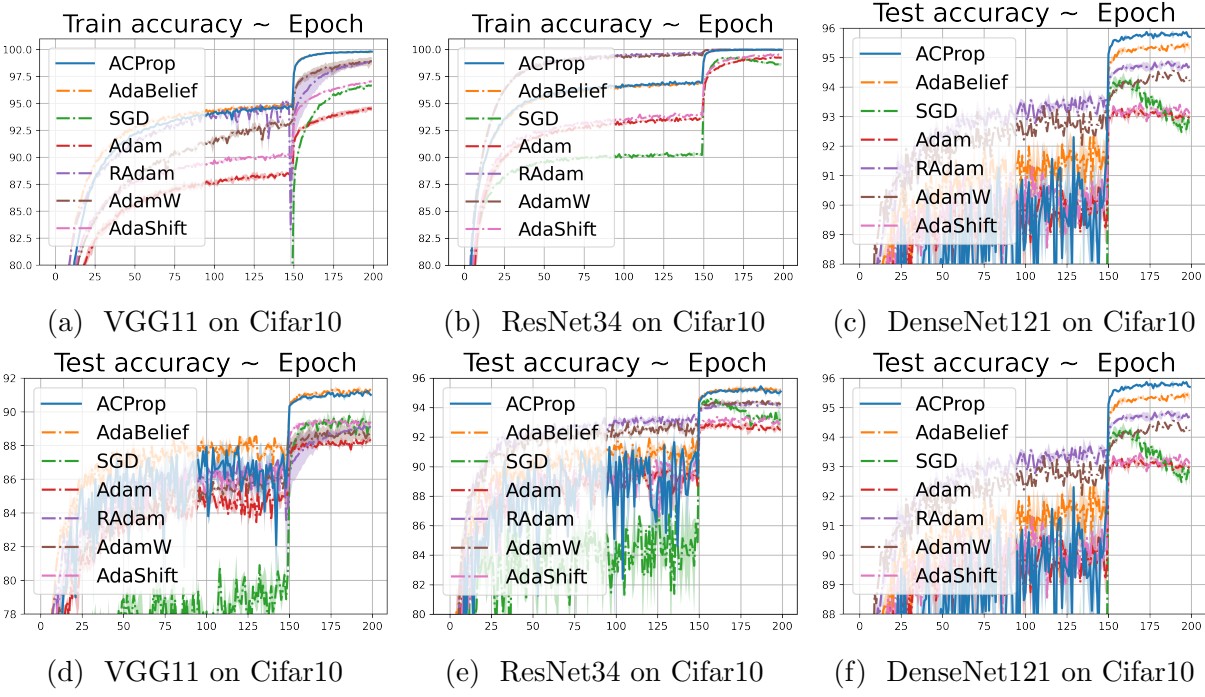

(a) VGG11 on Cifar10   (b) ResNet34 on Cifar10   (c) DenseNet121 on Cifar10

(d) VGG11 on Cifar10   (e) ResNet34 on Cifar10   (f) DenseNet121 on Cifar10

Figure 15: Training (top row) and test (bottom row) accuracy of CNNs on Cifar10 dataset.

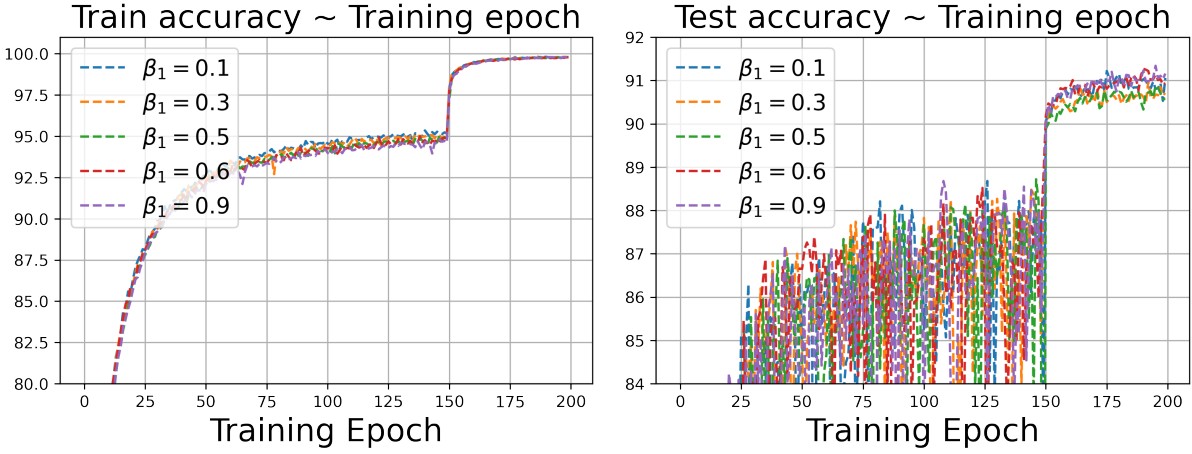

Figure 17: The training and test accuracy curve of VGG11 on CIFAR10 with different $\beta_1$ values.

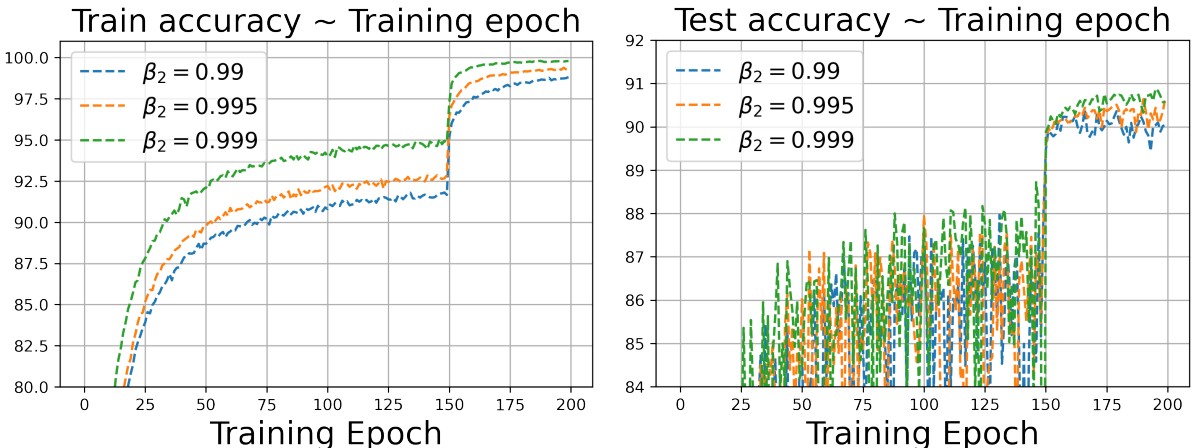

Figure 19: The training and test accuracy curve of VGG11 on CIFAR10 with different $\beta_2$ values.

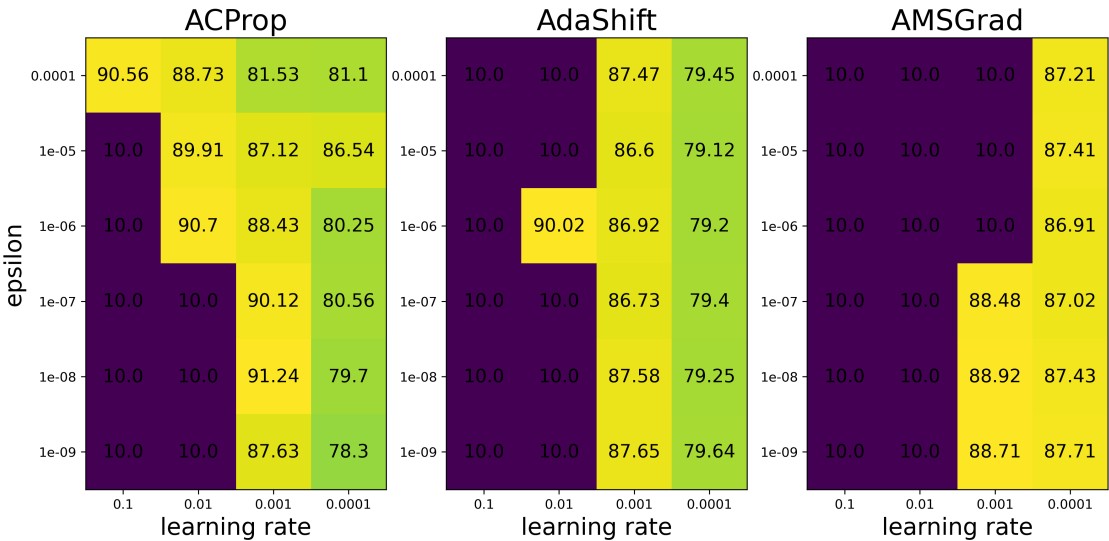

Figure 20: Test accuracy of VGG-11 on CIFAR10 trained under various hyper-parameter settings with different optimizers

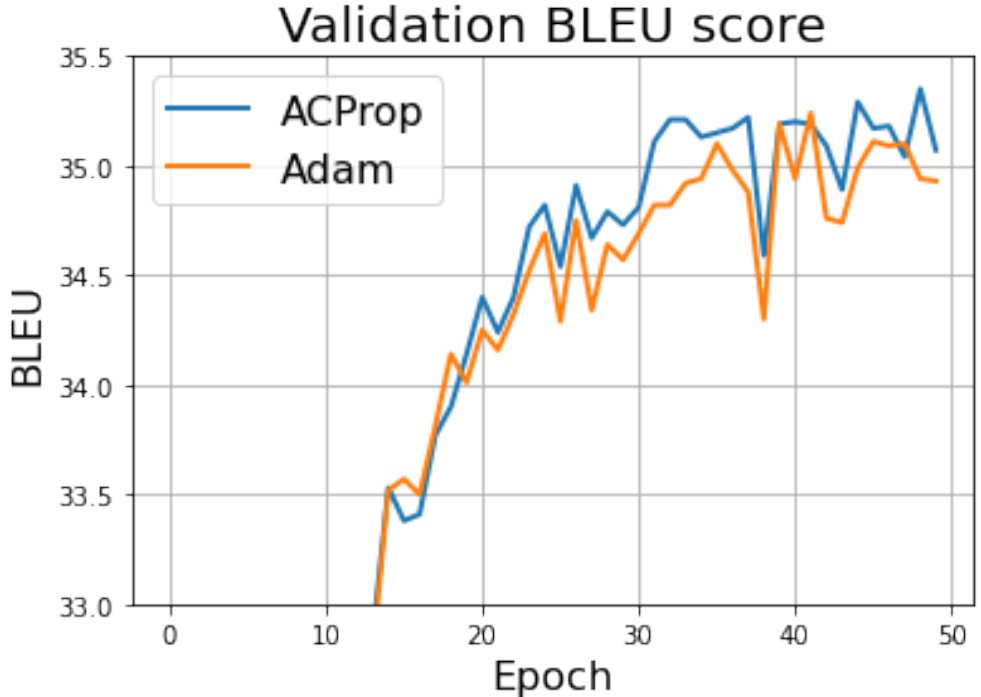

Figure 21: BLEU score on validation set of a Transformer-base trained with ACProp and Adam

## 3.3 Neural Machine Translation with Transformers

We conducted experiments on Neural Machine Translation (NMT) with transformer models. Our experiments on the IWSLT14 DE-EN task is based on the 6-layer transformer-base model in fairseq implementation [2]. For all methods, we use a learning rate of 0.0002, and standard invser sqrt learning rate schedule with 4,000 steps of warmup. For other tasks, our experiments are based on an open-source implementation[3] using a 1-layer Transformer model. We plot the BLEU score on validation set varying with training epoch in Fig. 21, and ACProp consistently outperforms Adam throughout the training.

## 3.4 Generative adversarial networks

The training of GANs easily suffers from mode collapse and numerical instability [3], hence is a good test for the stability of optimizers. We conducted experiments with Deep Convolutional GAN (DCGAN) [4], Spectral-Norm GAN (SNGAN) [5], Self-Attention GAN (SAGAN) [6] and Relativistic-GAN (RLGAN) [7]. We set $\beta_1 = 0.5$, and search for $\beta_2$ and $\epsilon$ with the same schedule as previous section. Our experiments are based on an open-source implementation [4].

---

[2]https://github.com/pytorch/fairseq
[3]https://github.com/DevSinghSachan/multilingual_nmt
[4]https://github.com/POSTECH-CVLab/PyTorch-StudioGAN

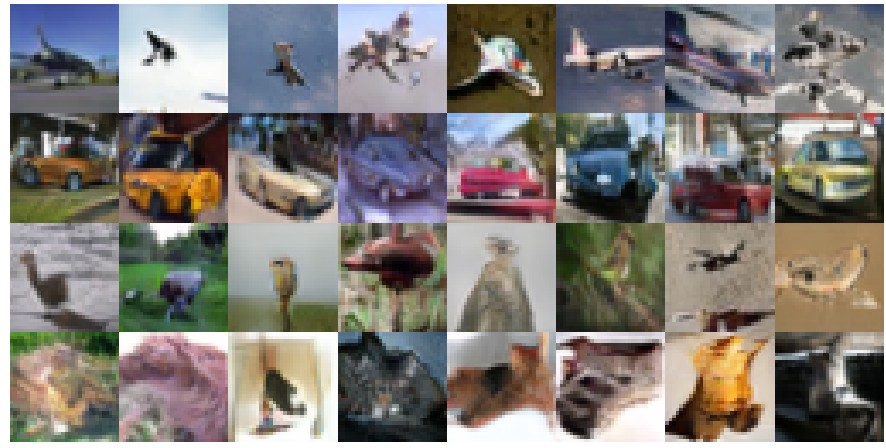

Figure 22: Generated figures by the SN-GAN trained with ACProp.

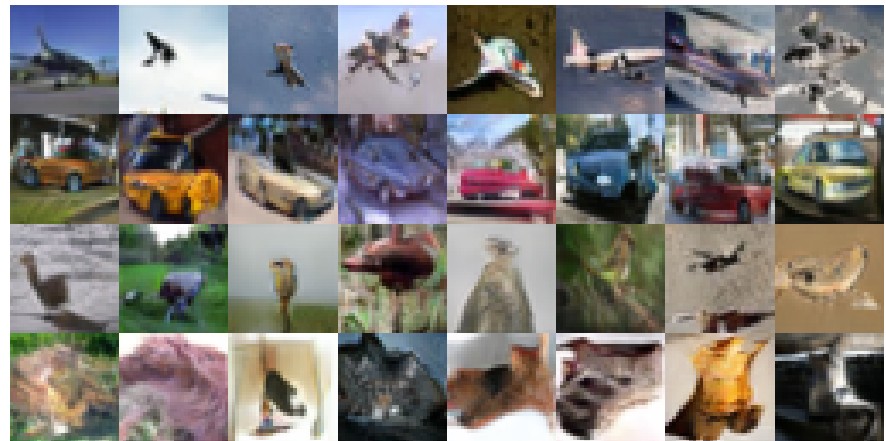

Figure 23: Generated figures by the SA-GAN trained with ACProp.

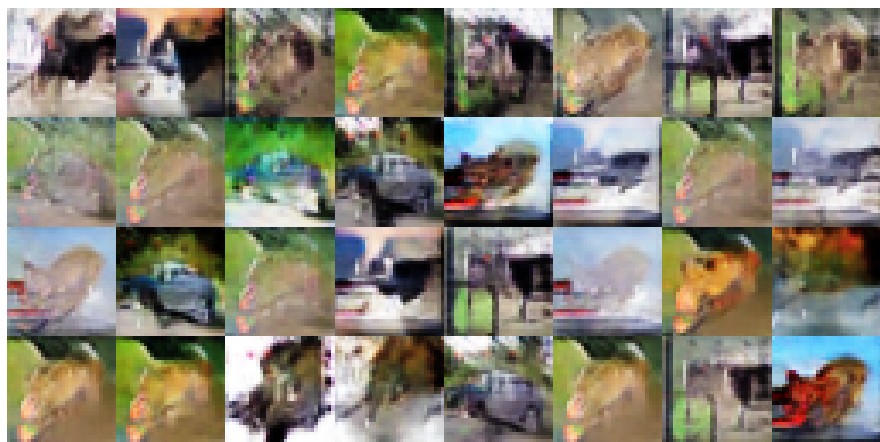

Figure 24: Generated figures by the DC-GAN trained with ACProp.

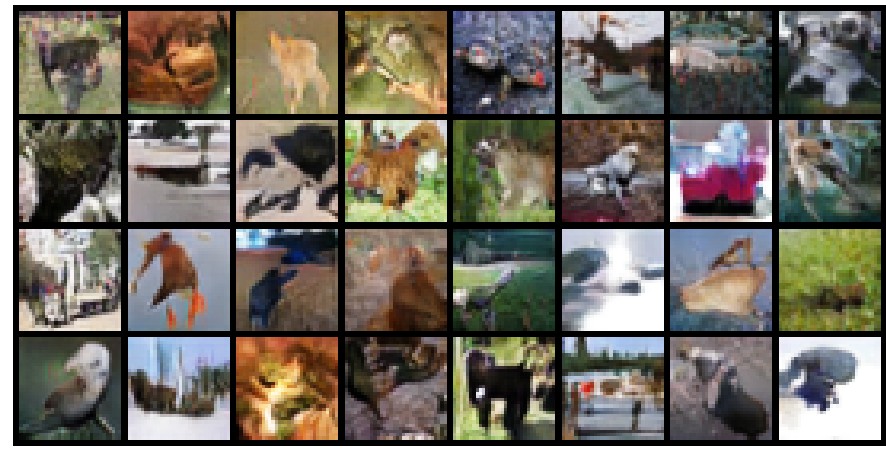

Figure 25: Generated figures by the RL-GAN trained with ACProp.