# OpenReview forum: "Momentum Centering and Asynchronous Update for Adaptive Gradient Methods"
_NeurIPS.cc/2021/Conference — NeurIPS 2021 Poster_

### Official Review · Reviewer_yEdV · 2021-07-06

**Rating:** 7
**Confidence:** 4

**Summary:**

This paper proposes to combine centering of second momentum with asynchronous update for adaptive learning rate optimizer, showing that the proposed optimizer has: (1) a weaker convergence condition on sequence online convex optimization problem than its counterparts; (2) a convergence rate that matches the oracle; (3)  strong empirical performance in terms of generalization and training stability.

**Main Review:**

This paper is overall well written and has a moderate theoretical contribution. However, the experiments can be conducted and/or explained better to help understand the results. Besides, some points need to be clarified.

Issues:

- It should be mentioned that ACProp in the experiments does not use momentum, i.e., mt is only used for the estimation of st.

- Given the previous point, the comparison between ACProp and other optimizers can be done better, e.g., disable momentum for other optimizers. Otherwise, it is hard to conclude whether centering is the key to better empirical performance.


- The learning rate used In Figure 6 (a) and (b) should be given. Given that centered version generally has a large effective learning rate at the beginning, using a same learning rate for centered and uncentered would be inappropriate. If it is using a same learning rate, it might be better to tune the learning rate such that the curves in the initial stage basically overlap.


- It is unclear why the proposed optimizer leads to better generalization. It is better to have some investigations and/or explanations in the paper.

- It is interesting to show how rectification (RAdam) and/or decoupled weight decay (AdamW) work when combined with ACProp and other optimizers. It might help to understand the experiments, e.g., what are the fundamental issues in these optimizers and how the issues are solved.


Details:

- Figure 8, there are typos for the range of eps. It is also not inconsistent with the correlated figure in the Appendix.


- It is better to mention that AdaShift assumed the independence of gradients, which holds if minibatches are randomly sampled and is violated in the sequence online convex optimization problem.

**Score increased from 6 to 7, after rebuttal.**

**Time Spent Reviewing:**

8

---

> ### Author Response · Authors · 2021-08-06
> **Response to reviewer yEdV**
>
> Thanks for your careful review, we address your concerns below.
>
> **1.** Thanks for the suggestion, we’ll emphasize this in the updated version.
>
> **2. Disable momentum for other optimizers**
>
> Considering suggestions from you and R1, we re-run experiments on Cifar10, and added RMSProp and AdaShift with n=1, these two optimizers use $g_t$ rather than the momentum version of gradient in the denominator. ACProp outperforms AdaShift with n=1, where the only difference is centering, validating its effectiveness. (results: https://openreview.net/forum?id=LY-o87_w_x4&noteId=T3Gi60h_c5)
>
> **3. Learning rate**
>
> Thanks for the suggestion, we’ll update the figure in revision. To remove the difference caused by learning rate, for all experiments we sweep through different combinations of learning rate and $\epsilon$, and plot a few examples in Fig.20 in the appendix. ACProp still outperforms other methods though their hyperparameters are tuned independently.
>
> **4. Reasons for better generalization**.
> * The improvements of ACProp over AdaBelief might come from that AdaBelief update direction is closer to “sign descent”, because the update in AdaBelief is $g_t / \sqrt{ \beta_2 s_{t-1} + (1-\beta_2) g_t^2 }$, when $g_t^2 \gg s_{t-1}$, the update is close to $1/\sqrt{1-\beta_2}$, regardless that the gradient of different coordinates might be very different (e.g. $\vert \Delta x_1 \vert = \vert \Delta x_2 \vert$ for a 2-D weight even though $g_1 \neq g_2$). This problem is alleviated in ACProp because the numerator and denominator are independent. For a detailed explanation and visualization on toy examples please refer to 1.b) and 1.c) in response to reviewer 5voH (https://openreview.net/forum?id=LY-o87_w_x4&noteId=aCdgVmT_Zcu).
> * The improvements of ACProp over AdaShift come from the momentum centering, similar to the reason that AdaBelief outperforms Adam.
>
> **5. Rectification and decoupled weight decay**.
>
> Thanks for the suggestion. We experimented with VGG on Cifar10 as suggested, and did not observe significant improvements by combining ACProp with rectify or decoupled decay (on the contrary, the accuracy drops a bit). We are not able to draw conclusions just based on the following experiment, it’s possible that rectification and decoupled decay work for large models, but we are unable to compare thoroughly within the rebuttal limit, and would investigate it in future work.
>
> **Accuracy of VGG on Cifar10**
>
> | ACProp 	| ACProp+Rectify 	| ACProp+decouple 	| ACProp+rectify+decouple 	|
> |---	|---	|---	|---	|
> | 91.26 	| 91.1 	| 90.33 	| 90.69 	|

---

> ### Comment · Reviewer_yEdV · 2021-08-28
> **Response to the rebuttal.**
>
> I have read other reviews and the authors' feedback. My overall evaluation of this paper remains unchanged.
>
> In particular, I don't think the authors' response to my concerning points 4 and 5 is what I want. (i) I am concerning the generalization, not optimization. Better optimization does not necessarily imply better generalization. (ii) I do not solely want some experiment results. I want explanation so as to better understand these results. More experiments are to help the analysis and verification.
>
> As Reviewer Vu9t commented, comparing optimizers can be very hard, and the experiments sometime are highly unreliable (have many tricky details). So, I think it is better to not just show experiment results, but also try to explain it.
>
> Given the not significant theoretical contribution and my lack of confidence of the presented empirical results, I would keep my score as "marginally above the acceptance threshold".

---

> > ### Author Response · Authors · 2021-08-30
> > **Updated response**
> >
> > Thanks for your updated review, we misunderstood your question as asking for intuitive explanations, here we provide a more formal discussion that update direction is closer to gradient direction rather than sign descent helps both optimization and **generalization**.
> >
> > **1. Why sign descent harms generalization.**.
> > **a)** We quote the theorem and discussion from [1]
> > > **Lemma 3.1** Suppose there exists a scalar $c$ such that $X sign(X^\top y) = cy$. Then, when initialized at $w_0$ = 0, AdaGrad, Adam, and RMSProp all converge to the unique solution $w \propto sign(X^\top y)$
> >
> > >In other words, whenever there exists a solution of $Xw = y$ that is proportional to $sign(X^\top y)$, this is precisely the solution to which all above methods converge.
> >
> > We demonstrate above discussion in a figure for better visualization https://www.dropbox.com/s/bqbigq8fiqtoihv/margin_analysis.png?dl=0 . For this linear separable binary classification problem, the decision boundary by sign descent is 45 degree to the axis, while the decision boundary by gradient descent is the maximum-margin solution [2]. Hence sign descent affects generalization in this simple case.
> >
> > **b)** For neural networks, we can locally approximate it as a linear model, then it reduces to a similar problem to a) in the high-dimensional case, hence sign descent also affects the generalization of neural networks.
> >
> > **c)** In our previous response to you, with experimental validation in toy example, we confirmed that update direction of ACProp is closer to gradient direction rather than sign descent, while other adaptive optimizers behave closer to sign descent.
> >
> > **2. Why momentum-centering helps generalization**.
> > The centered second momentum is expected to have smaller values than uncentered momentum, hence the effective update stepsize is larger than the centered version, therefore it’s easier to escape bad local minima and find flat local minima, hence improve the generalization. The empirical validation (value of centered momentum vs uncentered momentum) is shown in Fig.6 in the main paper.
> >
> > Also, note that the ratio (centered momentum / uncentered momentum)  is data-dependent and different weights could have different scales, hence in general it’s hard to compensate by using a different base learning rate (a single data-independent scalar for all weights), see fig.8 on experiments with different learning rates, ACProp outperforms both AdaBelief and AdaShift, the same phenomena are observed with a cosine learning rate schedule (see updated response to reviewer Vu9t, each optimizer is under heavy hyper-param search https://openreview.net/forum?id=LY-o87_w_x4&noteId=GpHCuq0JL1)
> >
> > **3.Our theoretical contribution**.
> > Our theoretical contribution are two aspects:
> > > 1) we identify the weaker convergence condition for ACProp
> > > 2) we proved the convergence rate of $O(1/\sqrt{T})$, though the novel idea of the async update is first proposed in AdaShift, the authors did not provide a theoretical analysis on convergence rate, hence our work is a completion to the theoretical analysis.
> >
> > **4.Empirical validations**.
> > We respectfully disagree that our experiments are “unreliable”. We believe our experimental results are very strong and convincing, because we are comparing them to the following:
> > >**a) Reported results in the literature.** (3 experiments on ImageNet) For ImageNet, we compare with several different literatures [3,4,5], and achieve significant improvement compared with all of them.
> >
> > >**b) Results from high-starred github repositories.** (9 experiments) Our results for GAN, reinforcement learning and NLP tasks are github repos >1k stars, and our reproduced results are often better than reported.
> >
> > >**c) Results with extensive hyper-parameter search.** (1 experiment) There's no benchmark result for optimization on Cifar10, and our experiments in the submission was following literature [6], we thank the reviewer for pointing out the effect of learning rate schedule and weight decay, and conducted more experiments and demonstrated our improvement under new settings, please see the response to reviewer Vu9t
> >
> > Because our baselines are strong, and we achieve the best in 11 out of 13 experiments with statistically significant improvement, hence we believe our empirical validations are solid.
> >
> > [1] Wilson, Ashia C., et al. "The marginal value of adaptive gradient methods in machine learning." arXiv preprint arXiv:1705.08292 (2017).
> > [2] https://people.eecs.berkeley.edu/~jrs/189/lec/03.pdf.
> > [3] Choi, Dami, et al. "On empirical comparisons of optimizers for deep learning." https://openreview.net/forum?id=HygrAR4tPS.
> > [4] Defazio, Aaron, and Samy Jelassi. "Adaptivity without compromise: a momentumized, adaptive, dual averaged gradient method for stochastic optimization." arXiv preprint arXiv:2101.11075 (2021). by Facebook AI Research.
> > [5] Xie, Zeke, et al. "Positive-Negative Momentum: Manipulating Stochastic Gradient Noise to Improve Generalization." ICML 2021.
> > [6] Luo, Liangchen, et al. "Adaptive gradient methods with dynamic bound of learning rate." arXiv preprint arXiv:1902.09843 (2019).

---

> ### Comment · Reviewer_yEdV · 2021-08-30
> **I have raised my score to 7.**
>
> The authors have made a mass of additional experiments and explanations, trying to convince the reviewers. I feel my concerns are basically addressed. Hence, I raised my score to 7, assuming that these issues have already been well-addressed in the manuscript.

---

> > ### Author Response · Authors · 2021-08-30
> > **Thanks for your review**
> >
> > Thanks for your review and discussion, we appreciate your help and efforts!

---

### Official Review · Reviewer_Argb · 2021-07-16

**Rating:** 6
**Confidence:** 3

**Summary:**

This work studies the problem of adaptive gradient methods and it is relevant to the conference. The paper provides momentum centering and asynchronous update for adaptive gradient methods that combines several existing works. The paper prove its algorithm the standard $1/\sqrt{T}$ convergence rate for stochastic nonconvex case. There are some experiments on VGG + CIFAR10 associated with the paper, thus the practical performance of the approach is also validated.




**Limitations And Societal Impact:**

There is no discussion on the limitations and potential negative societal impact of their work.

**Main Review:**

The main contribution of the paper is the offering of momentum centering and asynchronous update for adaptive gradient methods. The problem is well-motivated and the writing is clear. There are some simulation results associated with the paper thus the proposed algorithm seems work from both the theory and simulations.

In short, the paper is technically sound and the developments are clear. The derived algorithm and rates seems to be correct and may be a useful contribution to the literature on adaptive gradient methods, showing a modest improvement over the state of the art. ​

**Time Spent Reviewing:**

3

---

> ### Author Response · Authors · 2021-08-06
> **Thanks for your review**
>
> Thanks for your review, please let us know if you need any clarifications on our paper.

---

> > ### Comment · Reviewer_Argb · 2021-08-25
> > **thanks for the additional experiments**
> >
> > thanks for the additional experiments with ResNet50 on ImageNet. I keep my score unchanged.

---

### Official Review · Reviewer_5voH · 2021-07-16

**Rating:** 5
**Confidence:** 3

**Summary:**

This paper proposed a new adaptive optimizer, ACProp, which combines centering of second momentum and asynchronous update. In essence, ACProp modifies an existing method AdaBelief by using s_{t-1} instead of s_{t} as the update denominator. It has been shown that ACProp has a weaker convergence condition in the example by Reddi et al. (2018). They also proved that ACProp has the same convergence rate of O(1\\sqrt{T}), which matches the oracle rate for the stochastic nonconvex case.

**Limitations And Societal Impact:**

Limitations have been discussed in Section 3 and 4.
No negative societal impact is applicable.

**Main Review:**

The proposed method is very similar to AdaBelief. The only difference is that ACProp uses s_{t-1} instead of s_{t} as the update denominator. It’s not clear how much difference there is between s_{t-1} and s_{t} during most period of the training procedure (i.e., when training loss curve starts to enter the flat part and only slowly decreases), especially given that \beta_2 is normally set to a value that is very close to 1 (e.g., 0.99 or 0.999).

What’s more, although the theoretical analysis of the proposed method ACProp looks interesting, I am concerned with how significant the change from AdaBelief to ACProp is in terms of the practical performance. The theoretical analysis has shown that ACProp is better than AdaBelief in some cases (e.g., weaker convergence condition in the example by Reddi et al. (2018)). However, in practice these cases might not be common enough to set ACProp clearly apart from AdaBelief. Actually, from the experiments in this paper, I tend to have the impression that ACProp has very close performance to AdaBelief in many cases (sometimes even worse than AdaBelief: see the row JA-EN in Table 3 and the row RLGAN in Table 4). For example, in Figure 7 (left and middle), blue curve (ACProp) and orange curve (AdaBelief) reach identical final test accuracy. In Table 2, the difference between AdaBelief and ACProp is only 0.38% (not sure how much of it might come from random noise), while the difference between AdaBelief and Adam is much larger, i.e., around 3.5%. Results from Table 3 and Table 4 also seem to suggest that ACProp has very similar performance with AdaBelief in most cases.

Given the above reasons, I am concerned that ACProp is just an incremental improvement over AdaBelief in practice.


**Time Spent Reviewing:**

6

---

> ### Author Response · Authors · 2021-08-06
> **Response to reviewer 5voH**
>
> Thanks for your review, we address the theoretical and empirical differences between AdaBelief and ACProp below.
>
> **1. Theoretical differences between ACProp and AdaBelief**
>
> We emphasize that a small difference in coding does not imply a small difference in the algorithm, examples include SGD vs SGD+momentum, Adam vs RMSProp, AdaBelief vs Adam, the change for all above modifications is one line of code, but above modifications are not “incremental”. The same is with ACProp vs AdaBelief.
>
> **a) Convergence rate and convergence condition is different**.
> Theoretically, ACProp achieves a convergence rate of $O(1/\sqrt{T})$, which is a big improvement over AdaBelief $O(logT/\sqrt{T})$. Furthermore, we show that ACProp has a much wider range of convergence in the hyper-parameter plane.
>
> **b) Trajectories during optimization are different for ACProp and AdaBelief**.
> It’s inappropriate to compare $s_t$ with $s_{t-1}$ at the same point, because ACProp and AdaBelief take different trajectories, and the accumulated difference is getting larger with training for more steps.
>
> We directly visualize the trajectories to show the difference is significant. Using all default hyper-params ($\beta_1=0.9, \beta_2=0.999$), we plot an example with the Rosenbrock function as follow: trajectories(https://www.dropbox.com/s/q4bk3bmw2j400af/rosenbrock_acprop.gif?dl=0), distance to optimum (https://www.dropbox.com/s/460hy40ctovbpot/dist_traj.png?dl=0 ), it’s clear that the optimization trajectory of AdaBelief is very different from ACProp, and ACProp trains much faster. We observe similar phenomena in other toy examples.
>
> Considering ACProp uses $g_t$ while AdaBelief uses $m_t$ in numerator, we also modified AdaBelief to use $g_t$ (named as AdaBeliefG, link to results: https://www.dropbox.com/s/qrmkvwyfd5dmfoo/rosenbrock_adabeliefG.gif?dl=0), and the trajectory almost overlaps with original AdaBelief. Hence we conclude that the trajectory difference between ACProp and AdaBelief is caused by the async update.
>
> **c) Intuitive explanation for the difference between ACProp and AdaBelief**.
> AdaBelief update is closer to sign descent (45\degree to the axis) than ACProp. This difference is just caused by the sync vs async update.
> The update for ACProp is        $g_t / \sqrt{s_{t-1}}$
> the update for AdaBelief is       $g_t / \sqrt{ \beta_2 s_{t-1} + (1.0-\beta_2) (g_t-m_{t-1})^2 } $
>
> Consider the case where the trajectory falls into a sharp valley, so from a turning point the gradient is much larger, e.g. $g_t^2 \gg s_{t-1}$.
> For simplicity, consider the weight is a 2-D vector $[x1,x2]$ with gradient $[g_{t,1}, g_{t,2}]$.
>
> * For AdaBelief, $\Delta x_1 = g_{t,1}/\sqrt{ \beta_2 s_{t-1,1} + (1.0-\beta_2) (g_{t,1}-m_{t-1,1})^2 } \approx 1/\sqrt{ (1.0-\beta_2) } $ since $g_t^2  \gg s_{t-1}, g_t^2 \gg m_{t-1}^2$, we can get $ \vert \Delta x_1 \vert \approx \vert \Delta x_2 \vert$, even though $g_{t,1}$ and $g_{t,2}$ are very different. Hence AdaBelief updates closer to sign descent (equal update stepsize for two coordinates) even though the true gradient have very different scales in different coordinates (e.g. $g_{t,1}\gg g_{t,2}$)
>
> * For ACProp, $\Delta x_1 = g_{t,1} / \sqrt{s_{t-1,1}}$, $\Delta x_2 = g_{t,2} / \sqrt{s_{t-1,2}}$, hence we **don’t** have $\vert \Delta x_1 \vert \approx \vert \Delta x_2 \vert$ as in AdaBelief, because $g_t$ only appears in numerator but not denominator in ACProp.
>
> The trajectory plot in b) is a good validation of above intuition. ACProp and SGD updates in the direction of true gradient (most obvious for visualization at beginning), while Adam, RMSProp and AdaBelief update direction is close to (1,1). This mismatch between update direction and true gradient direction could happen in both early phases and late phases (sharp local minima or enter a valley where the gradient flow turns direction) for AdaBelief, but would be much alleviated with ACProp. This could also be the reason that ACProp achieves even better generalization than AdaBelief.
>
> **2. ACProp outperforms AdaBelief significantly in practice**.
> ACProp outperforms other optimizers (including AdaBelief) in **11 out of 13** experiments. We argue that R2’s impression of ACProp’s improvement over AdaBelief is “incremental” comes from the **unfair baseline**. The reviewer is comparing AdaBelief to Adam as a baseline, while comparing ACProp to AdaBelief as baseline which is much stronger than Adam. A metaphor is “it’s much harder to improve from 95 to 100 (+5) than from 85 to 95 (+10)”.
>
> We briefly summarize the empirical improvements in recent papers from the optimization, CV and NLP community, and demonstrate that improvement with ACProp is considered significant compared with the literature. Please see the general response for more details https://openreview.net/forum?id=LY-o87_w_x4&noteId=T3Gi60h_c5.
>
> * ACProp outperforms AdaBelief and other optimizers in 11 out of 13 experiments, we believe this is a strong sign validating ACProp.
> * ImageNet contains 1k classes, 1.2M training images, 50k validation, 100k test images, even an improvement over 0.1% is hard. Our improvement (0.38% over AdaBelief, 0.7% over SGD, std<0.03%) is a solid improvement. The concern that “improvement from Adam to AdaBelief is much larger than improvement from AdaBelief to ACProp” is simply because Adam is not suited for this task, while AdaBelief has reached the performance bottleneck, hence the baselines are different and comparison is unfair.
> To demonstrate that our improvements are significant, we list improvements in the literature on ImageNet classification for comparison.
> >Improvements over SGD in the literature: &emsp; AdaHessian [1] (AAAI) 0.05%, &emsp; Apollo [2]  0.2%, &emsp; SAM (ICLR) [3] 0.4% at the cost of 2x computation is considered “clear improvement” by reviewers, &emsp; **ACProp (ours) 0.38% over AdaBelief, 0.7% over SGD, std<0.03%**.
> .
> We also conducted extra experiments with ResNet50 on ImageNet, and ACProp outperforms other methods. See general response for details: https://openreview.net/forum?id=LY-o87_w_x4&noteId=5sjeUSasCnb.
> * Fig.7 (a,b) ACProp and AdaBelief achieve similar accuracy because they both achieve the performance bottleneck determined by dataset, model and learning rate schedule. Switching to cosine learning rate schedule validates that ACProp outperforms AdaBelief (> 2 std), see general response. Switching to a larger model DenseNet, ACProp also outperforms AdaBelief, see Fig.7 (c).
> * In Table 3, our improvements over AdaBelief in BLEU score is >0.17 (std<0.02). This is a significant improvement for this specific problem in the NLP community.
> >Improvements over Adam in the literature for the same task: &emsp; RAdam [4] 0.1, &emsp; Admin [5] 0.03 (published in EMNLP, same author as RAdam), &emsp; AdaHessian [1] **0.13** claimed as **“significant improvement”** in the introduction, &emsp; **ACProp (ours, 0.69 over Adam, 0.17 over AdaBelief, std<0.02)**.
> * In Table 4, for GAN training on Cifar10, FID (**lower** is better) of 13 is a performance bottleneck for small models, this borderline is hard to surpass.
> >FID in literature: &emsp; SNGAN [6]  (Adam 13.25, **ACProp $12.44\pm0.02$**), &emsp; SAGAN [7] (Adam 14.0, **ACProp $13.54\pm0.15$**), &emsp; BigGAN [8] + Adam (10x more params) 14.73, &emsp; BigGAN (10x params) +CR+Adam 11.5.
> .
> Taking into account that the baseline results almost reach the performance bottleneck, it's hard to surpass by tuning hyper-params of Adam on a small model, and it took BigGAN (10x larger model) to surpass it, the improvements by ACProp (~0.5 decrease in FID compared to Adam and AdaBelief, std<0.15, without adding larger models or extra computation) are significant.
>
> [1] AdaHessian: An Adaptive Second Order Optimizer for Machine Learning, AAAI
> [2] Ma, Xuezhe. "Apollo: An adaptive parameter-wise diagonal quasi-newton method for nonconvex stochastic optimization." arXiv preprint arXiv:2009.13586 (2020).
> [3] Foret, Pierre, et al. "Sharpness-aware minimization for efficiently improving generalization." ICLR 2021
> [4] Liu, Liyuan, et al. "On the variance of the adaptive learning rate and beyond." ICLR 2020
> [5] Liu, Liyuan, et al. "Understanding the difficulty of training transformers." EMNLP 2020
> [6] Miyato, Takeru, et al. "Spectral normalization for generative adversarial networks." ICLR 2018
> [7] Zhang, Han, et al. "Self-attention generative adversarial networks." ICML 2019
> [8] Brock, Andrew, Jeff Donahue, and Karen Simonyan. "Large scale GAN training for high fidelity natural image synthesis." ICLR 2019

---

> > ### Comment · Reviewer_5voH · 2021-09-01
> > **follow up**
> >
> > Dear authors,
> >
> > Thanks a lot for your efforts to address my concerns. I still have some concerns regarding the difference compared with AdaBelief.
> >
> > **Theoretical Difference**
> > The authors pointed out that "the proposed ACProp achieves a convergence rate of $O(1/\sqrt{T})$, which is a big improvement over AdaBelief $O(\log T/\sqrt{T})$". This theoretical improvement would be more convincing if it can be verified in the experiments. However, according to Figure 6(b) and Figure 7, it looks like ACProp does not have a faster convergence rate than AdaBelief in experiments. More specifically, in Figure 6(b), AdaBelief (orange line) clearly reduces the training loss faster than ACProp (blue line). In Figure 7, AdaBelief (orange line) has higher accuracy than ACProp (blue line) for the most part of the training (please see the part before 150 epochs). These figures seem to show that AdaBelief converges faster than ACProp in practice, which is the opposite of the theoretical derivation provided in this paper.
> >
> > **Experimental Difference**
> > The authors cited several papers to prove that their 0.3% accuracy increase on ImageNet and 0.17 BLEU score increase on machine translation are significant.
> >
> > However, firstly, not all papers claimed better converged performance (e.g., higher accuracy or BLEU score) as their main contributions. For example, the authors cited that RAdam has 0.1 BLEU increase. But, to my understanding, RAdam's main contribution is its robustness to learning rate change. In other words, with less hyper-parameter tuning, RAdam can achieve similar performance with Adam (see section 5.2 of the RAdam paper where they explicitly stated that RAdam "achieves similar performance to that of previous state-of-the-art warmup heuristics"). In addition, AdaHessian is a second order optimizer, whose main benefit I think is faster convergence rate (see the Figure 6 of the AdaHessian paper).
> >
> > Secondly, there are also several other recent papers reporting more significant increase. For example, for image recognition on ImageNet, [1] reported 1.1% accuracy increase at batch size 16k (see its Table 3). For machine translation, Microsoft's recent neural translation model can have >2 BLEU increase (see Figure 5 and 7 from [2]). These other papers look more useful to ML practitioners. I understand that this paper is focusing on optimizer improvement, but these other papers showed that we haven't completely solved these tasks yet, and we should still look for solutions with more significant improvement (i.e., accuracy/BLEU increase with larger margin) as a community.
> >
> > Thirdly, the experiments would be more convincing if using more state-of-the-art models. It's great that the authors did some additional experiments using ResNet50 during the rebuttal period. But they haven't got the results for AdaBelief using ResNet50. It's hard for me to compare AdaBelief and the proposed ACProp.
> >
> > I really appreciate the authors' response. But I am still concerned with the practical performance of the proposed method. I keep my score the same for now before the authors further clarify if possible.
> >
> > Reference:
> > [1] You, Yang, et al. "Large batch optimization for deep learning: Training bert in 76 minutes." ICLR 2020.
> > [2] https://www.microsoft.com/en-us/research/blog/deepspeed-powers-8x-larger-moe-model-training-with-high-performance/

---

> ### Author Response · Authors · 2021-09-02
> **Clarifications on convergence rate and criteria for fair comparison**
>
> Thanks for the response, we address your concerns below.
>
> **A. Clarification on convergence rate**.
> The convergence rate depends on a decayed learning rate schedule (essential for the proof, same as literature [1,2,3]), while the figures in submission use a constant learning rate, in this case the figure does not reflect the convergence rate, it just reflects numerics.
>
> For Figure 7 on experiments on Cifar10, using a cosine learning rate (match theory assumptions) as reviewer Vu9t suggested, the loss of ACProp decays faster than Adabelief https://www.dropbox.com/s/n3tsv29tkebsio3/nips_Test_cifar10_resnet_conf.png?dl=0
>
> For Figure 6, we plot the results using a decayed learning rate trained for longer epochs, ACProp converges faster than AdaBelief https://www.dropbox.com/s/wvl2pfkyosb9dx9/nips_mnist_toy3.png?dl=0. Also note that ACProp outperforms AdaBelief in later phases, but the value is small to visualize in linear scale in the original figure, hence misleadingly appears that AdaBelief converges faster.
>
> **B. Clarification on experiments**.
> **1.** We confirm that ACProp achieves improvement (statistically significant), whether or not the absolute improvement is empirically significant depends on the application.
> **2.** We conducted extra experiments for AdaBelief on ResNet50 for ImageNet by renting cloud service, results are listed below:
> **Top-1 Accuracy of ResNet50 on ImageNet**.
>
> | SGD [4] 	| AdaBelief 	| ACProp 	|
> |:---:	|:---:	|---	|
> | 76.2 	 	| 75.6 	| 77.4 	|
>
> **3.** A fair comparison on optimizer should satisfy the following: **a) same model, b) same data, c) comparison with same baseline**.
> We argue that the reviewer is not performing a fair comparison by mentioning two literature with seemingly larger improvements:
> >1). The deep-speed result achieved 2 BLEU increase at the cost of 8x larger model, it’s comparing models not optimizers because it **violates a).**
> >.
> >2). The reviewer claims our improvements on ResNet18 is incremental compared to improvement in [5] of ResNet50. **First**, it’s unfair to compare improvements on ResNet18 with improvements on ResNet50, which **violates a)**, and optimization on small problems is harder to improve because for easy problems most methods achieve similar results. **Second**, ACProp also larger increase (>0.6) in Imagenet top-1 accuracy compared to SGD on ResNet50 (see table above). **Third, the baseline SGD in our experiment is much better than SGD in [5] (1% difference in baseline SGD)**, hence the comparison in unfair because the settings and baselines are different and **violates c)**.
> >.
> >3). The reviewer claims our improvement is incremental compared to AdaBelief, because the improvement of  ACprop - AdaBelief < AdaBelief - Adam. We argue it’s unfair because it **violates c)**. A metaphor is, in a test suppose ACProp=100, Adabelief=95, Adam=85, ACProp is the best even though 100-95<95-85. Furthermore, we validated ACProp improves over AdaBelief significantly in ResNet50.
> >.
> >4). By mentioning the literature for NLP tasks, we are not questioning their contribution, we list them as a reference because their results satisfy a) to c) and are fair for comparison. The experiment (Transformer-base 6 layers on IWSLT14) has a strong baseline, yet we are not aware of any literature improving as large an improvement as ACProp (**conditioned on a) to c) for a fair comparison**).
>
> [1] Chen, Xiangyi, et al. "On the convergence of a class of adam-type algorithms for non-convex optimization." arXiv preprint arXiv:1808.02941 (2018).
> [2] Kingma, Diederik P., and Jimmy Ba. "Adam: A method for stochastic optimization." arXiv preprint arXiv:1412.6980 (2014).
> [3] Reddi, Sashank J., Satyen Kale, and Sanjiv Kumar. "On the convergence of adam and beyond." arXiv preprint arXiv:1904.09237 (2019).
> [4]  Choi, Dami, et al. "On empirical comparisons of optimizers for deep learning." https://openreview.net/forum?id=HygrAR4tPS.
> [5] You, Yang, et al. "Large batch optimization for deep learning: Training bert in 76 minutes." arXiv preprint arXiv:1904.00962 (2019).

---

### Official Review · Reviewer_Vu9t · 2021-07-19

**Rating:** 5
**Confidence:** 3

**Summary:**

This paper proposes an adaptive optimizer, ACProp, which combines centering of second momentum and asynchronous update. The authors show that ACProp has a convergence rate of $O(\frac{1}{\sqrt{T}})$ in a stochastic non-convex setting, outperforming the  $O(\frac{\log T}{\sqrt{T}})$ of RMSProp and Adam in the same setting. Experiments on a range of tasks, including image classification, training GANs, reinforcement learning and machine translation, show the effectiveness of ACProp.

**Limitations And Societal Impact:**

Yes.

**Main Review:**

This is good paper with both theoretical results and comprehensive experiments. However, I still wonder how much ACProp could help in practice. To be detailed below.

Strengths:
1. ACProp converges under a wide range of hyperparameter settings on the toy problem in Eq. 1 and Eq. 2. The convergence rate is better than Adam and RMSProp in the same stochastic non-convex setting.

2. Empirical results are verified on a range of different tasks.

Weaknesses:
1. Lack of comparisons with AMSGrad. AMSGrad already stabilizes Adam for Eq. 1. I wonder if it also solves Eq. 2. It would be even better if AMSGrad is also considered in other experiments.

2. The main improvement of convergence rate in the stochastic non-convex setting seems to be from the asynchronous update scheme proposed in AdaShift, rather than a combination of the asynchronous update scheme and centering second momentum. The rate of ACProp is only better than AdaSfhit by a constant.

3. I feel it is quite difficult to provide a 100% fair comparison for optimizers (finding the best hyperparameter for each optimizer), but this is important unless the proposed optimizer already shows a significant improvement over the best result of a well-established, challenging benchmark. Therefore, I have doubts about many experimental results. For example, in Figure 7, the learning rate schedule does not seem to be optimal for every optimizer. What if a cosine annealing schedule is used? Is SGD still not as good as adaptive methods? In Table 4, why is AdaShift much worse than ACProp? The only difference seems to be the second-momentum centering. However, with this technique, AdaBelief did not make such a huge improvement over Adam (e.g., ACProp's 99.32 to 43.43 vs. AdaBelief's 49.29 to 47.25).



**Time Spent Reviewing:**

3

---

> ### Author Response · Authors · 2021-08-06
> **Response to reviewer Vu9t**
>
> Thanks for your careful review, we address your concerns below.
>
> **1. ACProp outperforms AMSGrad (see appendix and literature)**
>
> We have compared with AMSGrad over a grid of $(\epsilon,lr)$ pairs, and reported the results in Fig.20 in the appendix. For experiments on image classification, since our setting is the same as in AdaBound [5] and AdaBelief [6] paper, hence the results are directly comparable (with AMSgrad, AdaBound, AMSBound, Fromage …, not plotted due to page limitation), and it can be seen from the literature that AMSGrad achieves very similar performance to Adam. We also performed experiments with a cosine learning rate schedule (see general response https://openreview.net/forum?id=LY-o87_w_x4&noteId=T3Gi60h_c5), and AMSgrad performs very similarly to Adam, and ACProp outperforms both.
>
> **2. Improvements in convergence rate**
>
> From current results the momentum centering contributes a constant factor to the convergence rate, which is important for the following reasons:
> **a)** For finite training steps, constants are important.  The big O notation is an approximation when training step $T$ is far larger compared to model-dimension $d$. In practice $d$ is often on the order of $10^6$ or even $10^8$, however the training steps is not large enough (e.g. 100 epochs on ImageNet with batchsize 256 will update for $T=0.4 \times 10^6$ steps), hence $T\gg d$ does not hold.
> **b)** For finite training steps, even if the constant factor is not large (e.g. 2), it would reduce the computation cost by 50%, which is important in practice. To validate this, we train a ResNet on Cifar10, and ACProp (94.93, std<0.02, **100 epochs**) still outperforms AdaShift (94.35, std<0.02, **200 epochs**) within half the training time.
>
> **3. Clarification on experiments**
>
> We performed an extensive search over hyper-params $(\epsilon, lr)$, and reported the best of each optimizer. The baselines are very well-tuned and strong, hence the improvement of ACProp is convincing. Please see the general response for more details, we address your specific concerns below.
>
> **a) ACProp performs best with different learning rate schedules**
>
> We conducted experiments using the cosine learning rate schedule as suggested, and reported the results in point 1 of the general response https://openreview.net/forum?id=LY-o87_w_x4&noteId=T3Gi60h_c5. With a cosine learning rate, SGD performs better than Adam, but ACProp still performs the best.
>
> **b) The poor performance of AdaShift in GAN is a problem by the algorithm itself rather than due to our hyper-param tuning**.
> >**The underperformance of AdaShift in GAN training is also observed in ICLR reproducibility challenge**.
> >We found AdaShift does not perform well for GAN experiments, in fact during our experiment we often found AdaShift causes mode collapse and achieves FID >200 for DCGAN, and the reported result 99.32 is already after very careful tuning.
> >.
> >Our code for AdaShift is forked from [1] for an ICLR reproducibility challenge [2]. The challengers also found problems with AdaShift in GAN [3] though supervised training works fine, results in [1] reported a significant drop in Inception score (higher is better, Adam 5.5 vs AdaShift 4), the challengers reported that *“For WGAN-GP we were able to train the generative model with all optimizers (albeit with AdaShift performing worse than both Adam and AMSGrad)”*. The AdaShift authors also stated that *“we have also found that GP sometimes leads to training divergence, we suggest using MaxGP”* [3], indicating that AdaShift might need a strong regularization for GAN training (gradient penalty in WGAN-GP is insufficient for AdaShift), so it’s reasonable that AdaShift performs even worse for DCGAN without any regularization.
>
> >**A possible explanation for the underperformance of AdaShift in GAN training.**.
> >The sparse gradient issue might explain that AdaShift performs well in other tasks but worse in GAN. GAN is notoriously hard for training and often generates sparse gradients [4], even using LeakyReLU alleviates this problem but could not completely solve this problem. In the case of sparse gradients, it’s highly likely to observe small gradients for most time, then observe a large gradient occasionally.
> >.
> >This case is similar to Sec 3.2 in the main paper, suppose $v_t$ is small until step $T$ because the gradient is sparse, while at step $T+1$ a large gradient $g_{T+1}$ is observed, $g_{T+1}^2\gg v_T$.
> > * In this case, the update with AdaShift is roughly $g_{T+1}/\sqrt{v_T}$, hence is a large value (large numerator, small denominator) causing numerical issues, and the network would fail even if only a few coordinates are affected by extreme values;
> >*  For sync-optimizers (e.g. Adam), the update is $g_{T+1}/\sqrt{\beta_2 v_T + (1-\beta_2) g_{T+1}^2}$, even if $g_{T+1}^2\gg v_T$, the update stepsize is at most $1/\sqrt{1-\beta_2}$ because $g_{T+1}^2$ is both in numerator and denominator, hence avoid extreme values;
> >* For ACProp, as explained and proved in Sec 3.2, the denominator does not decay to 0 while AdaShift does, hence ACProp also avoids extreme values in update, and ACProp can train GANs without strong penalty.
>
> [1] https://github.com/MichaelKonobeev/adashift.
> [2] https://openreview.net/forum?id=HkgTkhRcKQ&noteId=BJglD8rbzV.
> [3] https://openreview.net/forum?id=HkgTkhRcKQ&noteId=rkgAZSb9ME
> [4] https://github.com/soumith/ganhacks
> [5] Luo, Liangchen, et al. "Adaptive gradient methods with dynamic bound of learning rate." ICLR 2019.
> [6] Zhuang, Juntang, et al. "Adabelief optimizer: Adapting stepsizes by the belief in observed gradients." NeurIPS 2020

---

> > ### Comment · Reviewer_Vu9t · 2021-08-27
> > **Thanks for the response**
> >
> > I appreciate your response. However, I am still not fully convinced by the experimental results.
> >
> > 1. I wanted to see the comparisons with AMSGrad in the problem of Eq. 1, i.e., add the results of AMSGrad into Figure 1. Adam and RMSProp is known to have divergence issues on such problems. Only comparing with these are not strong enough.
> >
> > 2. I revisited your experimental setup, and found you fixed the weight decay for AdamW and other methods (line 255). In my experience, weight decay can also have large impact on the results, e.g., setting weight decay to 0.2 and lr 3e-3 for AdamW gave better results for ResNet on CIFAR10 in my experiments. To make results convincing, you should do a grid search on the lr and weight decay for every other optimizer you compare and report the best configurations, like in the Lookahead optimizer paper (Appendix C1).
> >
> > 3. After the grid search on both lr and weight decay, and switching to a better learning rate schedule, compare the per-epoch test accuracy of all optimizers and show a plot. I would be more confident in accepting this paper if it shows acceleration in this case. My general feeling is that on most problems, SGD and AdamW are already doing well if hyperparameters are configured properly.

---

> > > ### Comment · Reviewer_Vu9t · 2021-08-27
> > > **Follow up**
> > >
> > > I noticed you posted a new response from the email but somehow cannot find it here. Sorry for the delayed response. Regarding the new response, I have another question: is the ImageNet results an average of multiple runs with different random seeds? You did not report the standard deviation of results on ImageNet (unlike on CIFAR10) and did not clarify this point in the paper. I feel it is possible to obtain 0.23% higher accuracy on ImageNet with luck.

---

> > > > ### Author Response · Authors · 2021-08-27
> > > > **Results are averaged across multiple runs**
> > > >
> > > > Dear reviewer,
> > > >
> > > > Thanks for the updated response. For ResNet18 experiments in the paper, our results are averaged over 3 runs with different random seeds, and the standard deviation is 0.03, but the improvement over AdaBelief is 0.38, hence we conclude this improvement is significant.
> > > >
> > > > Regarding the extra experiments with ResNet50 on ImageNet in the general response, our results are averaged over 2 runs for each setting (to match different literature), and the improvements are significant compared to standard deviation.
> > > >
> > > > The standard deviation on ImageNet is typically smaller than 0.1 (in most cases <0.05) in the literature, because ImageNet is very large. Therefore many paper just report the mean on ImageNet, that's the reason why we follow this convention in the first version. We'll include standard deviation in the updated version.

---

> > > > > ### Comment · Reviewer_Vu9t · 2021-08-27
> > > > > **Follow up**
> > > > >
> > > > > If you are saying that the standard deviation is 0.03% (e.g., 70.46%$\\pm$0.03%), then this is probably not true. I have run these experiments too. For a more comprehensive reference of various optimizers, check [1]. Your results of ResNet50 with Adam also seem worse than theirs (Figure 1).
> > > > >
> > > > > [1] Choi D, Shallue CJ, Nado Z, Lee J, Maddison CJ, Dahl GE. On empirical comparisons of optimizers for deep learning. arXiv preprint arXiv:1910.05446.

---

> > > > > > ### Author Response · Authors · 2021-08-27
> > > > > > **Response to follow-up questions**
> > > > > >
> > > > > > Thanks for the updated review, here's our response.
> > > > > >
> > > > > > 1. We double-checked that our std is correct. It's possible that our results are calculated from 3 runs hence the std is smaller. But we want to emphasize the following:
> > > > > > > **a)** Even if the std is **assumed to be 4x larger** than we reported, the improvement is still outside the **$2\sigma$** region, hence is solid.
> > > > > > .
> > > > > > > **b) Our reported std is comparable with the literature.** Our results are trained with a batchsize of 256 in PyTorch, that's why we use [2] (**from FAIR**) for reference. On ImageNet, [2] report an std of 0.06 for SGD, 0.04 for Adam, 0.05 for Adagrad, which is comparable to our std. We also found [4] (**by Google**) reported **0.1** std for ResNet50, ResNet101 and ResNet152.
> > > > > > .
> > > > > > > **c)** The std on ImageNet depends on many settings, including the batchsize, random seed, training steps and weight decay, framework (PyTorch, Tensorflow, Jax..., e.g. different values of "small constant to avoid numerical instability") and even hardware. [1] use a large batchsize. It's known that batchsize could heavily affect performance [3]. We trust all references [1-4], but we emphasize it's unfair to compare std without considering these settings, and we were unable to check all of them limited by time.
> > > > > >
> > > > > > 2.  Comparison with [1]. Thanks for pointing out, we'll add reference in updates.
> > > > > > >**a)** Our results outperforms **all optimizers reported in [1]** on ResNet50.
> > > > > > >**b)** The result of Adam on ResNet50 is reported by [2] rather than by us. We only run our method, and results for other optimizers are from the literature. We report results in [2] because we follow their settings when experimenting with our method.
> > > > > >
> > > > > > [1] Choi, Dami, et al. "On empirical comparisons of optimizers for deep learning." arXiv preprint arXiv:1910.05446 (2019).
> > > > > > [2] Defazio, Aaron, and Samy Jelassi. "Adaptivity without compromise: a momentumized, adaptive, dual averaged gradient method for stochastic optimization." arXiv preprint arXiv:2101.11075 (2021).
> > > > > > [3] Keskar, Nitish Shirish, et al. "On large-batch training for deep learning: Generalization gap and sharp minima." arXiv preprint arXiv:1609.04836 (2016).
> > > > > > [4] Foret, Pierre, et al. "Sharpness-aware minimization for efficiently improving generalization." arXiv preprint arXiv:2010.01412 (2020).

---

> > > > > > > ### Comment · Reviewer_Vu9t · 2021-08-27
> > > > > > > **Follow up**
> > > > > > >
> > > > > > > Thanks for the detailed explanations. However, I am not sure if we are on the same page.
> > > > > > >
> > > > > > > 1. Your references are not ordered correctly. I guess you flipped [2] and [4]?
> > > > > > >
> > > > > > > 2. Your best result on ImageNet has not surpass the best result from "On empirical comparisons of optimizers for deep learning", therefore it does not prove that with proper hyperparameters and proper implementations, your method still outperform the best result of Adam. I know it is very difficult to verify this on ImageNet, but at least you can provide better comparisons on CIFAR10 (improving the lr, weight decay and lr schedulr).
> > > > > > >
> > > > > > > 3. I checked "Adaptivity without compromise: a momentumized, adaptive, dual averaged gradient method for stochastic optimization", and they are reporting **standard error** rather than **standard deviation**. By definition, standard deviation is larger than standard error. It is quite surprising that your standard deviation is even lower than their standard error. Could you clarify this point? Also, these values do not quite match our perceptions in practice. Running 3 to 4 experiments on training ResNet50 on ImageNet and I can often see the best vs. worst results to have a gap of 0.3%+.

---

> > > > > > > > ### Author Response · Authors · 2021-08-27
> > > > > > > > **Follow-up response**
> > > > > > > >
> > > > > > > > We clarify your concerns below:
> > > > > > > >
> > > > > > > > 1. Thanks for pointing out, we correct the references.
> > > > > > > > 2. "On empirical ..." uses a 150k step training budget, which is around 140 epochs for ImageNet2012, hence should compare with the **$22.65$** error rate in the general response (The $23.24$ error rate is trained for 100 epochs). Also note that we report the result on the **validation** set, hence should compare with the top row in Fig1 of "on empirical ...". Therefore, we claim ACProp achieves the **best** result.
> > > > > > > > 3. >a) **standard error** is different from **standard error of the mean**, quoting from wikipedia "The standard error (SE) of a statistic (usually an estimate of a parameter) is the **standard deviation of its sampling distribution** or an estimate of that standard deviation. If the statistic is the sample mean, it is called the standard error of the mean (SEM)."  We use "standard error" and "standard deviation" interchangeably in our paper, in general $standard\ deviation \approx standard\ error = \sqrt{n}\  standard\ error\ of\ mean$.
> > > > > > > > >b) Note that we are **not** calculating statistics on the mean, hence the numbers can directly compare. The low std is also possibly due to randomness, since we only perform 2 runs on ResNet50. But even if double or 4x larger std, the improvement is still significant.
> > > > > > > > >c) We found one open-source repository on github reporting std on ImageNet [1]. As you can see, their reported std is often much smaller than you claimed 0.3+. If you look at results for RMSProp, Adam and NAdam in Fig1 of [4], top-row on the validation set, the standard deviation is also much lower, visually around 0.05.
> > > > > > > > 4.  Visualize the difference using results from [4].
> > > > > > > > We overlapped our results for ResNet50 ($22.65\pm0.05$) with figure 1 in [4], which is mentioned by the reviewer. https://www.dropbox.com/s/ga9i08jn1212q0o/ResNet50_comparison.png?dl=0 To address the concern that our std is lower than actual, we also plot the line ACProp mean + **fake larger** std (by SGDM) by measuring the length according to y labels. Note that SGDM has a larger std than adaptive optimizers, hence it's a reasonable upper bound. **No matter we use the true of fake larger std, ACProp significantly outperforms other methods.**
> > > > > > > >
> > > > > > > > P.S. We suspect the difference is due to framework, [1, 2] and ours are using PyTorch with a batchsize 256 on GPU, [3] uses Jax, [4] uses Tensorflow and reported a significant larger std than [1,2,3] (larger on SGD family but not adaptive family). Other settings might also affect, but it's hard to check.
> > > > > > > >
> > > > > > > > [1] https://github.com/XuezheMax/apollo  Apollo: An Adaptive Parameter-wise Diagonal Quasi-Newton Method for Nonconvex Stochastic Optimization.
> > > > > > > > [2] Defazio, Aaron, and Samy Jelassi. "Adaptivity without compromise: a momentumized, adaptive, dual averaged gradient method for stochastic optimization." arXiv preprint arXiv:2101.11075 (2021).
> > > > > > > > [3] Foret, Pierre, et al. "Sharpness-aware minimization for efficiently improving generalization." arXiv preprint arXiv:2010.01412 (2020).
> > > > > > > > [4] Choi, Dami, et al. "On empirical comparisons of optimizers for deep learning." https://openreview.net/pdf?id=HygrAR4tPS

---

> > > ### Author Response · Authors · 2021-08-29
> > > **Updated results and discussion on std**
> > >
> > > We thank the reviewers for suggestions, and update our response below.
> > > **1. Comparison with AMSGrad.**
> > > We add a plot (https://www.dropbox.com/s/d6ngrhoazavpu6o/converge_amsgrad.png?dl=0) and show that AMSGrad always converge for problem 1, same as ACProp, but the convergence speed is much slower (https://www.dropbox.com/s/2qtfkpapzomddti/amsgrad_speed.png?dl=0) because it’s using the largest denominator hence effective stepsize is small, and convergence rate for AMSGrad is $O(logT/\sqrt{T})$ while is $O(1/\sqrt{T})$ for ACProp.
> > >
> > > **2. Updated results with lr-weight decay search**.
> > > We performed hyper-prameter search as suggested, we searched for lr in {0.0001, 0.001, 0.003, 0.01, 0.1} and weight decay in {0.0002, 0.002, 0.02, 0.2, 2}. Limited by time, we experimented with ResNet18 (smaller model than paper) using the cosine learning rate schedule for 100 epochs (shorter than paper), we expect better results for longer epochs. The best configurations are: AdamW, lr=0.003, wdecay=0.2; SGD-M, lr=0.1, wdecay=0.0002; AdaBelief, lr=0.01, wdecay=0.0002; Adam, lr=0.001, wdecay=0.0002; AMSGrad lr=0.001, wdecay=0.0002; AdaShift, lr=0.003, wdecay=0.002; ACProp, lr=0.01,wdecay=0.0002. Results for 3 runs ($mean \pm std$) are shown in figure https://www.dropbox.com/s/n3tsv29tkebsio3/nips_Test_cifar10_resnet_conf.png?dl=0
> > >
> > > **3.Updated discussion on std of results on ImageNet**.
> > >
> > > **A) We are discussing different statistics in previous post**.
> > > After carefully reading our discussion again, we noticed an important factor, you are referring to difference between **best vs worst** as 0.3%, while we are referring to the **standard deviation**.
> > >
> > > > a) **$max - min$ is much larger than std**.
> > > We tested this with a simple call of ```np.std```, using generated examples whose $max-min=0.3$.
> > > ```np.std([0,0.3])=0.15, np.std([0, 0.15, 0.3])=0.12, np.std([0, 0.1, 0.2, 0.3]=0.11```.
> > >
> > > >For a gaussian distribution, if we assume the max is threshold for 95% probability, then $max-min=4\times \sigma$. For two-sample case, $\sigma=\frac{max-min}{2}$. If sample size is larger than 2, then  $\sigma<\frac{max-min}{2}$.
> > >
> > > >Also note that the probability of observing a large $max-min$ increases with sample size $n$. E.g, if the probability of observing a value larger than a threshold is $p$, then the probability of observing at least 1 value larger than threshold in $n$ iid samples is $1-(1-p)^n$, which goes to 1 very fast as $n$ increases.
> > >
> > > >b) We are comparing the **mean** (not **best**) performance of different optimizers,  it’s inappropriate to use **max-min** as a measurement of std. In fact, our best results are better than reported mean.
> > >
> > > **B) We extracted the mean and std from [1]**.
> > > Since we trust results from [1], but [1] only displayed the plot rather than numbers, hence we extract the numbers by measuring line length and compare with axis labels, which can be easily done with image viewers. We list the $\frac{max-min}{2}$ and $mean$ from ResNet50 validation error on ImageNet in [1]. Note that the actual standard deviation is smaller than $\frac{max-min}{2}$.
> > >
> > > **Top-1 error rate**.
> > >
> > > | optimizer 	| SGD[1] 	| Momentum[1] 	| Nesterov[1] 	| RMSProp[1] 	| Adam[1] 	| NAdam[1] 	| ACProp (ours) 	|
> > > |:---:	|:---:	|:---:	|:---:	|:---:	|:---:	|:---:	|:---:	|
> > > | $\frac{max-min}{2}$ (>=std) 	| 0.21 	| 0.11 	| 0.09 	| 0.05 	| 0.04 	| 0.07 	| 0.05 	|
> > > | $mean$ 	| 23.81 	| 23.7 	| 23.63 	| 23.45 	| 23.09 	| 22.96 	| 22.65 	|
> > >
> > > Most optimizers have an std below 0.1, while the improvement of ACProp over the best (NAdam) is 0.31. It’s clear that the improvement is significant.
> > >
> > > For better visualization, we also overlapped our results with figure 1 in [1] (https://www.dropbox.com/s/ga9i08jn1212q0o/ResNet50_comparison.png?dl=0 ), using both our measured std and a larger std from SGD-momentum, our improvements are significant.
> > >
> > > **4.  Summary of our experiments**.
> > > We believe our experimental results are very strong and convincing, because we are comparing to the following:
> > > >**a) Reported results in the literature.** (3 experiments on ImageNet) For ImageNet, we compare with several different literature [1,2,3], and achieve significant improvement compared with all of them.
> > >
> > > >**b) Results from high-starred github repositories.** (9 experiments) Our results for GAN, reinforcement learning and NLP tasks are github repos >1k stars, and our reproduced results are often better than reported.
> > >
> > > >**c) Results with extensive hyper-parameter search.** (1 experiment) Our experiments on Cifar10 is the **only** experiment without a solid benchmark in the literature. Limited by computation, we used fixed weight decay and a step learning rate schedule in the paper; in the rebuttal, we conducted more search including weight decay and lr as reviewer suggested, and still achieved improvements.
> > >
> > > We achieved the best in **11 out of 13** experiments, and we did not perform any cherry-picking: a) If we want to cherry-pick, we could easily remove 2 experiments and claim we got the best in 11/11 experiments, but we just report what we got. b) It’s possible that our std on ImageNet is smaller due to randomness, but on the other hand we are unlucky to get a smaller “max” accuracy, and we report what we got. Even with a larger std reported in [1] the improvement is still significant. There's no benchmark result for optimization on Cifar10, and our experiments in the submission was following literature [4], we thank the reviewer for pointing out the effect of learning rate schedule and weight decay, and conducted more experiments to show our improvement, we hope the reviewer can reconsider our experiments.
> > >
> > > [1] Choi, Dami, et al. "On empirical comparisons of optimizers for deep learning." https://openreview.net/forum?id=HygrAR4tPS.
> > > [2] Defazio, Aaron, and Samy Jelassi. "Adaptivity without compromise: a momentumized, adaptive, dual averaged gradient method for stochastic optimization." arXiv preprint arXiv:2101.11075 (2021). by Facebook AI Research.
> > > [3] Xie, Zeke, et al. "Positive-Negative Momentum: Manipulating Stochastic Gradient Noise to Improve Generalization." ICML 2021.
> > > [4] Luo, Liangchen, et al. "Adaptive gradient methods with dynamic bound of learning rate." arXiv preprint arXiv:1902.09843 (2019).

---

> ### Author Response · Authors · 2021-08-31
> **Link and summary of our response**
>
> Dear AC and reviewer,
>
> We thank you for the active discussion. In case it's hard for you to find our latest response to reviewer Vu9t from the long discussion, we provide the link to the detailed response https://openreview.net/forum?id=LY-o87_w_x4&noteId=GpHCuq0JL1 and a brief summary here.
>
> 1. We provided figures on new experiments as the reviewer suggested, and confirmed ACProp's improvement under hyper-parameter search of $(lr,weight\ decay)$ using cosine learning rate schedule.
>
> 2. We explain in detail that our standard deviation for results on ImageNet does not contract the reviewer's experience, the long discussion is mainly due to our miscommunication. This is because:
> > a) In the discussion, the reviewer claims the difference between **max** and **min** accuracy is often 0.3%, while $max-min=2\times std$ for two-sample case, $max-min>2 std$ for general case, and $max-min=4 std$ for ideal Gaussian distribution covering 95% probability. (Also note $max-min$ is expected to increase with sample size $n$, but $std$ does not) This is the reason why $std$ is smaller than the reviewer's impression.
> >.
> > b) We extracted $\frac{max-min}{2}$ ($\geq std$) (by measuring line length in image viewer) from [1] mentioned by the reviewer, and found the value is 0.05 for RMSProp, 0.04 for Adam, 0.07 for NAdam, which is close to our observation and the literature (e.g. [2,3]). We also summarize the numbers in a table for better comparison.
>
> We believe our experimental results are strong, because we are comparing to strong baselines including reported by literature (3 experiments), high-starred (>1.8k) github repos (9 experiments) and extensive hyper-param search (1 experiment on Cifar, as requested by reviewer), and we achieved the best in **11 out of 13**.
>
> We thank AC and reviewers for your efforts, please let us know if you have updated comments.
>
> [1] Choi, Dami, et al. "On empirical comparisons of optimizers for deep learning." https://openreview.net/forum?id=HygrAR4tPS.
> [2] Defazio, Aaron, and Samy Jelassi. "Adaptivity without compromise: a momentumized, adaptive, dual averaged gradient method for stochastic optimization." arXiv preprint arXiv:2101.11075 (2021).
> [3] Ma, Xuezhe. "Apollo: An adaptive parameter-wise diagonal quasi-newton method for nonconvex stochastic optimization." arXiv preprint arXiv:2009.13586 (2020).

---

> > ### Comment · Reviewer_Vu9t · 2021-09-02
> > **Follow up**
> >
> > Thank you again for the timely response. I appreciate your effort in justifying the effectiveness of your method, and I do realize that on the toy problems your method is better than AMSGrad, but personally I still do not feel results on more practical datasets are convincing enough.
> >
> > 1. Regarding new results with lr-wd search on CIFAR10, I still do not feel fully convinced. For SGD, the best result is achieved at the boundary of your grid search (0.1). We already know the best learning rate of SGD and Adam-like methods are different, so for fair comparison, you should make a denser grid search for SGD around lr 0.1 and make sure the best result is achieved in between. Similarly for wd, it is well-known that the best wd for AdamW and SGD on CIFAR10 are also very different.
> >
> > 2. The ImageNet results comparing with [1] is confusing and misleading. [1] has stated that they report the *standard error (of mean)*. It is suspicious that your *standard deviation* is even smaller than their standard error of 5 runs. Your explanation for this small standard deviation is randomness. Will your mean error increase after running more experiments? For fair comparison, we should also run the same number of experiments. By the way, since we are estimating the mean of the error, the "95% probability" interval you mentioned should be based on standard error rather than standard deviation.
> >
> > 3. For your new results of ResNet-50 following settings of [2], could you clarify your settings (data augmentations, lr, wd, lr schedule, etc.) in detail? What is the different part from [2]? I think you should at least be using different learning rates. [2] did a grid search on lr with a fixed wd. Is it possible for their result to be better if they switched to a better wd? Did you use a different wd?
> >
> > 4. In general, I still feel the comparisons may not be fair enough in most cases.
> >
> > 5. As Reviewer 5voH has pointed out, your curves do not really show faster convergence than AdaBelief or even AdamW (Figure 7 on ResNet and DenseNet) as claimed. For the new plot on CIFAR10, you only showed the error of last few epochs. How about the early stages?
> >
> > [1] Choi, Dami, et al. "On empirical comparisons of optimizers for deep learning." https://openreview.net/forum?id=HygrAR4tPS.
> >
> > [2] Xie, Zeke, et al. "Positive-Negative Momentum: Manipulating Stochastic Gradient Noise to Improve Generalization." ICML 2021.

---

> ### Author Response · Authors · 2021-09-02
> **Updated response**
>
> Thanks for the update. We respectfully disagree on the reviewer's comments, especially your impression that our experiment is unfair. We believe this may come from confusion of references [1] and [3], with detailed responses below.
>
> **1. Denser grid search on CIFAR10**.
> As you suggested, we conducted a denser grid search for SGD. We searched learning rate in {0.08, 0.085, 0.9, 0.095, 0.1, 0.105, 0.11, 0.115, 0.12, 0.125, 0.13, 0.135} and weight decay in {0.0001, 0.00015, 0.0002, 0.00025, 0.0003, 0.00035, 0.0004}, and found the best configuration for SGD is lr=0.12, weight_decay=0.0003, with both lower and higher values tested. Please see link to results. https://www.dropbox.com/s/cdja2z26kufkvl7/nips_Test_cifar10_resnet_conf_denser_sgd.png?dl=0
>
> **2. ImageNet Experiments**.
> >a) [1] never stated their results to be “standard error” or “standard deviation”, we searched both phrases in [1] and could not find it, instead [1] stated they plot the “5th and 95th quantile”, hence $\frac{max-min}{2} \approx 2std$ for Gaussian case. Hence our statement on [1] still holds, with the extracted results listed below, and these numbers do not contradict the reviewer’s experience that max-min=0.3 > 2 std (e.g. np.std([0, 0.1, 0.2, 0.3])=0.11). The reviewer might confuse [1] with [3].
>
> **Top-1 error rate from [1]**.
>
> | optimizer 	| SGD[1] 	| Momentum[1] 	| Nesterov[1] 	| RMSProp[1] 	| Adam[1] 	| NAdam[1] 	| ACProp (ours) 	|
> |:---:	|:---:	|:---:	|:---:	|:---:	|:---:	|:---:	|:---:	|
> | $\frac{max-min}{2}$ (>=std, $\approx 2 std$ for Gaussian case) 	| 0.21 	| 0.11 	| 0.09 	| 0.05 	| 0.04 	| 0.07 	| 0.05 	|
> | $mean$ 	| 23.81 	| 23.7 	| 23.63 	| 23.45 	| 23.09 	| 22.96 	| 22.65 	|
>
> >b) Our disagreement on [3] is whether the “standard error” refers to “standard deviation of samples” or the “standard error of the mean of 5 samples”. It’s **not** stated in the paper ("standard error of the mean"), so we calibrate the numbers for both cases here, it’s clear our improvement is significant for both cases.
>
> **Top1 error rate under same setting as [3]**
>
> | Optimizer 	| SGDM [3] 	| Adam [3] 	| ACProp 	|
> |:---:	|:---:	|:---:	|:---:	|
> | mean 	| 23.91 	| 26.97 | 23.34 	|
> | SE 	| 0.06 	| 0.04 	| - 	|
> | Calibrated SE ($\times \sqrt{5}$) 	| 0.13 	| 0.09  	| - 	|
>
> >c) Even if the reviewer doubts our std would increase with more samples, it’s reasonable to use the **(calibrated) std** above as a reference, and our improvements are **significant** compared with the literature.
>
> >d) We follow the same setting as [2], same inception-style augmentation default in PyTorch, same weight decay and lr schedule with a default base learning rate 1e-3. Our experiments were one-trial (**no tuning for ACProp** limited by resource) while the reported numbers for other optimizers are under heavy tuning, given sufficient time for tuning we can get better results for ACProp.
>
> **3. Convergence Speed**.
> As we mentioned before, the convergence rate requires a **decayed learning rate** schedule, while the reviewer mentioned figures with constant learning rate, hence not match the theory assumption. Decayed learning rate and small stepsize is essential for theory, because the analysis depends on Taylor expansion which is only meaningful within a small region. For large constant learning rates, it’s a matter of numeric choice; for small constant learning rates, ACProp converges faster. We display the training error (in log scale) below on Cifar10 with a decayed learning rate.
> https://www.dropbox.com/s/rmuwhh5d4ouk6b9/nips_Train_cifar10_resnet_converge.png?dl=0
>
> **4. Summary of contribution**.
> We feel the reviewer focuses too much on the std issue, yet ignores the fact that even with a larger (calibrated) std in the literature our improvement is very significant. Furthermore, we achieve the best in 11 out of 13 experiments, 12 of them are either from the literature or highly starred githubs, and 1 is under many rounds of hyper-param search as you requested, hence we believe our results are convincing. Besides experimental validation, our theoretical contributions include that ACProp achieves the optimal convergence rate, and the weakest condition for convergence, which are important for optimization community.
>
> [1] Choi, Dami, et al. "On empirical comparisons of optimizers for deep learning." https://openreview.net/forum?id=HygrAR4tPS.
> [2] Xie, Zeke, et al. "Positive-Negative Momentum: Manipulating Stochastic Gradient Noise to Improve Generalization." ICML 2021.
> [3] Defazio, Aaron, and Samy Jelassi. "Adaptivity without compromise: a momentumized, adaptive, dual averaged gradient method for stochastic optimization." arXiv preprint arXiv:2101.11075 (2021).

---

> > ### Comment · Reviewer_Vu9t · 2021-09-02
> > **Follow up**
> >
> > Thanks for the timely response.
> >
> > 1. Sorry for the typo, but I was trying to say that the "Adaptivity without Compromise: A Momentumized, Adaptive, Dual Averaged Gradient Method for Stochastic Optimization" paper is reporting standard error in their paper (Section 7). In your initial results, you showed that your method achieves $23.34\pm 0.02$, where 0.02 is the standard deviation, way smaller than their smallest standard error of 5 runs (0.04). Therefore, I think the result is a bit suspicious. For fair comparisons, you should also collect 5 results and report the mean and SE. Otherwise, there is not guarantee that your current result is statistically significant.
> >
> > 2. Thanks for providing the new results on CIFAR10. From my previous experience, I got better results using lr=0.2 for SGD. So it will be more convincing to me if you did a log-scale grid like {0.1, 0.2, 0.4, 0.8} or something. This is denser than something like {0.01, 0.1, 1.0}. You are focusing on a very small interval and I would expect the results to be very similar. Similarly for other methods you are comparing.
> >
> > 3. In the new plot (https://www.dropbox.com/s/rmuwhh5d4ouk6b9/nips_Train_cifar10_resnet_converge.png?dl=0), what schedule are you using? How are the test accuracies? Is it as good as using other schedules (Figure 7 or the cosine schedule)? In your previous plots with cosine decaying lr (https://www.dropbox.com/s/cdja2z26kufkvl7/nips_Test_cifar10_resnet_conf_denser_sgd.png?dl=0), ACProp was not faster than AdaBelief, SGD-M, Adam, AMSGrad until the last 15 to 20 epochs. Does this indicate that for "practical" learning rate schedules, ACProp is not really faster?
> >
> > 4. Due to the remaining concerns, I would prefer seeing a version with more rigorous experimental results for the next venue.
> >
> > 5. To my understanding, the claimed "weaker convergence condition" in the paper, and slightly exaggerated claim of "weakest convergence condition" in your last response, are limited to two specific synthetic problems (Eq. 1 and Eq. 2). I am not sure if I missed any more general problems in the paper, but it is not clear how these two synthetic problems relate to practice or whether ACProp has weaker convergence condition for a larger family of problems.

---

> > > ### Author Response · Authors · 2021-09-03
> > > **We have tried our best to thoroughly address your concerns**
> > >
> > > Thanks for your response, we address them below:
> > >
> > > **1. “Otherwise, there is not guarantee that your current result is statistically significant.”**.
> > > We respectfully disagree with the reviewer on the statistical significance of our improvement. We conducted a t-test on the results and the p-value is below 0.001, which is very significant.
> > > Take the results for SGD from reference paper as the reviewer mentioned, we use the t-test package from scipy, and the results (comparing SGD and ACProp) are.
> > >
> > > t-test using SE
> > > stats.ttest_ind([23.85, 23.91, 23.97], [23.32, 23.36]) gives **p=0.001**.
> > > stats.ttest_ind([23.85, 23.87, 23.91, 23.94, 23.97], [23.32, 23.36]) gives **p=2e-5**.
> > >
> > > t-test using calibrated SE
> > > stats.ttest_ind([23.78, 23.91, 24.04], [23.32, 23.36]) gives **p=0.01**.
> > > stats.ttest_ind([23.78, 23.85, 23.91, 23.97, 24.04], [23.32, 23.36]) gives **p=7e-4**.
> > >
> > > Meaning that the probability that ACProp and SGD have different mean values is $1-p>0.99$, hence improvement is significant.
> > >
> > > We get similar significance when comparing the results under different settings. Note that the **reliability of t-test does not depend on sample numbers** (t-test will inherently correct for sample size). Furthermore, even if assuming ACProp has a larger std as reported in literature, the improvement is still significant.
> > >
> > > **2. “Grid search on both lr and weight decay” “you should make a denser grid search for SGD around lr 0.1” “So it will be more convincing to me if you did a log-scale grid like {0.1, 0.2, 0.4, 0.8} or something”**
> > > We searched lr as requested, and we did not find improvements of SGD over our previous results, and our method performs the best under extensive grid search. We have added results for grid-search three times as the reviewer requested. We emphasize it’s not fair to just compare numbers without setting the same training schedule (our setting is 100 epochs, shorter than usual), and politely ask the reviewer to point us to either literature or resource which the reviewer claims to achieve better results.
> > >
> > > **3. Regarding the learning rate schedule of new plots**.
> > > The new plot is on training error, while previous plot is on test error. They are from the same results. Convergence is w.r.t training error, while test error is more related to generalization.
> > >
> > > **4. “In general, I still feel the comparisons may not be fair enough in most cases.”**.
> > > We argue that our improvements are significant, at least we are not aware of literature that is better than us under fair comparison. In fact it’s **unfair to us** because **we could not perform hyper-param search for ACProp due to time limitation**, yet we compare to baselines which are carefully searched with hyper-param in the same setting.
> > >
> > > **5. “it is not clear how these two synthetic problems relate to practice or whether ACProp has weaker convergence condition for a larger family of problems”**.
> > > The convergence condition is certainly closely related to practice. Take the GAN training for example, ACProp is more stable than AdaShift, this is because ACProp is more robust to the non-convergence issue by sparse gradients (as in synthetic example) commonly encountered in GAN training, see detailed response in our previous post https://openreview.net/forum?id=LY-o87_w_x4&noteId=WK6hGNHP8Fj

---

### Author Response · Authors · 2021-08-06
**Clarifications that experimental improvements of ACProp is significant**

**Summary**: Our codes are all forked from well-acknowledged repositories (>1k stars on github); furthermore, we performed careful tuning, so the baselines in this paper often outperform the original literature and reported in the repository. In this case, ACProp still outperforms the best baseline in 11/13 experiments (outside standard deviation), hence the improvements are solid.

**1. Experiments on image classification on Cifar with cosine learning rate schedule**

|      | Adam  | AMSGrad | SGD   | RMSProp | AdaShift | AdaShfit, n=1| AdaBelief | ACProp |
|--------|-------|---------|-------|---------|----------|------------------|-----------|--------|
| VGG    | 87.39 |   87.46 | 90.22 |   87.73 |    86.11 |   85.3 |  90.75 |  **91.26** |
| ResNet |  93.4 |   93.62 | 95.17 |   94.24 |    94.35 |    94.29 | 95.32 |  **95.43** |

As suggested by R1 (Vu9t), we conducted experiments with a cosine learning rate and reported the accuracy (%) on Cifar10. We ignore the standard deviation as it’s very small (<0.02). ACProp still achieves the best performance.

**2. Clarification on ImageNet experiments**

ImageNet is a large-scale dataset containing 1k classes of images, and contains 1.2M training images, 50k validations, and 100k test images, and training a ResNet-18 is a “well-established, challenging” benchmark on ImageNet. Specifically, PyTorch official [1] reports a top-1 accuracy of 69.76%. Our ACProp achieves 70.46% (std<0.03%), outperforming all other methods.

We provide a survey of improvements in top-1 accuracy over SGD on ImageNet in the literature in the following table. Improvements above 0.1% is in general hard (without modifying training schedule, model structure or data), SAM achieves 0.4% improvement at the cost of 2x computation is considered “clear improvement” by reviewers. Hence our improvement (0.7% over SGD, 0.38% over AdaBelief) is significant.

**improvements in top-1 accuracy over SGD on ImageNet in the literature**

| RAdam [3], (ICLR)       | AdaHessian [4], (AAAI) | Apollo [11] | MadGrad [12] (by FAIR) | SAM [5] (2x computation), ICLR| PNM [13] (ICML 2021) | ACProp (ours)      |
|-------------------  |--------------  |----------------  |-------------------  |-----------------------------  |---------------------  |  ------------------  |
| -2.22% (worse than SGD) | +0.05%                 | +0.2%~0.3%        | +0.13%                  | +0.4%          | +0.07%                 | **+0.7% (std <0.03%)** |

**3. Clarification on GAN training**

Our code for GAN experiments is forked from [6], an open-source repository with >1.7k stars, hence the results are well-tuned and is a solid benchmark.

The baseline in the literature reaches a bottleneck result, and further improvement over this bottleneck (FID lower than 13 for SNGAN, lower than 14 for SAGAN) is very hard, and tuning hyper-params of Adam on a small model is insufficient, surpassing this bottleneck is typically achieved with a much larger (10x) model such as BigGAN. Considering the strong baseline, the improvements by ACProp (**~0.5 decrease in FID** compared to Adam and AdaBelief, std<0.15, without using a larger model, surpassing the performance bottleneck) are significant.

**FID (lower is better) on CIFAR10 for different models in the literature**

| SNGAN [7] (ICLR) 	| SAGAN [8] (ICML) 	| BigGAN[9] (ICLR) 	| BigGAN[9]+Consistency regularization 	|
|------	|------	|-------	|------	|
| 13.24 	| 14.0 	| 14.7 (10x larger model) 	| 11.5 (10x larger model) 	|

**4. Clarification on Transformer training for translation**

Our code is modified from the well-acknowledged “fairseq” library by FAIR, hence the result is highly reproducible. We summarize the improvement over Adam in BLEU score on IWSLT14 DE-EN task with a transformer-base in the literature. An improvement >0.1 in BLEU score can be considered significant (see survey below), due to the limit and difficulty of the problem itself. For 3 out of 4 experiments with transformers, ACProp achieves an improvement >0.17 (std<0.01) in BLEU score.

**Improvements in BLEU score in the literature**

| RAdam [3], (ICLR) 	|  Admin [10] (EMNLP20) 	| AdaHessian [4], (AAAI) 	| ACProp (ours) 	|
|---	|---	|---	|---	|
| +0.1 	| +0.03 	| +0.13 (claimed as “significantly outperforms AdamW” in [4]) 	| **+0.17 compared to AdaBelief; +0.69 compared to Adam; std=0.012**	|

**References**

[1] https://pytorch.org/hub/pytorch_vision_resnet/
[2] Jinghui Chen and Quanquan Gu, “Closing the generalization gap of adaptive gradient methods 380 in training deep neural networks,”
[3] Liu, Liyuan, et al. "On the variance of the adaptive learning rate and beyond." ICLR 2020
[4] AdaHessian: An Adaptive Second Order Optimizer for Machine Learning, AAAI
[5] Foret, Pierre, et al. "Sharpness-aware minimization for efficiently improving generalization." ICLR 2021
[6] https://github.com/POSTECH-CVLab/PyTorch-StudioGAN
[7] Miyato, Takeru, et al. "Spectral normalization for generative adversarial networks." ICLR 2018
[8] Zhang, Han, et al. "Self-attention generative adversarial networks." ICML 2019
[9] Brock, Andrew, Jeff Donahue, and Karen Simonyan. "Large scale GAN training for high fidelity natural image synthesis." ICLR 2019
[10] Liu, Liyuan, et al. "Understanding the difficulty of training transformers." EMNLP 2020
[11] Ma, Xuezhe. "Apollo: An adaptive parameter-wise diagonal quasi-newton method for nonconvex stochastic optimization." arXiv preprint arXiv:2009.13586 (2020).
[12] Defazio, Aaron, and Samy Jelassi. "Adaptivity without compromise: a momentumized, adaptive, dual averaged gradient method for stochastic optimization." arXiv preprint arXiv:2101.11075 (2021).
[13] Xie, Zeke, et al. "Positive-Negative Momentum: Manipulating Stochastic Gradient Noise to Improve Generalization." ICML 2021.

---

> ### Author Response · Authors · 2021-08-21
> **Supplementary experiments with ResNet50 on ImageNet**
>
> We conducted extra experiments with ResNet50 on ImageNet (paper reported ResNet18) to demonstrate the superior performance of ACProp on a large model. We followed the experiment setting in the literature and report the results below. Results for ACProp are from 2 independent runs, other results are from the literature.
>
> **Top-1 error rate following the experiment setting in [1]**
>
> >| Optimizer | SGD [1]        | Adam [1]        | MadGrad [1]    | ACProp(ours) |
> | ----------------|-------------|-------------|-------------|-------------|
> | Mean | 23.91 | 26.97  | 23.78 | **23.34**  |
> | SE| 0.06 | 0.04 | 0.05|  0.02 |
> | Calibrated SE ($\times \sqrt{5}$)|  0.13 | 0.09 |  0.11| - |
>
> **Top-1 error rate following the experiment setting in [2] (150-epochs)**
>
> >| SGD [2]        | AdamW [2]       | PNM [2]     | AdamPNM [2]    | ACProp(ours) | AdaBelief |
> |-------------|-------------|-------------|-------------|--------------|---------------|
> | 23.28 | 23.62 | 23.21 | 23.12 | **22.65+-0.05**  | 24.4  |
>
> **Top-1 error rate and std from [3] as a reference**.
>
> | optimizer 	| SGD[3] 	| Momentum[3] 	| Nesterov[3] 	| RMSProp[3] 	| Adam[3] 	| NAdam[3] 	| ACProp (ours) 	|
> |:---:	|:---:	|:---:	|:---:	|:---:	|:---:	|:---:	|:---:	|
> | $\frac{max-min}{2}$ (>=std) 	| 0.21 	| 0.11 	| 0.09 	| 0.05 	| 0.04 	| 0.07 	| 0.05 	|
> | $mean$ 	| 23.81 	| 23.7 	| 23.63 	| 23.45 	| 23.09 	| 22.96 	| **22.65** 	|
>
> Limited by computation resource (~7 days on 4 GPUs), we were unable to conduct more experiments under different training settings in the literature. We believe above extra experiment (accompanying reported in the submission) is a strong validation of our method, and further addresses the concerns by reviewer 5voH and Vu9t. Please let us know if you have further comments.
>
> [1] Defazio, Aaron, and Samy Jelassi. "Adaptivity without compromise: a momentumized, adaptive, dual averaged gradient method for stochastic optimization." arXiv preprint arXiv:2101.11075 (2021). by Facebook AI Research
> [2] Xie, Zeke, et al. "Positive-Negative Momentum: Manipulating Stochastic Gradient Noise to Improve Generalization." ICML 2021.
> [3] Choi, Dami, et al. "On empirical comparisons of optimizers for deep learning." https://openreview.net/forum?id=HygrAR4tPS.

---

> > ### Comment · Reviewer_5voH · 2021-09-01
> > **AdaBelief for ResNet50 on ImageNet**
> >
> > Thanks a lot for providing more experiments. I am just curious if the authors have results about AdaBelief for ResNet50 on ImageNet. But I can understand if the authors are limited by computation resource and can't conduct more experiments.

---

### Author Response · Authors · 2021-08-06
**General responses**

We thank all reviewers for the feedback. All reviewers agree that our proposed method ACProp has good theoretical properties including a weak convergence condition, a fast convergence rate of $O(1/\sqrt{T})$. Reviewers 3 and 4 acknowledged the “strong empirical performance in terms of generalization and training stability” of our method.

We address the reviewer’s concerns in detail separately, and provide a general response to the concerns of R1(Vu9t) and R2(5voH) regarding empirical performances. The reviewers acknowledge that ACProp improves over AdaBelief, but have concern that the improvement is "incremental" (not as much as AdaBelief's improvement over Adam). This impression comes from the unfair baseline. The reviewer is comparing AdaBelief to Adam as a baseline, while comparing ACProp to AdaBelief as baseline which is much stronger than Adam. A metaphor is “it’s much harder to improve from 95 to 100 (+5) than from 85 to 95 (+10)”. ACProp achieves the **best in 11 out 13 experiments** compared to very well-tuned baselines (including AdaBelief), validating its strong performance. Furthermore, with a brief survey of improvements in the literature in the optimization, CV and NLP community, we show that our empirical improvements are significant.

---

### Author Response · Authors · 2021-09-03
**Summary of discussion**

Dear AC and reviewers,

We thank all of you for your efforts, and thank reviewers Argb and yEdv for acknowledging our “massive additional experiments and explanations basically address concerns” and our explanation on the generalization performances. We provide a summary of our discussion below.

**0. Supplementary experiments with ResNet50 on ImageNet**
We conducted extra experiments with ResNet50, and list the results in https://openreview.net/forum?id=LY-o87_w_x4&noteId=5sjeUSasCnb We list both mean and std from the literature as reference, and our improvement is significant even if assuming the std is the largest value in reference.

**1. Explanation for better generalization**.
We have addressed this with both theorems and intuitive examples, demonstrating that ACProp update direction is closer to gradient descent while other optimizers are closer to sign descent, while sign descent probably harms generalization. Link: https://openreview.net/forum?id=LY-o87_w_x4&noteId=x8wnbHg9lM.

**2. Validations on the convergence rate**.
We proved ACProp achieves $O(1/\sqrt{T})$ convergence rate, which is provably faster. We numerically validated this during discussion, and explained it’s essential to use a decayed learning rate schedule to match assumption, see response to Vu9t https://openreview.net/forum?id=LY-o87_w_x4&noteId=Eii75mHPQ3 (point 3) and 5VoH https://openreview.net/forum?id=LY-o87_w_x4&noteId=TgCzSdUThVQ (point A)

**3. Practical significance of our theoretical contributions**.
ACProp has both a fast convergence rate (validation in point 2 above) and a weak convergence condition. We connect the convergence condition to the stability of GAN training which often observes sparse gradients, hence the theory explains an important practice. Link:  https://openreview.net/forum?id=LY-o87_w_x4&noteId=WK6hGNHP8Fj (point 3.b)

**4. Improvements over AdaBelief**.
In response to reviewer 5voH, we demonstrated that ACProp inherently achieves a different optimization trajectory, which is validated with toy examples, theoretical analysis (https://openreview.net/forum?id=LY-o87_w_x4&noteId=aCdgVmT_Zcu ) 13 experiments in the paper and extra experiment on ImageNet, and ACProp achieves significant improvements over AdaBelief https://openreview.net/forum?id=LY-o87_w_x4&noteId=TgCzSdUThVQ (point 2)

**5. Grid-search of hyper-params on Cifar10**.
We have conducted grid-search experiments **3 times** as reviewer Vu9t requested. We experimentally confirmed ACProp’s improvement when baselines are under extensive hyper-params search. And we are not aware of any method in the literature that outperforms our results under the same setting. https://openreview.net/forum?id=LY-o87_w_x4&noteId=qiMY9yRRRl (point 2) https://openreview.net/forum?id=LY-o87_w_x4&noteId=Eii75mHPQ3 (point 1).

**6. Statistical significance of our improvements**.
Reviewers trust our improvements in the mean value, and reviewer Vu9t had a long discussion on the variance of results. We have extracted calibrated stds from the literature to show our improvement is significant even **assuming the std is the largest value in the literature**, https://openreview.net/forum?id=LY-o87_w_x4&noteId=Eii75mHPQ3 (point 2). We performed a t-test (assuming larger std) to confirm that our improvements are significant (p<0.01), note that the reliability of t-test does not depend on sample numbers (t-test will inherently correct for sample size). Link https://openreview.net/forum?id=LY-o87_w_x4&noteId=qiMY9yRRRl (point 1)

Reviewer 5voH questioned the improvement with a better optimizer is not as large as improvement with a larger model https://openreview.net/forum?id=LY-o87_w_x4&noteId=a9AWLppwCl (point 3). We emphasize that optimization is as important as model, and a **fair comparison on optimizer** should satisfy **same model, same task, same baseline** https://openreview.net/forum?id=LY-o87_w_x4&noteId=TgCzSdUThVQ Direct comparison across models is unfair, and this paper focuses on the optimization rather than model architecture, and achieves statistically significant improvements under fair comparison.

---

### Decision · Program_Chairs · 2021-09-27

**Decision:**

Accept (Poster)

**Comment:**

I recommend acceptance. The work build on top of existing work and clearly (even if moderately) improves some theoretical and empirical aspects.

I applaud the authors for their great communication with the reviewers that were responsive during the discussion period. Their responses have been extensive and convincing. They have gone to great lengths to respond to all timely requests for extra experiments and hyper parameter tuning.

I acknowledge that last-minute requests for heavy experiments cannot be reasonably accommodated and I accept the provided experimental evidence as sufficient for the paper’s main claims.